# A Fyn biosensor reveals pulsatile, spatially localized kinase activity and signaling crosstalk in live mammalian cells

Ananya Mukherjee[1,2†], Randhir Singh[1†], Sreeram Udayan[1], Sayan Biswas[1], Pothula Purushotham Reddy[3], Saumya Manmadhan[1], Geen George[1], Shilpa Kumar[1], Ranabir Das[3], Balaji M Rao[4]*, Akash Gulyani[1]*

[1]Institute for Stem Cell Science and Regenerative Medicine, Bangalore, India; [2]SASTRA University, Thanjavur, India; [3]National Centre for Biological Sciences, Bangalore, India; [4]North Carolina State University, Raleigh, United States

**Abstract** Cell behavior is controlled through spatio-temporally localized protein activity. Despite unique and often contradictory roles played by Src-family-kinases (SFKs) in regulating cell physiology, activity patterns of individual SFKs have remained elusive. Here, we report a biosensor for specifically visualizing active conformation of SFK-Fyn in live cells. We deployed combinatorial library screening to isolate a binding-protein (F29) targeting activated Fyn. Nuclear-magnetic-resonance (NMR) analysis provides the structural basis of F29 specificity for Fyn over homologous SFKs. Using F29, we engineered a sensitive, minimally-perturbing fluorescence-resonance-energy-transfer (FRET) biosensor (*FynSensor*) that reveals cellular Fyn activity to be spatially localized, pulsatile and sensitive to adhesion/integrin signaling. Strikingly, growth factor stimulation further enhanced Fyn activity in pre-activated intracellular zones. However, inhibition of focal-adhesion-kinase activity not only attenuates Fyn activity, but abolishes growth-factor modulation. *FynSensor* imaging uncovers spatially organized, sensitized signaling clusters, direct crosstalk between integrin and growth-factor-signaling, and clarifies how compartmentalized Src-kinase activity may drive cell fate.

*For correspondence:
bmrao@ncsu.edu (BMR);
akashg@instem.res.in (AG)

[†]These authors contributed equally to this work

**Competing interests:** The authors declare that no competing interests exist.

## Introduction

Regulation of cell fate and behavior is achieved through complex and interconnected signaling networks acting in concert (*Cai et al., 2014*; *Devreotes et al., 2017*; *Ridley et al., 2003*). For precise signaling, activities of key signaling proteins are tightly compartmentalized in the cell, both spatially and temporally (*Depry et al., 2015*; *Gulyani et al., 2011*; *Komatsu et al., 2011*). Especially important to cellular controls are signaling nodes that function downstream of multiple receptor classes and help in signal integration. Src family kinases (SFKs) are such key signaling nodes; activated by cell adhesion receptors, integrins, receptor tyrosine kinases (RTKs, including growth factor receptors) and Gprotein-coupled receptors (GPCRs) among others (*Abram and Courtneidge, 2000*; *Giannone and Sheetz, 2006*; *Grande-García et al., 2007*; *Parsons and Parsons, 2004*; *Thomas and Brugge, 1997*). Src-kinases critically influence cell fate; regulating cell shape, migration and adhesion, survival and growth, stemness and differentiation making them important therapeutic targets in multiple diseases (*Chetty et al., 2015*; *Gujral et al., 2014*; *Kim et al., 2009*; *Lewis-Tuffin et al., 2015*; *Nygaard et al., 2014*; *Saad, 2009*; *Timpson et al., 2001*; *Zhang et al., 2013*; *Zhang et al., 2014*). Despite their importance, intracellular activity patterns of individual Src family kinases with spatial and temporal precision are still unclear. Visualization of active kinases in cells is essential to understand how Src kinases integrate signals and regulate multiple, and sometimes opposing processes, with fidelity and precision.

**eLife digest** Cells contain networks of signaling proteins that can respond to a variety of cues from the surrounding environment. Often the cell's response to these cues is not just controlled by the level of protein, but by changing the activity of signaling proteins. For example, a signaling protein in humans and other mammals known as Fyn regulates a number of different processes, including when a cell grows, dies, or develops a specialist role.

Defects in the activity of Fyn are associated with several diseases in humans including cancer and Alzheimer's disease. However, it remains unclear how Fyn contributes to these diseases, or how the protein is able to precisely coordinate responses to multiple different cues in healthy individuals. This is largely because there are no readily available tools that are able to specifically detect where and when this protein is active in cells.

Researchers often use fluorescent proteins called biosensors as tools to detect where specific proteins are located in living cells over time. Now, Mukherjee, Singh et al. have developed a new biosensor named FynSensor to monitor the active form of Fyn in mammalian cells.

Microscopy imaging of FynSensor in several different cell types showed that although Fyn was present everywhere, it was only active in certain areas. In these areas the protein switched between an active and inactive state, with clear 'pulses' of signaling activity lasting a couple of minutes in response to specific cues. These areas of high Fyn activity behaved like signaling hubs in which several different cues integrate together before Fyn triggers an appropriate cell response.

These results shed light on how Fyn is able to precisely control many different processes in cells. In the future, *FynSensor* could be used to rapidly screen for drug-like molecules to treat cancer, Alzheimer's disease and other conditions linked with defects in Fyn activity. Furthermore, the *FynSensor* could be adapted to allow researchers to study other signaling proteins in humans and other animals.

Currently available fluorescent SFK sensors have severe limitations. Most available biosensors, especially genetically encoded ones, do not directly report the intracellular distribution of active kinases since they rely on detecting the phosphorylation of a 'pseudosubstrate' peptide (*Liao et al., 2012*; *Ouyang et al., 2008*; *Ouyang et al., 2019*; *Seong et al., 2011*; *Wang et al., 2005*). As a result, biosensor readout – extent of phosphorylation of 'sensor' peptides – can be affected by the activities of kinases as well as cellular phosphatases. Confounding readouts further, it has been reported that sensors can get trapped in an 'ON' state owing to strong intramolecular interactions between phosphorylated substrate peptides and their respective recognition motifs (*Komatsu et al., 2011*; *Regot et al., 2014*); and therefore are unable to report the turning OFF of kinase activity. Another major drawback of current sensors is that multiple kinases, especially closely related ones, can phosphorylate these substrate-based sensors in a promiscuous manner leading to a lack of specificity.

Study of SFK activity is especially complicated by the presence of multiple closely related Src family members, including three ubiquitously expressed kinases c-Src, c-Yes and Fyn. While there is some functional redundancy, individual SFKs also perform critical, exclusive roles in the cell (*Zhang et al., 2014*; *Kuo et al., 2005*; *Lowe et al., 1993*; *Lowell and Soriano, 1996*; *Marchetti et al., 1998*; *Molina et al., 1992*; *Palacios-Moreno et al., 2015*). Therefore, despite decades of study, it is not always clear which kinase(s) is activated in a given cellular context? In this regard, fluorescent biosensors that report the activation of individual Src kinases in live cells would be extremely valuable in clarifying kinase activity and function. However, such specific biosensors for individual Src kinases are still limited (*Gulyani et al., 2011*; *Koudelková et al., 2019*; *Paster et al., 2009*; *Stirnweiss et al., 2013*), with current sensors generally giving only a pan-SFK readout with limited spatial and temporal resolution.

Here, we present a fluorescent biosensor specific for the SFK-Fyn. Fyn, is ubiquitously expressed and regulates cell migration, epithelial to mesenchymal transition (EMT), cancer metastasis, immune-response, axonal guidance and patterning, and synaptic functions (*Gujral et al., 2014*; *Du et al., 2016*; *Lewin et al., 2010*; *Meriane et al., 2004*; *Posadas et al., 2016*; *Salter and Kalia, 2004*). Interestingly, among several SFKs implicated, recent evidence shows Fyn may specifically control

EMT and metastatic progression (*Gujral et al., 2014*). Overall, Fyn is a critical player in multiple cell/ tissue types, integrating signaling through multiple receptor classes (*Palacios-Moreno et al., 2015*; *Martín-Ávila et al., 2016*; *Yadav and Denning, 2011*) and controlling cell fate. However, there is currently no tool, including specific antibodies, to visualize Fyn activity in cells. As a consequence, there is no information on intracellular Fyn dynamics in live cells. A Fyn biosensor would bridge this gap and offer a method to address Fyn functional complexity as a key signaling node.

Generation of biosensors is often limited by a lack of naturally available binders that can detect active states of proteins in real-time. To address this, we have developed a strategy of using artificially engineered binders that recognize the active form of target proteins in live cells (*Gulyani et al., 2011*). Like other SFKs (*Arold et al., 2001*; *Boggon and Eck, 2004*; *Kinoshita et al., 2006*; *Noble et al., 1993*; *Sicheri and Kuriyan, 1997*), Fyn is maintained in an inactive or closed form through multiple intramolecular interactions (*Thomas and Brugge, 1997*). Activation of the kinase involves disruption of these interactions leading to an open conformation, with more exposed surfaces (*Gulyani et al., 2011*; *Young et al., 2001*). Taking advantage of this, we screened a combinatorial protein library to isolate a binding protein (F29) that specifically binds to a region of Fyn (SH3 domain) that is more exposed in the open conformation. For screening, we chose a library generated by mutagenizing the Sso7d protein - a highly stable protein derived from the hyperthermophilic archaeon *Sulfolobus solfataricus*. We have previously shown that Sso7d mutant-libraries can be used to isolate binding proteins for a wide range of targets (*Gera et al., 2012*; *Gera et al., 2011*; *Gocha et al., 2017*; *Hussain et al., 2013*). Also, since Sso7d has no known interactions with mammalian intracellular proteins, we reasoned that Sso7d would serve as an ideal 'inert' scaffold for generating an intracellular biosensor. Our analysis shows that F29 indeed binds specifically to Fyn, with little to no cross-reactivity with other SFKs. Nuclear magnetic resonance (NMR) spectroscopy analysis of F29 in complex with the target reveals the structural basis of the specificity of F29 binding to Fyn.

An efficient readout of the molecular recognition of the target by the binding protein is central to constructing an effective biosensor. Here, we used our new binder/F29 to construct a genetically encoded Fyn biosensor (*FynSensor*) based on fluorescence resonance energy transfer (FRET). *FynSensor* is robust, sensitive and faithfully reports Fyn activation and regulation. We also show how *FynSensor* expression in cells is minimally perturbing, with no measurable changes in either cellular morphodynamics or downstream signaling. Significantly, *FynSensor* enables direct visualization of active Fyn in live cells, revealing Fyn activation to be spatially compartmentalized, polarized and pulsatile. To the best of our knowledge, this is the first reported visualization of activated Fyn in live cells; notably, unavailability of suitable antibodies currently precludes even routine detection of active Fyn. Strikingly, *FynSensor* imaging shows the presence of spatially localized signaling clusters sensitive to integrin and growth factor signaling; and reveals how growth factor response in cells is influenced by integrin-dependent protein activity. More broadly, *FynSensor* shows how spatially compartmentalized activation of key signaling proteins may lead to efficient signal integration and precise control of cell physiology. Further, our results provide a framework for systematic development of intracellular biosensors for specific Src-family and other kinases in general.

## Results

### Screening yields binder specific for active Fyn

Specifically recognizing active Fyn is critical for constructing an intracellular sensor. Activation of SFKs is accompanied by large conformational changes; the active kinase adopts a more open conformation with significantly reduced intramolecular interactions (*Thomas and Brugge, 1997*). Specifically, prior evidence (*Gulyani et al., 2011*; *Young et al., 2001*) suggests that the SH3 domain in SFKs is more accessible in the active conformation relative to the closed, inactive state (*Figure 1A*). Therefore, a binding protein targeting the SH3 domain of Fyn can be expected to preferentially bind the active form of the kinase (*Gulyani et al., 2011*). Such an active state binder can then be engineered as a fluorescent biosensor for reporting intracellular Fyn activation (*Figure 1A,B*). Accordingly, we sought to generate a binding protein specific for Fyn SH3 domain. To this end, we screened a yeast surface display library of Sso7d mutants (library generated by randomizing ten Sso7d residues), using magnetic screening and fluorescence associated cell sorting (FACS), adapting methods described previously (*Gera et al., 2011*; *Gera et al., 2013*) (*Figure 1—figure supplement*

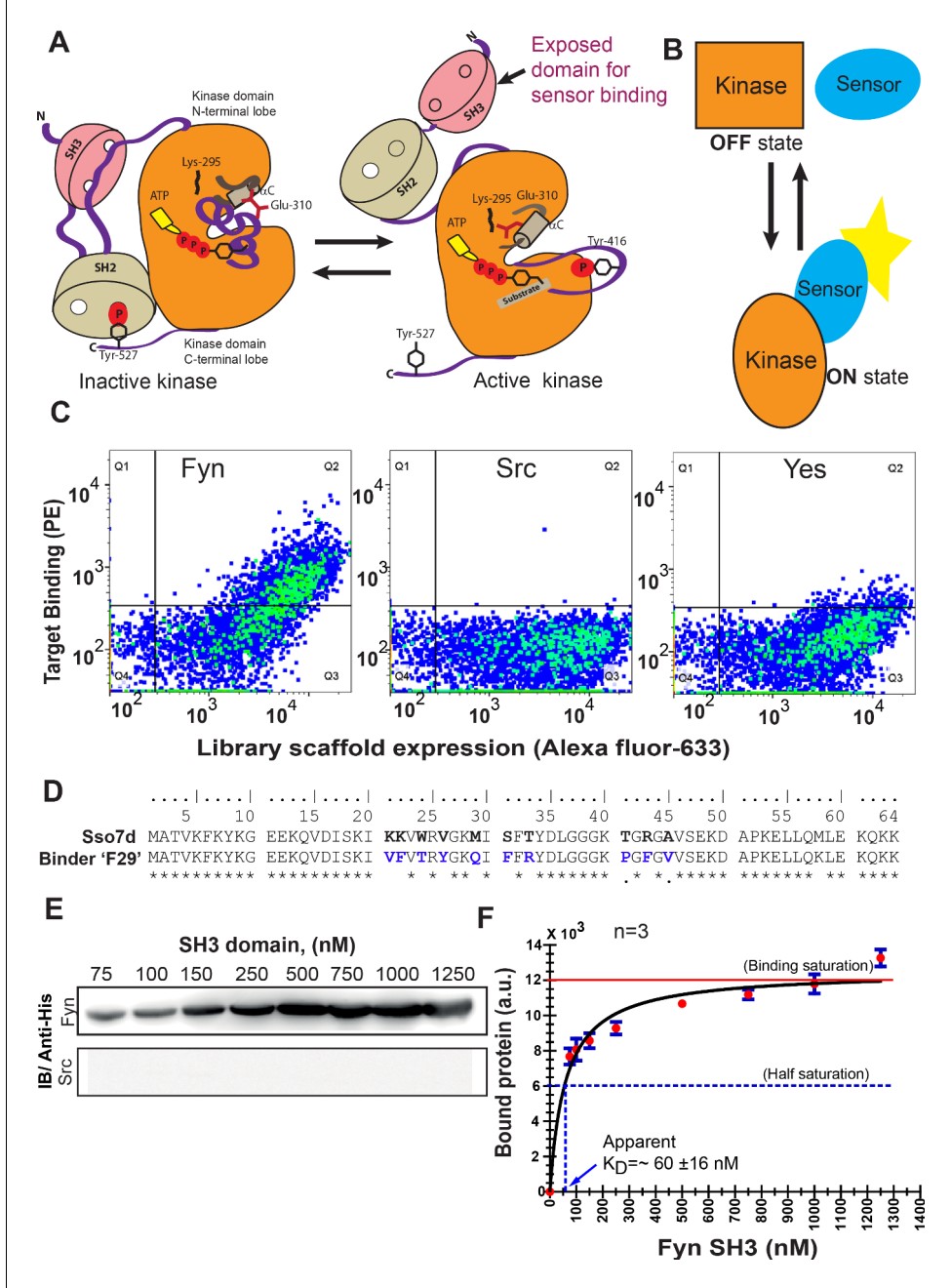

**Figure 1.** High-throughput screening yields a new protein binder for generating a specific biosensor for active Fyn. (**A**) Rationale for a biosensor targeting active conformation of SFK, Fyn. A cartoon depicts how activation of Fyn kinase leads to a more open conformation and buried regions (including its SH3 domain) becoming more accessible for biosensor binding. Inactive (*left*) SFKs are held in a closed conformation through two key intramolecular interactions (for instance see *Cell*, 2001, 105, 115–12) that are lost in the open state (*right*). (**B**) Schematic diagram representing sensing of active and open state of kinase by an active state sensor. (**C**) High-throughput combinatorial library screening yields a protein binder specific for Fyn SH3 domain. Analysis of the yeast clone 'F29' using fluorescence activated cell sorting (FACS) shows F29 preferentially binds Fyn while showing little or no binding to SH3 domains of ubiquitously expressed SFKs, c-Src and Yes (1 μM each). Here, fluorescence signal due to biotin-labeled target SH3 (Streptavidin-PE) is plotted against signal due to yeast cells expressing the c-myc-tagged Sso7d scaffold (Alexa 633 labeled anti-c-myc). Detection of fluorescence signal due to Fyn-SH3 along with anti-c-myc fluorescence in a single yeast cell, indicates expression of full-length F29 fusion-protein and binding to labeled Fyn-SH3 target (also see *Figure 1—figure supplement 2*). (**D**) Comparison of amino acid sequences of new binder F29 with original Sso7d scaffold. 10 Sso7d residues randomized for library generation are

*Figure 1 continued on next page*

*Figure 1 continued*

shown in bold while corresponding new residues in binder F29 are shown in blue. (E) Binder F29 is able to recognize Fyn SH3 outside the context of yeast surface with recombinantly expressed and purified F29 showing saturable binding to Fyn SH3. Immunoblot showing pull-down of His-tagged Fyn or Src SH3 domains (75 nM to 1.25 μM) using GST-F29 fusion protein immobilized as bait on glutathione sepharose beads. His-tagged SH3 protein detected via anti-6X-His antibody (also see *Figure 1—figure supplement 3*). (F) Amount of bound Fyn SH3 plotted against total input Fyn SH3. The data from three independent binding experiments (n = 3) was plotted and fitted assuming 1:1 binding using a GraphPad Prism module. The apparent $K_D$ values were calculated using *Equation 1* (refer Materials and methods). Values are mean ± s.e.m.

The online version of this article includes the following source data and figure supplement(s) for figure 1:

**Source data 1.** Analysis of F29 binding to Fyn-SH3.
**Figure supplement 1.** Combinatorial library screening for a Fyn specific binder using yeast surface display of Sso7d variants.
**Figure supplement 2.** Screening yields new binder F29 that is highly specific for Fyn SH3.
**Figure supplement 3.** Newly identified, isolated protein binder F29 binds Fyn SH3 domain.

---

*1A,B*). Stringent negative selection steps were included to eliminate binders that exhibited cross-reactivity to the SH3 domains of seven other SFKs, specifically c-Src, c-Yes, Fgr, Blk, Lck, Lyn, and Hck. The pool of yeast cells obtained after magnetic selection were further subjected to multiple rounds of FACS to enrich Fyn SH3-binding yeast cells (*Figure 1—figure supplement 1C*). Single clones were then picked and assessed for specific binding to Fyn-SH3. One such clone (referred to as F29 hereafter) showed specific binding to Fyn-SH3 domain, with little or no cross-reactivity with SH3 domains from the other highly homologous SFKs mentioned above (*Figure 1C* and *Figure 1— figure supplement 2*). Sequence of F29 and its comparison to wild-type Sso7d is shown in *Figure 1D*.

We further examined if F29 retained binding to Fyn-SH3 when removed from the context of the yeast cell surface. F29 fused with glutathione-*S*-transferase(GST-F29) was recombinantly expressed and purified, and immobilized on glutathione sepharose beads. Immobilized GST-F29 could pull down 6xHistidine (6xHis) -tagged Fyn-SH3, as detected by immunoblotting using an anti-6xHis anti-body. In contrast, no signal was seen when the Fyn SH3 pulldown experiment was done with GST-saturated beads, or when the SH3 domain of Src was pulled down using GST-F29 beads (*Figure 1— figure supplement 3*, *Figure 1E*). Taken together these results confirm that F29 binds specifically to the Fyn-SH3. We further used pulldown experiments to estimate the binding affinity ($K_D$) of the interaction between F29 and Fyn-SH3. Briefly, immobilized GST-F29 was incubated with varying concentrations of Fyn-SH3, and the amount of bead-bound Fyn-SH3 was quantified. Upon fitting the data to a monovalent binding isotherm, the apparent $K_D$ of the F29-Fyn-SH3 interaction was estimated as 60 ± 16 nM (*Figure 1F*, *Figure 1—source data 1*).

## Molecular basis of specific Fyn recognition by F29

F29 specifically binds Fyn SH3 while showing little or no cross-reactivity to SH3 domains of other highly homologous SFKs. To investigate the molecular basis of the specificity of F29 binding to Fyn-SH3, we used NMR spectroscopy to determine the structure of F29 in complex with Fyn-SH3. For this, we first analyzed the structure of isolated F29 (*Figure 2*). The $^{15}$N-edited Heteronuclear Single Quantum Correlation (HSQC) spectrum of purified F29 (isolated from *E. coli* grown in $^{13}$C, $^{15}$N-labeled media) showed well-dispersed peaks, indicating proper folding of the protein despite extensive mutagenesis of the Sso7d scaffold (*Figure 2—figure supplement 1A*). Complete chemical shift assignments of the backbone and side chain atoms using a series of 2D and 3D NMR experiments were then used to calculate a structure by the CS-ROSSETTA algorithm (*van der Schot et al., 2013*). The structure of F29 superimposed well with Sso7d (*Figure 2—figure supplement 2A*) indicating a conservation of the Sso7d fold in the binder.

To obtain the structure of Fyn-SH3:F29 complex, $^{15}$N-edited HSQC spectra of $^{15}$N-labeled free Fyn SH3 domain (residues 87 to 139 of h-Fyn) as well as in complex with F29 (unlabeled) were acquired. The differences between chemical shifts in the free and bound Fyn-SH3 provided the Chemical Shift Perturbations (CSPs) in Fyn-SH3 due to F29 binding (*Figure 2—figure supplement 1B,D*). High CSPs point to specific Fyn SH3 residues likely present at the binding interface in the

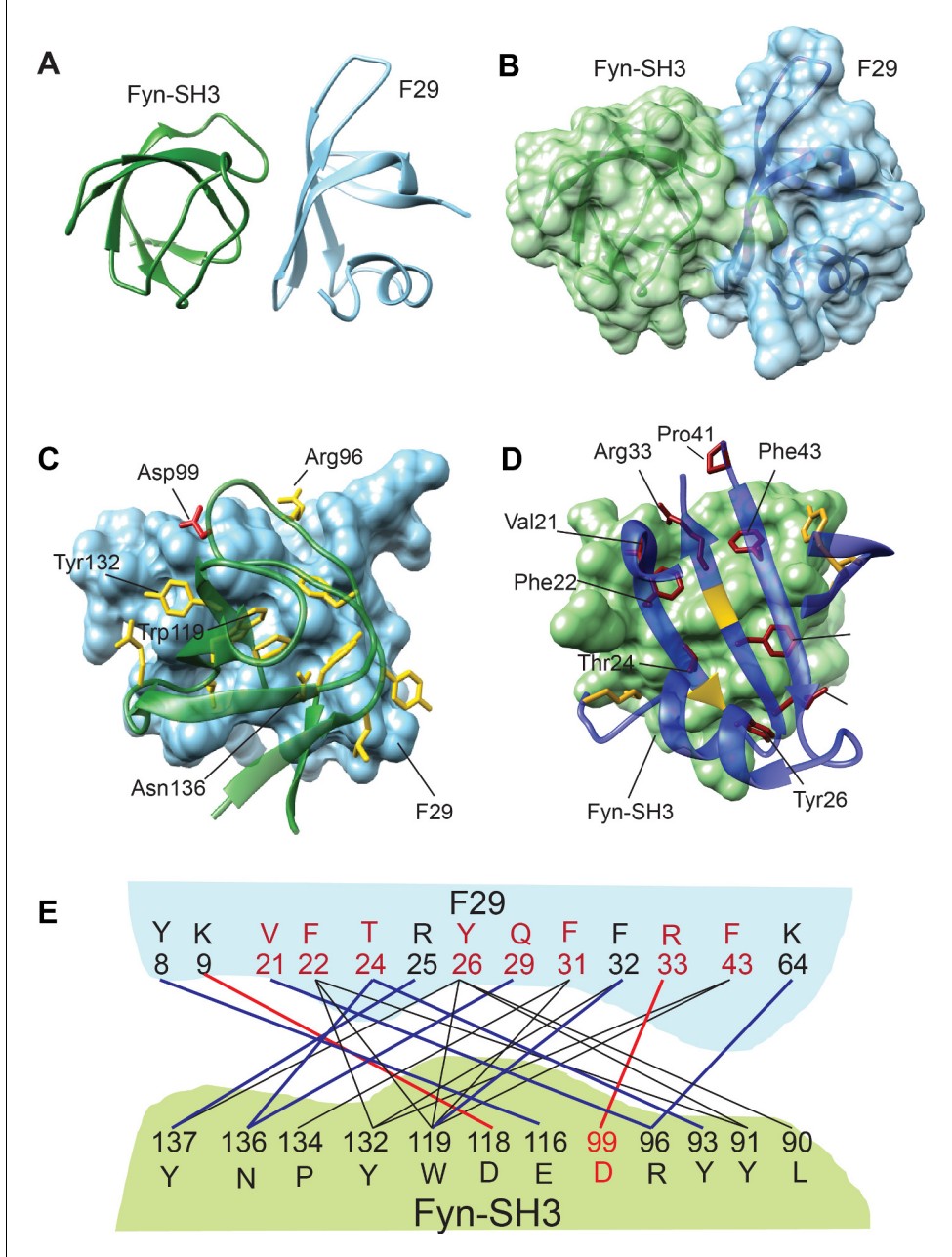

**Figure 2.** Structure and binding interface of Fyn-SH3:F29 complex by NMR. (**A**) The lowest energy structure from Haddock is provided, where Fyn-SH3 is colored green and F29 is colored in light blue. (**B**) The surface representation of **A**, where the surface of Fyn-SH3 and F29 are colored in green and light blue, respectively. (**C**) *Ribbon diagram of contact interface*: A view of the contact interface where the F29 surface is represented in light blue and Fyn-SH3 is shown in green ribbon. The interacting side chains of Fyn-SH3 are shown in yellow. A key interacting residue, Asp99, that forms a salt-bridge with F29 and has been subjected to mutagenesis is shown in red. (**D**) *Complementary view of interface*: Complementary view of the interface with Fyn-SH3 in green and F29 in blue (ribbon diagram) is shown. The interacting side chains of F29 residues are shown and identified here. They are colored in dark red if randomized and colored in gold otherwise. The non-randomized residues Arg25 and Phe32 interact with Fyn-SH3 through the backbone atoms and hence, their ribbons are colored in gold. Non-interacting regions of F29 far away from the interface are hidden for clarity. (**E**) Schematic showing Fyn-SH3:F29 amino acids involved in interfacial contacts. The hydrophobic and aromatic-aromatic contacts between specific residues are shown in black lines, the hydrogen bonds are shown in blue lines and salt-bridges are shown in red

*Figure 2 continued on next page*

*Figure 2 continued*

lines. Residues of F29 are colored in red if randomized and colored in black otherwise. Asp99 in Fyn-SH3 is colored in red (Also see *Figure 2—figure supplement 3B*).

The online version of this article includes the following source data and figure supplement(s) for figure 2:

**Figure supplement 1.** NMR analysis yields structure of the Fyn-SH3 complex with newly isolated binder.

**Figure supplement 2.** Structures of F29 bound to target Fyn SH3 domain.

**Figure supplement 3.** F29 specifically binds Fyn SH3 independent of the yeast cell surface and binding specificity can be switched through targeted mutagenesis.

**Figure supplement 3—source data 1.** Analysis of the extent of Fyn SH3 pulldown by F29 and its mutants.

complex with F29 (*Figure 2—figure supplement 1D*). Similarly, the CSPs in $^{15}$N-labeled F29 upon binding Fyn-SH3 were also obtained (*Figure 2—figure supplement 1A,C*); these values help identify the F29 binding interface in the complex. A 3D $^{15}$N-edited Nuclear Overhauser Effect Spectroscopy (NOESY) experiment, performed on a complex of $^{15}$N, $^{2}$H-labeled Fyn-SH3 and unlabeled F29, led to the unambiguous assignment of ~7 intermolecular NOEs from the spectra. These NOEs helped identify through-space interfacial interactions between residues. Using the structure of Fyn-SH3 (pdb: 3UA6), the structure of F29 obtained above, and the CSPs as ambiguous restraints, and NOEs as unambiguous restraints, the structure of Fyn-SH3:F29 complex was solved by Haddock (*de Vries et al., 2010*; *van Zundert et al., 2016*).

After analysis, the most populated cluster had 200 structures (*Supplementary file 1A*) and the lowest energy structure and its surface representation are shown in *Figure 2—figure supplement 2B* and *Figure 2A,B*. The contact residues at the interface of the complex are shown in *Figure 2C, D*. At the binding interface several non-covalent interactions including hydrophobic-aromatic, hydrogen bonds and salt bridges can be observed (*Supplementary file 1B*). Eight of the ten randomized residues as well as five originals (SSo7d scaffold) amino acids of F29 show interactions with twelve amino acids of Fyn SH3 (*Figure 2E*). Analysis of the binding interface reveals that residues Phe22, Thr24, Tyr26, and Phe31, 32, 43 of F29 show multiple interactions with the Fyn SH3 domain, with SH3 residues Tyr137, Asn136, Tyr132, Trp119, Arg96 and Tyr91 featuring prominently in these interactions. Interestingly, Lys9 (an original Sso7d residue) and Arg33 (randomized residue) on the F29 binder showed exclusive salt bridge interactions with Asp118 and Asp99 of Fyn SH3 domain, respectively (*Figure 2E*).

SH3 domains naturally interact with proteins with poly-proline motifs (*Li, 2005*). To gain insight into how F29 binds and specifically recognizes Fyn SH3, we compared the F29-Fyn SH3 complex with structures of FYN-SH3 bound to poly-Proline peptides (pdb: 4EIK and pdb: 3UA7) (*Figure 2—figure supplement 1E,F*). The buried surface area of F29:Fyn SH3 complex is much larger (800 Å$^2$, ΔG = −7 kcal/mol) compared to that of the poly-Pro:Fyn SH3 complex (pdb: 3UA7, 330 Å$^2$, ΔG = −5 kcal/mol). The number of contacts is also considerably larger in the F29 (84 contacts) versus the poly-Proline (40 contacts). Moreover, the orientation of the poly-Prolines at the interface differs from the orientation of the F29 interfacial beta-sheet by ~30°. Hence, although the F29 interface overlaps with the poly-Proline interface, it is significantly distinct in terms of contacts, surface area and orientation. Residues of Fyn-SH3 that are involved in binding F29 as well as Poly-Pro-1/2 polypeptides are shown in *Figure 2—figure supplement 1G*.

We then used site-directed mutagenesis to examine contributions of key interfacial residues to F29-Fyn SH3 binding. A salt bridge between the residue R33 of F29 and D99 of Fyn SH3 appears prominent. Indeed, the introduction of an R33A mutation in F29 resulted in reduced binding to Fyn-SH3 domain, as evidenced by pull-down analysis (*Figure 2—figure supplement 3A*, *Figure 2—figure supplement 3—source data 1*). These results confirm that R33 in F29 and D99 in Fyn-SH3 contribute substantially to the F29-Fyn-SH3 binding interaction. Interestingly, a Pro41Ala mutant of F29 also showed reduced binding to Fyn-SH3 (*Figure 2—figure supplement 3A*). Although the NMR analyses do not show Pro41 to form direct interfacial contacts with SH3, the reduced binding of this mutant may be attributed to perturbation of local structure. Proline41 forms a critical hinge preceding β3 strand (*Figure 2D*); mutation of the proline likely disrupts the structure the β-sheet that contains multiple key binding contacts with Fyn-SH3.

These structural analyses also offer insight into the molecular basis for the specificity of F29 binding to Fyn-SH3 over highly homologous SH3 domains of other SFKs. While the binding interface in the Fyn-SH3-F29 complex reveals multiple interactions (*Figure 2D,E*), the residue D99 in Fyn-SH3 that forms a salt bridge with R33 of F29 is unique to Fyn among the ubiquitously expressed SFKs as well as Lyn and Fgr (*Figure 2—figure supplement 3B*). Therefore, we hypothesized that D99 in Fyn-SH3 may contribute to the binding specificity of F29 for Fyn-SH3 vis-à-vis to at least some of the SFK SH3 domains. To test this hypothesis, we generated a mutant version of the SH3 domain from Src (Src-SH3) wherein the native threonine was replaced with aspartic acid (T99D). Strikingly, T99D Src-SH3, but not wild-type Src-SH3, showed detectable binding to F29 in pulldown assays (*Figure 2—figure supplement 3C*). These results suggest that D99 in Fyn-SH3 contributes to the binding specificity of F29 for Fyn over other SFKs, notably Src. Nevertheless, specificity of F29 for Fyn-SH3 over Yes-SH3 (containing E99) (*Figure 1C,E*) and Lyn/Fgr (containing D99, *Figure 1—figure supplement 2*), suggests that binding specificity results from a unique combination of multiple interactions at the binding interface that may require further investigation.

## F29 enables construction of *FynSensor*, a FRET biosensor for active Fyn

F29 efficiently and specifically binds Fyn SH3 in vitro. We then examined if F29 is able to recognize full-length, cellular Fyn and if it preferentially binds the active form of Fyn. To test this, either WT Fyn kinase or the CA mutant (*Takeuchi et al., 1993*) (active and open conformation) was exogenously expressed in human embryonic kidney (HEK-293T) cells and the cell lysate treated with immobilized GST-F29. Fyn pulldown and immunoblotting showed that the CA mutant was more efficiently pulled down by F29 as compared to the WT Fyn kinase (*Figure 3A*, *Figure 3—source data 1*). On the contrary, GST-alone control beads showed no significant pulldown of either WT or CA Fyn. These data indicate that F29 preferentially binds active Fyn; the SH3 domain targeted by F29 is more accessible in the CA mutant compared to the WT Fyn kinase. Further, similar GST-pulldown and immunoblot analysis showed that immobilized F29 (GST-F29), but not GST-alone control, was able to bind and pull-down endogenous or native Fyn from cell lysates (*Figure 3B*). These results confirm efficient binding of F29 to cellular Fyn.

Taken together, our results so far show F29 to be a promising candidate for generating an intracellular biosensor for active Fyn. Accordingly, we conceptualized a biosensor design based on FRET wherein the binding of F29 to active Fyn would result in an increase in FRET between suitably placed donor and acceptor fluorophores (*Figure 3C*). To construct a genetically encoded biosensor, Fyn was labeled with a FRET donor fluorescent protein (mCerulean) while F29 was tagged with a FRET acceptor protein (mVenus). While F29 already shows high specificity to Fyn, our design further ensures that FRET signal can arise only from labeled F29 binding to labeled active Fyn. For generating the fluorescently labeled Fyn, we chose to introduce mCerulean (FRET donor) between the unique (UD) and the SH3 domains of full-length Fyn kinase (*Paster et al., 2009*; *Stirnweiss et al., 2013*) (*Figure 3C*). The choice of this insertion position is designed to increase the probability of efficient FRET when acceptor-labeled F29 binds the exposed SH3 domain in active Fyn. Another critical design consideration was to minimize any perturbation caused due to mCerulean insertion into Fyn. To this end, we selected a flexible poly-(glycine-serine) peptide linker (*Trinh et al., 2004*) to insert mCerulean within Fyn (also see Materials and method and *Figure 3—figure supplement 1* for details on molecular engineering). We then examined the suitability of the designed biosensor constructs for cellular imaging. Immunoblotting analysis on cell lysates showed robust intracellular expression of full-length mCerulean-Fyn fusion proteins (hereby referred to as 'mCer-Fyn'), confirming that the fusion proteins remain intact and resistant to proteolysis in cells (*Figure 3—figure supplement 2A–I*). Further, mCer-Fyn fusions expressed in cells showed expected fluorescence; with spectra from cells closely resembling unmodified mCerulean (*Figure 3—figure supplement 2C*). Additionally, the mVenus-F29 fusion biosensor protein also expressed well, was resistant to proteolysis (*Figure 3—figure supplement 3A*) and showed the expected fluorescent signatures resembling unmodified mVenus (*Figure 3—figure supplement 3B*). Importantly, we then tested if labeled Fyn kinase retains activity and is regulated appropriately. Src family kinases possess autocatalytic activity; a tyrosine residue (Y420 in h-Fyn) in the activation loop is phosphorylated when the protein gets activated (*Cooper and MacAuley, 1988*; *Barker et al., 1995*). Significantly, immunoblot analysis showed that mCer-Fyn retains autocatalytic activity similar to the wild-type unmodified protein in cells. Further quantification of Fyn Y-420 autophosphorylation levels, show no significant differences

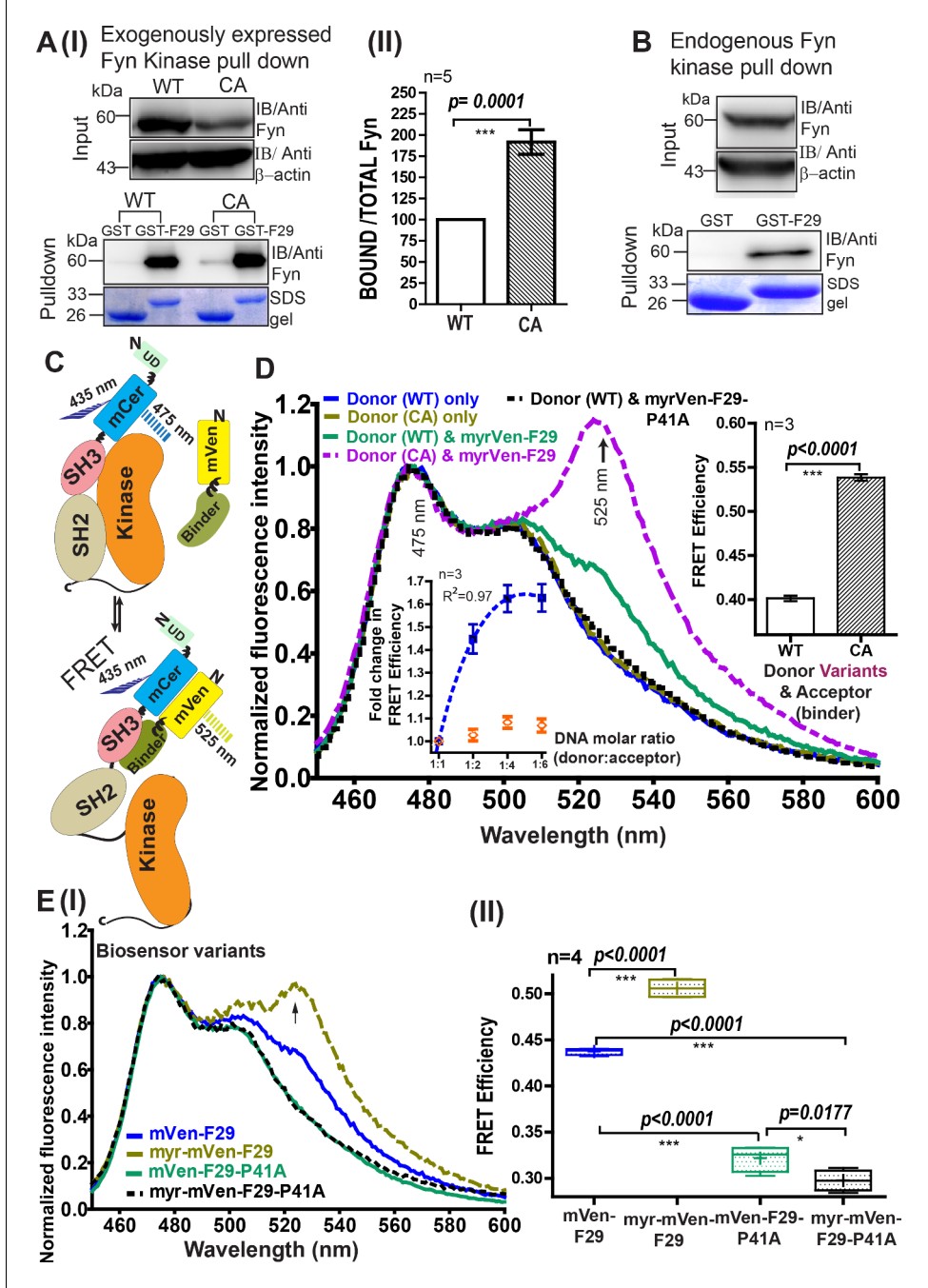

**Figure 3.** Developing a FRET biosensor for sensing active Fyn in cells. (**A**) F29 preferentially binds active Fyn kinase. (I) Immunoblots showing GST-F29 pulldowns of cellular Fyn after transfecting cells with either Fyn WT (wild type) or Fyn CA (constitutively active kinase). (II) Extent of pulldown with either CA or WT Fyn. The band intensity of pulled-down Fyn kinase from five such experiments (n=5) was quantified and normalized to expression level. Binder F29 is able to pull-down CA mutant of Fyn to a greater extent than WT. Values are mean ± s.d from five independent experiments (n=5). Data was compared using Student's unpaired one-tailed t-test on GraphPad Prism. (**B**) F29 binds full-length, endogenous cellular Fyn kinase. Immunoblots, using anti-Fyn antibody, show pulldown of full-length endogenous Fyn from cell lysates by GST-F29 immobilized on glutathione beads (n = 3). GST-coated beads are shown as controls. Coomassie stained gels of the bait proteins, either GST-F29 or GST used for pulldown are also shown. (**C**) Schematic diagram and model for an intermolecular FRET biosensor for visualizing active Fyn in cells. The donor fluorophore (mCerulean) is inserted in between Fyn kinase unique (UD) and SH3 domains using flexible linker sequences to minimally disrupt kinase function (also *see **Figure 3—figure***

*Figure 3 continued on next page*

*Figure 3 continued*

*supplements 2* and *5*). Newly developed Fyn active-state binder, F29, is fused with acceptor fluorophore (mVenus). In the cell, F29 binder (FRET-acceptor) would bind the exposed SH3 domain of active Fyn (FRET donor), resulting in an intermolecular FRET between donor and acceptor fluorophores (*bottom panel*). In contrast, F29 would be expected to show significantly lower binding and reduced or no FRET when Fyn is inactive (*upper panel*). (D) Sensing active Fyn in cells through binding-dependent FRET. Fluorescence spectral measurements from HEK-293T cells co-transfected with FRET-donor (WT or CA Fyn-mCerulean) alone OR along with a FRET-acceptor (either Fyn binder mVenus-F29 or the non-binding mutant, mVenus-F29P41A). Specifically, the normalized emission spectra of transiently transfected cells when excited at 435 nm (corresponding to donor mCerulean excitation) are shown. Spectra has been normalized relative to emission intensity at 475 nm (corresponding to mCerulean emission). Overall, the spectra show that FRET ('sensitized' acceptor emission at 525 nm) is greater for CA Fyn as compared to WT Fyn. No FRET signal is observed (sensitized 525 nm emission) with donor alone or non-binding mutant F29-P41A. *Left inset figure* shows that binding and increase in FRET efficiency is saturable. Using raw fluorescent intensity values the FRET efficiency was calculated as per the equation $F_a/(F_a+F_d)$, where $F_a$ is acceptor and $F_d$ is donor raw fluorescence intensity, respectively. Increasing the amount of labeled binder (mVenus-F29/acceptor) in cells while maintaining constant Fyn (donor) levels causes an increase in FRET till saturation is reached (blue closed squares), indicating binding dependent FRET. From three independent experiments (n=3) , a nonlinear best fit was plotted ($R^2$ = 0.97) using GraphPad Prism (blue, dotted line). In contrast, acceptor-labeled non-binder control (F29 P41A mutant) shows little or no FRET (orange open symbol) showing FRET response is binding dependent and specific. *Right inset figure* shows a bar plot of FRET efficiency comparing WT and CA Fyn for a 1:2 donor:acceptor molar ratio. From three such independent experiments (n=3) , data was compared using Student's unpaired one-tailed t-test on GraphPad Prism. Values are mean ± s.e.m. (E) Membrane tagging of F29 increases biosensor sensitivity and FRET efficiency (calculated as above). (I) The emission spectra of cells co-transfected with Fyn (WT) and, either mVenus-F29 or myr-mVenus-F29 at a 1:4 donor: acceptor molar ratio on 435 nm excitation are shown. Spectra for corresponding non-binding controls (mVenus-F29P41A with and without myristoyl tags) are also shown. (II) Box-and-Whisker plot showing observed FRET efficiency (calculated as above) in cells when myristoyl-tag is added to the FRET-acceptor (mVenus-F29) or its non-binding control mutant P41A. Here, Whiskers represents the lowest and highest observed values of FRET efficiency, while Box spans the interquartile region with the ends demarcating the lower and upper quartile values. The mid-line marks the median. Plus symbol (+) represents the mean FRET efficiency. Data from four such independent experiments (n=4) were analyzed using Student's unpaired one-tailed t-test on GraphPad Prism (respective p values are indicated on panel).

The online version of this article includes the following source data and figure supplement(s) for figure 3:

**Source data 1.** Analysis of F29 pulldown of Fyn and its constitutively active CA mutant.

**Source data 2.** Analysis of *FynSensor* FRET efficiency with changing the cellular levels of acceptor F29 and non-binding control.

**Source data 3.** Analysis of *FynSensor* FRET response towards WT Fyn and constitutively active (CA) Fyn.

**Source data 4.** Analysis of biosensor FRET efficiency observed with various labeled F29 and non-binding constructs.

**Figure supplement 1.** Step-wise representation of PCR based cloning methods for biosensor generation.

**Figure supplement 2.** Labeled Fyn biosensor, mCer-Fyn is active, fluorescent, resistant to proteolysis and shows activity (autophosphorylation) levels identical to unlabeled, wild-type kinase.

**Figure supplement 2—source data 1.** Analysis of autocatalytic activity of labeled and unlabeled Fyn.

**Figure supplement 3.** Labeled F29 expressed in cells is resistant to proteolysis, shows expected fluorescence in cells.

**Figure supplement 4.** Knockdown of endogenous Fyn kinase expression using RNA interference in HEK293T cells.

**Figure supplement 4—source data 1.** Quantification of protein levels of Fyn, c-Src and c-Yes in HEK293T Fyn knockdown cells.

**Figure supplement 5.** *FynSensor* (mCerulean labeled) Fyn is signaling competent and rescues ERK phosphorylation in Fyn knockdown cells similar to unlabeled Fyn.

**Figure supplement 5—source data 1.** Quantification of phosphorylated and total ERK levels in HEK293T cells.

**Figure supplement 5—source data 2.** Quantification of phosphorylated and total ERK levels as a function of Fyn expression in HEK293T cells.

**Figure supplement 6.** Expression of labeled F29 at levels used for *FynSensor* imaging does not cause artefactual activation of kinase.

**Figure supplement 6—source data 1.** Quantification of levels of total and active Fyn in U2OS cells.

**Figure supplement 7.** Cellular expression of *FynSensor* or Fyn binder 'F29' does not perturb cellular morphodynamics.

**Figure supplement 7—source data 1.** Quantification of cell-perimeter and area in *FynSensor* expressing cells.

**Figure supplement 8.** Labeled Fyn localization is unaffected by F29 binder.

*Figure 3 continued on next page*

*Figure 3 continued*

**Figure supplement 9.** Ectopically expressed, labeled Fyn localization is similar to endogenous kinase.

**Figure supplement 9—source data 1.** Quantification of labeled Fyn expression levels in Fyn-KD HEK 293 T cells relative to endogenous kinase levels in control cells.

**Figure supplement 10.** *FynSensor,* labeled Fyn and F29 binder do not perturb downstream signalling at the expression levels used for *FynSensor* imaging in U2OS cells.

**Figure supplement 10—source data 1.** Quantification of phophorylated and total ERK levels in *FynSensor* expressing U2OS cells.

**Figure supplement 11.** *FynSensor,* labeled Fyn and F29 binder do not perturb downstream signaling in C2C12 cells.

**Figure supplement 11—source data 1.** Quantification of phosphorylated and total ERK levels in *FynSensor* expressing C2C12 cells.

---

between mCer-Fyn and unlabeled wt-Fyn (*Figure 3—figure supplement 2A,B*, *Figure 3—figure supplement 2—source data 1*). Importantly, these data show that mCerulean insertion is minimally disruptive to kinase activity and that the engineered kinase appears to be regulated similar to unmodified Fyn.

To further test if our Fyn fusion (mCer-Fyn) is functional and can accurately reflect cellular Fyn dynamics, we examined if mCer-Fyn can modulate downstream signaling and functionally replace endogenous, wild-type Fyn. For this we carefully assayed Fyn effects on downstream signaling, as measured through extracellular signal-related kinase (ERK) phosphorylation and performed Fyn knockdown rescue in cells, comparing both unlabeled as well as Fyn-fusion (mCer-Fyn) constructs. We first established that RNA-i leads to a specific knockdown of cellular Fyn protein (*Figure 3—figure supplement 4A*) but not the closely related Src and Yes kinases in HEK293T cells (*Figure 3—figure supplement 4B,C*, *Figure 3—figure supplement 4—source data 1*). This Fyn knockdown significantly attenuates the levels of ERK phosphorylation in cells (*Figure 3—figure supplement 5A, C*, *Figure 3—figure supplement 5—source data 1*). This is consistent with earlier reports documenting Fyn's role in modulating ERK activity and downstream signaling (*Wary et al., 1998*). Notably, expressing either mCer-Fyn or wt, unlabeled Fyn in 'Fyn-knockdown' cells significantly rescues this reduction in ERK phosphorylation. Further, the extent of increase in this ERK phosphorylation, mediated through ectopic Fyn expression in 'Fyn-knockdown' cells, was observed to be the same for wild-type, unlabeled Fyn as well as mCer-Fyn (*Figure 3—figure supplement 5B–D*, *Figure 3—figure supplement 2—source data 1*). These results unequivocally demonstrate that our labeled Fyn (mCer-Fyn) is regulated appropriately, 'signaling competent' and can functionally replace native, untagged Fyn; thereby making it suitable for faithfully reporting cellular Fyn dynamics.

Biosensor variants were then tested for efficacy in reporting intracellular Fyn activity. For this, we expressed biosensor constructs in HEK-293T cells and recorded fluorescence spectra from live cells (*see* Materials and methods). When cells co-expressing FRET donor (mCer-Fyn) and FRET acceptor (mVenus-F29) were excited at 435 nm (mCerulean/donor excitation), emission peaks were seen at ~475 nm (mCerulean) as well as at ~525 nm corresponding to the peak emission wavelength of the acceptor fluorophore (mVenus) (*Figure 3D*). Appearance of acceptor emission (mVenus,~525 nm) on donor excitation (435 nm) clearly shows that FRET occurs between mCer-Fyn and mVenus-F29. This confirms that mVenus-F29 can recognize intracellular Fyn in live cells and this recognition can be reported through a robust FRET signature. We then tested if this FRET signal is sensitive to the relative stoichiometry of the target (Fyn) and the binder F29 as would be expected with a binding-induced FRET. Indeed, increasing the amount of intracellular mVenus-F29 while maintaining constant expression of mCer-Fyn, led to a concomitant increase in FRET signal, followed by saturation of this FRET increase. Here, the saturation in FRET signal suggests that even increasing concentrations of F29 binder in cells does not cause an inadvertent or artefactual activation of Fyn kinase, especially since F29 binder expression in cells does not lead to any measurable changes in autocatalytic activity of Fyn (Y-420 autophosphorylation) (*Figure 3—figure supplement 6*, *Figure 3—figure supplement 6—source data 1*). Importantly, the FRET signal in cells is specific and dependent on mVenus-F29 binding to mCer-Fyn, since making a single point mutation in F29 abolishes the FRET response. If the non-binding variant of F29 (mVenus-F29-P41A) is co-expressed with mCer-Fyn in cells, little or no FRET is seen, even at higher concentrations of the non-binding control (*Figure 3D*

left inset, *Figure 3—source data 2*). Further, the FRET signal was higher in cells co-expressing mVenus-F29 and constitutively active Fyn, relative to wild-type Fyn (*Figure 3D* right inset, *Figure 3—source data 3*). This shows that F29 binding and biosensor readout (FRET response) is indeed sensitive to the activation status of Fyn (*Figure 3D*).

Fyn kinase has two acylation marks (a myristoyl and a palmitoyl group) that make it preferentially localized to the cell membrane (*van't Hof and Resh, 1999*). Therefore, we reasoned that localization of F29 to the cell membrane would increase its proximity to membrane-bound Fyn, resulting in an increase in FRET signal and consequently greater dynamic range of the biosensor. To test this hypothesis, we added a myristoylation('myr') sequence (MGSSKSKPKDPS) to the F29 binder (*Victor and Cafiso, 1998*). Indeed, addition of a myristoylation signal led to a substantial increase in FRET signal (*Figure 3E*, *Figure 3—source data 4*). To further test the fidelity of this enhanced FRET response observed with myristoylated-F29, we tested the non-binding variant (P41A) of the myr-mVenus-F29 binder. Notably, myristoylated non-binding mVenus-F29 P41A showed little or no FRET signal when co-expressed with mCer-Fyn (*Figure 3E*). Taken together, these results clearly show that the FRET signal observed with the F29-based Fyn biosensor is indeed due to specific binding of F29 to the active form of Fyn, and not just due to non-specific membrane localization and incidental proximity. The biosensor constructs –mCer-Fyn and myr-mVenus-F29 (referred to as binder) – are collectively referred to as *FynSensor* hereafter.

Src family kinases, including Fyn are known to strongly regulate cell adhesion as well as the cytoskeleton. Several SFK substrates and interacting partners are reported to be actin binding proteins, regulators of Rho family GTPases and other proteins that help remodel and regulate the actomyosin network (*Ridley et al., 2003*; *Etienne-Manneville and Hall, 2001*; *Huveneers and Danen, 2009*; *Roca-Cusachs et al., 2012*). We therefore reasoned that measuring the cell morphodynamics (*Gulyani et al., 2011*; *Hodgson et al., 2016*) would be a sensitive way to examine if moderate-to-low expression of labeled Fyn and the F29 binder significantly and measurably perturb the cell. First, the expression levels of *FynSensor* constructs labeled Fyn (mCer-Fyn) as well as mVenus-F29 were quantified in U2OS cells to ensure that the binder expression causes no artefactual activation of Fyn kinase (levels of pTyr416/total Fyn; *Figure 3—figure supplement 6*). We then quantitatively examined temporal changes in cell area and perimeter of cells expressing either *FynSensor (mCer-Fyn: myr-mVenus-F29*, DNA ratio 1:2, also see *Figure 3D*) or the binder alone or a control construct (myr-tagged-mVenus) in the adherent osteosarcoma U2OS cells at the expression levels specified. We find no significant change in cellular morphodynamics as a result of *FynSensor* expression the binder alone as compared to controls cells (*Figure 3—figure supplement 7*, *Figure 3—figure supplement 7—source data 1*). We further confirmed that the expression of Fyn-binder does not perturb the overall intracellular localization of labeled Fyn (*Figure 3—figure supplements 8* and *9*, *Figure 3—figure supplement 9—source data 1*). Immunofluorescence data also shows that labelled Fyn (mCer-Fyn) localization is similar to endogenous Fyn (*Figure 3—figure supplement 9A*).

While *FynSensor* and F29 expression had no effect on cell morphodynamics and Fyn localization, we also examined if *FynSensor* expression significantly perturbs downstream signaling. For this, we again used the extent of ERK phosphorylation as a sensitive readout of Fyn signaling. We find that neither expression of *FynSensor* nor binder alone in adherent osteosarcoma U2OS cells under the conditions used has any measurable effect on ERK phosphorylation levels (*Figure 3—figure supplement 10*, *Figure 3—figure supplement 10—source data 1*). Similarly, expression of *FynSensor* constructs (labeled Fyn + binder) in C2C12 mouse myoblasts (*Figure 3—figure supplement 11*, *Figure 3—figure supplement 11—source data 1*) or Fyn-knockdown HEK-293 cells did not significantly change levels of p-ERK (*Figure 3—figure supplement 5*). These data collectively show that *FynSensor* expression under these specified conditions does not significantly perturb downstream signaling. In light of these data showing little or no cellular perturbation, we have used the same *FynSensor* expression conditions for all further biosensor imaging studies.

## *FynSensor* reports localized active Fyn in live cells

Our results show that *FynSensor* produces a FRET signal upon specific binding of F29 to Fyn-SH3, resulting in a sensitive readout of active intracellular Fyn. We further evaluated *FynSensor* for its ability to report spatial patterns of active Fyn in single, living cells. To investigate the spatial FRET response generated by *FynSensor*, we first employed the 'fluorescence recovery after acceptor photobleaching (APB)' method. APB analysis is a robust method of confirming and quantifying FRET in

cells (*Karpova and McNally, 2006*), where recovery of donor fluorescence on APB is seen as a strong indicator of proximity-induced FRET. Energy transfer from the donor to the acceptor molecules results in quenching of donor fluorescence, and this quenching is relieved when proximal acceptor molecules undergo photobleaching. Indeed, when adherent osteosarcoma U2OS cells expressing *FynSensor* were illuminated with increasing doses of 514 nm laser light (wavelength corresponding to acceptor absorption; mVenus photobleaching), we observed a substantial, dose-dependent recovery of donor (mCerulean) fluorescence. This confirms FRET between mCer-Fyn and mVenus-F29. Significantly, APB-induced recovery of donor fluorescence was greater at cell edges (*Figure 4A,B*, *Figure 4—source data 1*), suggesting spatial variation in Fyn activity. Notably, in control cells expressing a mutant form of labeled-F29 (P41A), acceptor photobleaching showed no such dose-dependent recovery of donor fluorescence (*Figure 4C*, *Figure 4—source data 2*). Since the P41A mutant lacks the ability to bind Fyn-SH3, these data show that APB-induced recovery of donor fluorescence and the FRET response are due to F29 binding to target and not due to incidental proximity or optical artifacts. Finally, when cells were treated with a known pharmacological inhibitor of Src family kinase including Fyn, SU6656 (*Blake et al., 2000*), we observed a significant reduction in the amount of donor fluorescence being recovered (*Figure 4—figure supplement 1*, *Figure 4—figure supplement 1—source data 1*). This reduction in FRET (F29 binding to Fyn) on inhibitor treatment is interesting and points to allosteric changes in Fyn conformation on inhibitor binding. While ATP-competitive inhibitors like SU6656 directly bind the catalytic domain, they can also stabilize an inactive conformation of Src family kinases and modulate the accessibility of its key regulatory domains to ligands/binding proteins (*Krishnamurty et al., 2013*). Therefore, our data (reduced *FynSensor* FRET) suggests that indeed Fyn likely adopts a closed conformation in response to inhibitor treatment, thereby reducing SH3 domain accessibility/binding to F29. The sensitivity of *FynSensor* FRET to inhibitor binding in fact suggests that the labeled kinase shows allosteric conformational changes similar to those seen with untagged SFKs, further indicating that the biosensor faithfully reports Fyn conformational dynamics. Taken together, these results further confirm that the *FynSensor* readout is the result of specific binding of F29 binding to Fyn-SH3, and selectively reports the active form of Fyn (*Figure 4B,C* and *Figure 4—figure supplement 1*). The *FynSensor* readout also shows hitherto uncharacterized spatial patterns in intracellular active Fyn in live cells, with greater Fyn activity closer to the cell edge.

## *FynSensor* reveals pulsatile Fyn activity and spatially compartmentalized signal integration

We used *FynSensor* to examine the spatial and temporal dynamics of Fyn activity, especially in light of the important role played by Fyn as a signaling node functioning downstream of distinct receptors classes. We reasoned that ability to specifically image the active conformation of a single SFK would provide new insight into how signal integration may occur. We first examined Fyn activation dynamics in serum-starved cells plated on fibronectin (FN) that show constitutive protrusion retraction cycles. For imaging activity dynamics, we used ratiometric sensitized emission (normalized acceptor emission due to donor excitation) to quantify FRET signals from the sensor (*Gordon et al., 1998*) (Materials and methods). Ratiometric sensitized emission measurements allow facile and rapid imaging of FRET responses in cells with both spatial and temporal resolution.

Sensitized emission (FRET) imaging with *FynSensor* in FN-plated serum-starved U2OS cells revealed spatially localized Fyn activity, with clear intracellular zones showing higher total FRET ($FRET_T$). Such intracellular zones showing enhanced *FynSensor* $FRET_T$ were consistently observed in all cells in the data set (*Figure 5A–I*). Since all cells spontaneously appear to show regions of differential Fyn activity, we employed quantitative image analysis to probe these patterns of Fyn activity. An automated cell quadrant analysis of intracellular $FRET_T$ levels provides clear evidence of differential and compartmentalized Fyn activity. Briefly, every cell imaged was divided into quadrants in an automated manner and FRET response for each quadrant was analyzed over time. These analyses show substantial and significant differences in Fyn activation levels (*FynSensor* $FRET_T$) between intracellular quadrants (designated as 'low' and 'high' FRET quadrants for consistency and comparison) (*Figure 5A–II*, *Figure 5—source data 1*). Importantly, the spatial patterns of Fyn activation revealed through *FynSensor* imaging are robust and are unlikely to be an artifact of imaging, since a single point mutation in the binder substantially abolishes the FRET response seen. When mCer-Fyn is co-expressed with the non-binding P41A mutant of F29, little or no FRET signal is seen in single cells,

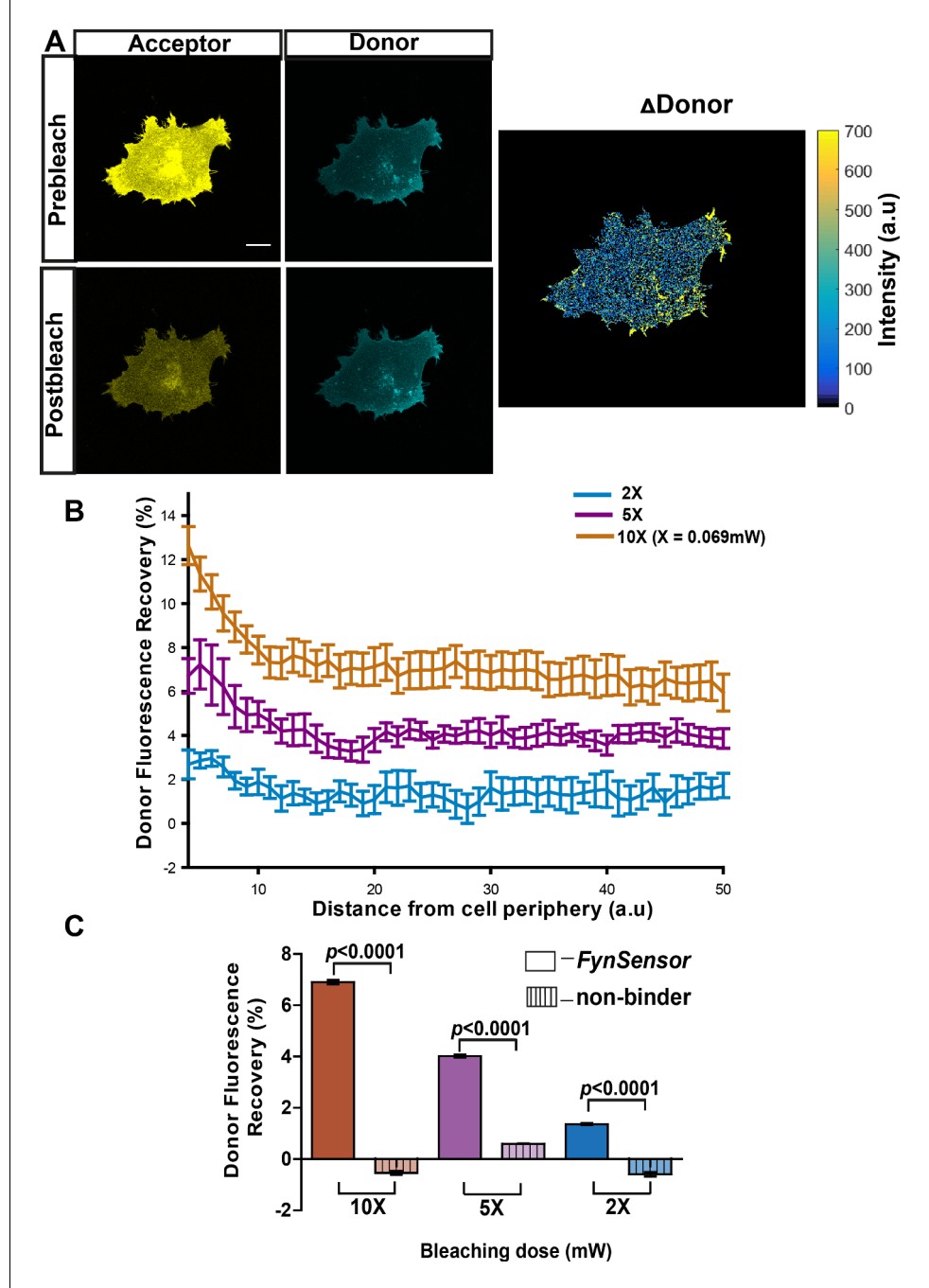

**Figure 4.** *FynSensor* allows direct visualization of spatially localized active Fyn in cells. Acceptor photo-bleaching (APB) experiment for visualizing active Fyn in U2OS cells co-expressing FRET donor (Fyn WT) and acceptor (myr-mVenus-F29) *FynSensor* constructs. (**A**) APB shows cellular FRET and localized Fyn activity. Representative figure shows donor and acceptor fluorescence images, prior to and after laser-based bleaching of acceptor with 514 nm laser pulse. Also shown is the corresponding ΔDonor image (see Materials and methods) (Scale bar = 10 μm). APB experiments reveal areas showing greater recovery of donor fluorescence after APB indicating spatially localized Fyn activity (**B**) APB-induced recovery of donor fluorescence is greater at the cell periphery and sensitive to bleaching light dosage. Percentage recovery of donor fluorescence in *FynSensor*-expressing cells, after acceptor photobleaching at different light dosage (using *Equation 2*, see Materials and methods), plotted as a function of distance from the cell edge. Extent of recovery also shown at different light dosage. Bleaching is carried out for 10 s using the 514 nm laser line at 2X, 5X and 10X doses (X = 0.069 mW). Lines represent mean of n = 14 cells for 10X, n = 9 cells for 5X, n = 7 cells for 2X, respectively. Values are mean ± s.e.m. (**C**) FRET signal measured through

*Figure 4 continued on next page*

*Figure 4 continued*

the APB method is specific (binding dependent) as well as sensitive to APB light doses. Cumulative APB-induced percentage recovery of donor fluorescence observed at the cell periphery in *FynSensor*-expressing cells (as in **B**), plotted as a function of light dosage (solid bars). Also, shown extent of recovery when donor-labeled Fyn is used with acceptor-labeled non-binding P41A mutant of F29 (*dashed bars*). Extent of recovery with non-binding control is minimal and significantly lower than observed for F29 *FynSensor.* Bars represent mean of n = 14 cells for 10X, n = 9 cells for 5X and n = 7 cells for 2X when binder/F29 is used and n = 14 cells for 10X, n = 10 cells for 5X and n = 8 cells for 2X when non-binder/F29P41A is used, respectively. Student's unpaired one-tailed t-test was used to determine the p-value. Graph was made using GraphPad Prism. Values are mean ± s.e.m.

The online version of this article includes the following source data and figure supplement(s) for figure 4:

**Source data 1.** Light dosage-dependence of donor fluorescence recovery in acceptor photo-bleaching experiment with F29 biosensor.

**Source data 2.** Light dosage-dependence of donor fluorescence recovery in acceptor photo-bleaching experiment with F29 biosensor and non-binding mutant.

**Figure supplement 1.** *FynSensor* FRET readout is sensitive to kinase activity of Fyn.

**Figure supplement 1—source data 1.** Quantification of donor fluorescence recovery after acceptor photo-bleaching in cells treated with SFK inhibitor.

---

clearly showing that *FynSensor* FRET is dependent on F29 binding to active Fyn (*Figure 5B*, *Figure 5—source data 2*, *Figure 5—figure supplement 1*, *Figure 5—figure supplement 1—source data 1*, *Figure 5—video 1*). Overall, the spontaneous compartmentalizing of Fyn activity was observed in all cells and is a striking finding revealed through *FynSensor* imaging. (*Figure 5A*).

We demonstrate that the spatially enhanced $FRET_T$ patterns observed are unlikely to be an artifact of an unequal or spatially-constrained intracellular distribution of the F29 binder. *Figure 5—figure supplement 2* (*Figure 5—figure supplement 2—source data 1*) shows that while the *FynSensor* FRET levels are non-uniform and spatio-temporally patterned, the fluorescently labeled F29 (myr-mVenus-F29) is homogeneously distributed in cells. While increased recruitment of Fyn may contribute to the establishment of regions of 'high Fyn activity' in serum-starved, FN-plated cells, our data analysis shows that differential Fyn activity observed cannot solely be ascribed to increased localization of Fyn. *Figure 5—figure supplement 3* (*Figure 5—figure supplement 3—source data 1*) show that intracellular regions showing higher $FRET_T$ also show significantly higher levels of donor-normalized FRET (donor-normalized FRET = $FRET_T$/mCer-Fyn fluorescence), which normalizes the overall activity for protein levels. This is also illustrated (*Figure 5—figure supplement 4*, *Figure 5—figure supplement 4—source data 1*) by examples of regions that have almost similar levels of the kinase (mCer-Fyn localization as visualized in the donor channel in regions marked 1 and 2) but show significantly different levels of kinase activity (FRET index image, regions 1 and 2), reaffirming the point that increases in FRET signals are just not due to increase in kinase localization.

We surmised that spatially enhanced Fyn activity seen in serum-starved cells on fibronectin may be caused by differential integrin signaling. To test if this Fyn activation in serum-starved cells, as visualized through *FynSensor*, is indeed sensitive to integrin signaling, we treated cells with an inhibitor of focal adhesion kinase (FAK), PF-562271 (*Mills et al., 2015*; *Stokes et al., 2011*) and examined *FynSensor* response (*Figure 5C–I*, *Figure 5—video 2*). Focal adhesion kinase (FAK) is a critical transducer of integrin signaling and is known to partner SFKs in mediating cellular responses (*Renshaw et al., 1999*; *Cheng et al., 2014*). Strikingly, when serum-starved, FN-plated U2OS cells were treated with PF-562271, there was an immediate and drastic reduction in the *FynSensor* FRET response (*Figure 5C* II-IV, *Figure 5—source data 3*). This reduction in activity is accompanied by a substantial dampening of the temporal changes seen in levels of active Fyn in cells (see below). These results clearly demonstrate that localized Fyn activity, reported through *FynSensor*, is sensitive to focal adhesion kinase activity; and supports the model that compartmentalized Fyn activity arises through differential integrin signaling in serum-starved adherent cells.

To confirm that these spatial patterns of Fyn activity are not just limited to any one specific cell-type, we performed *FynSensor* imaging in mouse C2C12 myoblasts apart from U2OS osteosarcoma cells. Notably, *FynSensor* again shows Fyn activity to be spatially patterned in the serum-starved C2C12 myoblast cells (*Figure 6A*). Quantitative quadrant analysis confirms differential Fyn activity ($FRET_T$) in distinct intracellular zones (*Figure 6A–II*, *Figure 6—source data 1*). To assess if cells with

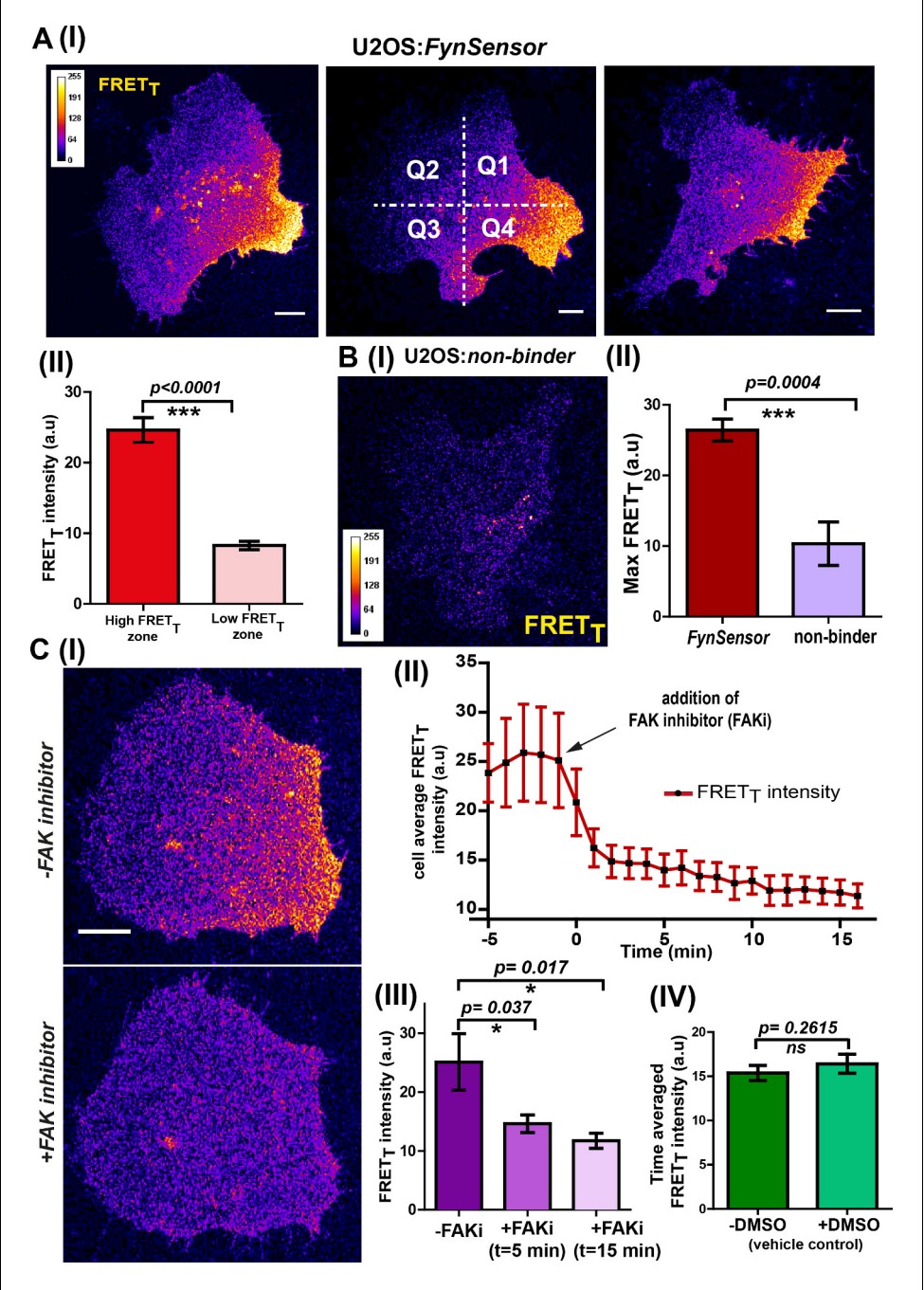

**Figure 5.** *FynSensor* reveals integrin-dependent Fyn activity to be spatially compartmentalized. (**A**) Spontaneous formation of zones enriched in active Fyn is seen in serum starved cells plated on fibronectin (FN). (I) Representative confocal fluorescence micrographs showing sensitized emission (total FRET: $FRET_T$) levels indicative of active Fyn in serum starved-U2OS cells expressing *FynSensor* (Scale bar = 10 µm). For quantitative image analysis of compartmentalized Fyn activity, we have divided the cell into quadrants labelled as Q1-Q4 as shown. (II) Bar graph comparing *FynSensor* FRET levels in distinct cellular zones. The mean of Max-$FRET_T$-HFQ (HFQ: high-FRET quadrant, Q4 in panel I) and Max-$FRET_T$-LFQ (LFQ: low-FRET quadrant, Q2 in I) values (see Materials and methods) are plotted. Values are mean ± s.e.m. Student's paired one-tailed t-test has been used to determine the p-value (n = 37 cells). (**B**) *FynSensor* FRET readout is dependent on F29 binding Fyn, with the non-binding control mutant of F29 (P41A) showing no or significantly reduced $FRET_T$ signal compared to *FynSensor*. (I) Confocal fluorescence micrographs showing sensitized emission (total FRET: $FRET_T$) levels for cells expressing mCer-Fyn and non-binding control binder mVenus-F29 P41A. (II) Comparison of $FRET_T$ levels for *FynSensor* with

*Figure 5 continued on next page*

*Figure 5 continued*

non-binding control. The bar graph shows the average maximum $FRET_T$ obtained (Max-$FRET_T$-HFQ) for non-binder (n = 5 cells) and *FynSensor* (n = 37 cells). Values are mean ± s.e.m. Student's unpaired one tailed-t-test was used to determine the p-value. (C) Fyn activity levels are sensitive to inhibition of focal adhesion kinase (FAK), a known mediator of integrin signaling. (I) Confocal fluorescence micrographs showing *FynSensor* $FRET_T$ levels before and after treatment with 10 µM of FAK inhibitor (PF 562271). Loss of basal Fyn activity is observed in cells when treated with inhibitor but not with DMSO (vehicle control). (II) Time course of Fyn activity in response to FAK inhibition. Average fluorescence intensity profile ($FRET_T$) of *FynSensor* cells (n = 5) over time, before and after addition of FAK inhibitor. (Scale bar = 10 µm). (III) Quantifying *FynSensor* FRET on FAK inhibition. Bar graph shows the average $FRET_T$ intensity of cells before FAK inhibitor treatment (-FAKi), five mins after FAKi treatment and 15 mins after FAKi treatment (n = 5 cells). Values are mean ± s.e.m. Student's paired one tailed-t-test was used to determine the p-value. (IV) FAK-inhibitor induced reduction in *FynSensor* FRET levels is NOT seen with vehicle control. Bar graph shows time-averaged, mean cell $FRET_T$ intensity before and after treatment with the vehicle control (n = 8 cells). Values are mean ± s.e.m. Student's paired one tailed-t-test was used to determine the p-value.

The online version of this article includes the following video, source data, and figure supplement(s) for figure 5:

**Source data 1.** Quantification of *FynSensor* FRET levels in low and high activity zones in U2OS cells.
**Source data 2.** Quantification of *FynSensor* non-binding control FRET levels in low and high activity zones in U2OS cells.
**Source data 3.** Quantification of *FynSensor* FRET levels in cells in treated with FAK inhibitor.
**Figure supplement 1.** *FynSensor* FRET readout is highly sensitive to F29 binding to Fyn kinase in cells. Non-binding mutant of *FynSensor* fails to show significant FRET response.
**Figure supplement 1—source data 1.** Quantification of F29 non-binding mutant FRET levels across cell zones.
**Figure supplement 2.** Localized Fyn activity (spatially localized *FynSensor* FRET) is not an artifact of binder localization.
**Figure supplement 2—source data 1.** Quantification of *FynSensor* and F29 localization levels across cell.
**Figure supplement 3.** Spatio-temporal modulations in Fyn activity are not simply due to changes in local Fyn protein concentration.
**Figure supplement 3—source data 1.** Quantification of donor-normalized FRET levels in low and high activity zones.
**Figure supplement 3—source data 2.** Comparison of *FynSensor* $FRET_T$ levels and Fyn kinase localization at different time points.
**Figure supplement 4.** Localized Fyn activity patterns are distinct from patterns of Fyn localization.
**Figure supplement 4—source data 1.** Quantification of $FRET_T$ levels and kinase localization across selected cellular zones.
**Figure 5—video 1.** FRET response in human osteosarcoma cell-line, U2OS cells expressing non-binder mutant of *FynSensor*.
https://elifesciences.org/articles/50571#fig5video1
**Figure 5—video 2.** Effect of FAK inhibition on Fyn activity in human osteosarcoma cell-line, U2OS cells.
https://elifesciences.org/articles/50571#fig5video2

moderately overexpressed levels of Fyn kinase also show similar spatial-activation patterns, we performed imaging experiments in HEK-293T cells where the endogenous Fyn has been depleted through RNA-i and then complemented with *FynSensor* (*Figure 3—figure supplement 9*, *Figure 6—figure supplement 1*, *Figure 6—figure supplement 1—source data 1*). Biosensor imaging in these HEK cells indeed showed activation patterns very similar to those previously observed in U2OS and C2C12 cells (*Figure 6B*, *Figure 6—source data 2*). The observation of similar, spatially enhanced patterns of Fyn activity in these very distinct cell types shows that *FynSensor* responses are highly robust and is suggestive of conserved spatially modulated signaling mechanisms.

To further confirm that the conserved patterns of Fyn activity are not impacted by changes in *FynSensor* expression levels, we systematically examined *FynSensor* FRET profiles as a function of expression. In all three cell types examined, the spatial patterns remain conserved independent of the precise levels of labeled Fyn (mCer-Fyn) as well as the binder (mVenus-F29) (*Figure 6C*, *Figure 6—source data 3*, *Figure 6—figure supplement 2*, *Figure 6—figure supplement 2—source data 1* and *Figure 6—figure supplement 2—source data 2*).

Another consistent and striking feature revealed by *FynSensor* is the temporal bursts of Fyn activity. $FRET_T$ levels show clear oscillations over time in serum-starved cells plated on FN and imaged

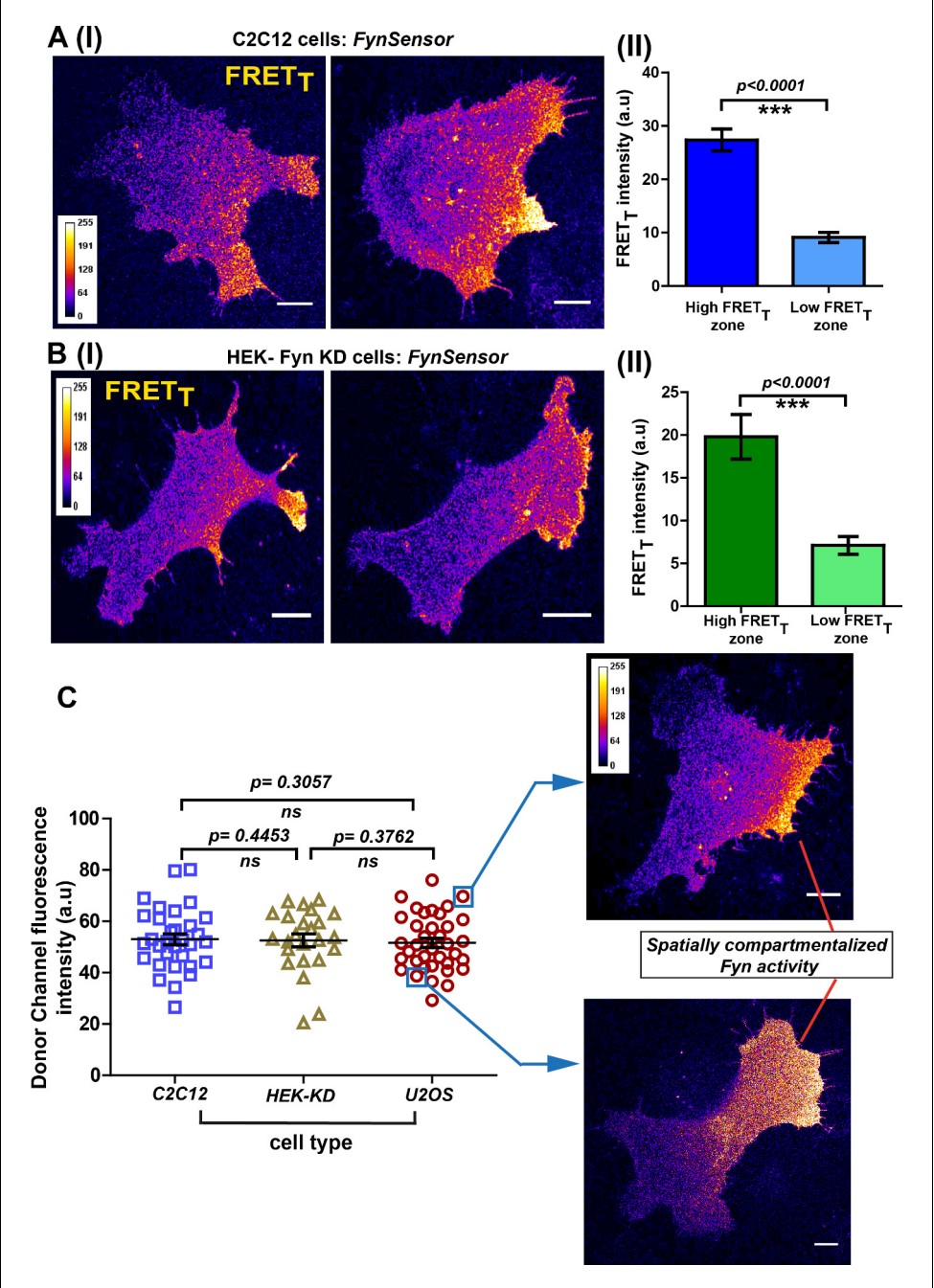

**Figure 6.** Conserved spatial pattern of Fyn activity in distinct cell-types. (**A**) *FynSensor* in mouse myoblast C2C12 cells reveals spontaneously compartmentalized Fyn activity, similar to that observed in U2OS cells. (I) Representative confocal fluorescence micrographs of serum-starved cells showing intracellular zones enriched with active Fyn when plated on FN (Scale bar = 10 μm). Quantitative image analysis, as previously described, yielded zones of high FRET$_T$ and low FRET$_T$. (II) Bar graph comparing *FynSensor* FRET levels in distinct cellular zones. Graph compares the mean of Max-FRET$_T$-HFQ and Max-FRET$_T$-LFQ in serum-starved cells. Values are mean ± s. e.m. Student's paired one-tailed t-test has been used to determine the p-value (n = 32 cells). (**B**) *FynSensor* imaging reveals spatially localized Fyn activity patterns when *FynSensor* is ectopically expressed in HEK293T Fyn-knockdown cells. (I) Confocal micrographs show zones of high and low FRET$_T$, showing spatially localized Fyn activity, similar to that seen with U2OS and C2C12 myoblasts. (II) Bar graphs comparing the mean of Max-FRET$_T$-HFQ and Max-FRET$_T$-LFQ in serum-starved cells confirm differential FRET levels across intracellular zones. Values are mean ± s.e.m. Student's paired one-tailed t-test has been used to determine the p-value (n = 24 cells). (**C**)

*Figure 6 continued on next page*

*Figure 6 continued*

Spatial patterns of locally enhanced Fyn activity revealed by *FynSensor* imaging are unaffected by *FynSensor* expression levels. *FynSensor* imaging reveals locally enhanced, compartmentalized Fyn activity independent of sensor expression levels. To assess the effect of kinase expression on the FRET patterns observed in all the distinct cell types, we quantified the donor (m-Cerulean) fluorescence intensity for all the cells used in our study. Scatter-plot comparing donor fluorescence across different cell types, n = 32 cells for C2C12, 37 for U2OS and 24 for HEK-KD cells. Mean ± s.e.m is denoted. Student's unpaired one-tailed t-test has been used to determine the p-value. Representative cells from the U2OS data set, having either relatively 'high' (above the mean value of 51.61 ± 1.805 fluorescence intensity units, denoted by *teal* box) or 'low' (below the mean value, *teal* box) *FynSensor* expression (donor/m-Cer fluorescence intensities), show very similar spatially compartmentalized Fyn activity pattern, suggesting that the activity patterns observed in different cells are not subject to expression artifacts.

The online version of this article includes the following source data and figure supplement(s) for figure 6:

**Source data 1.** Quantification of *FynSensor* FRET levels in low- and high-activity zones in C2C12 cells.
**Source data 2.** Quantification of *FynSensor* FRET levels in low- and high-activity zones in Fyn-KD HEK293T cells.
**Source data 3.** Quantification of expression of *FynSensor*-Fyn (mCer-Fyn) in different cell-types showing similar activity patterns.
**Figure supplement 1.** Ectopic *FynSensor* expression in Fyn knockdown HEK-293T cells for 'knockdown rescue' biosensor imaging.
**Figure supplement 1—source data 1.** Quantifying the expression of Fyn kinase (labeled or unlabeled) in Fyn KD-HEK293T cells relative to control cells.
**Figure supplement 2.** *FynSensor* FRET spatial patterns and oscillations observed are independent of expression levels of labeled (*FynSensor*) Fyn.
**Figure supplement 2—source data 1.** Quantification of donor:acceptor ratios in different cell types used for *Fyn-Sensor* imaging experiments.
**Figure supplement 2—source data 2.** Analysis of spatio-temporal Fyn activity patterns in U2OS cells expressing different levels of Fyn kinase.

(*Figure 7A*, *Figure 7—source data 1*, *Figure 7—figure supplement 1*, *Figure 7—figure supplement 1—source data 1*, *Figure 7—video 1*). Interestingly, these oscillations in Fyn activity revealed through *FynSensor* imaging are seen in multiple cell types and appears to be a conserved feature of Fyn activation dynamics. $FRET_T$ profiles show temporal oscillations in C2C12 myoblast (*Figure 7B*, *Figure 7—source data 2*, *Figure 7—figure supplement 2*, *Figure 7—figure supplement 2—source data 1*), Fyn-KD HEK-293T (*Figure 7C*, *Figure 7—source data 3*, *Figure 7—figure supplement 3*, *Figure 7—figure supplement 3—source data 1*) as well as U2OS cells (*Figure 7A*) expressing *FynSensor*. We also tested if these Fyn activity pulses arise due to any sudden changes in kinase localization. *Figure 5—figure supplement 3B–I* clearly shows that along with the *FynSensor* $FRET_T$ profiles, even the donor/mCerulean normalized FRET levels show pulsatile behavior. This shows that pulses observed are not due to fluctuations in Fyn concentration levels but arise due to direct and rapid regulation/modulation of Fyn activity (*Figure 5—figure supplement 3B*, *Figure 5—figure supplement 3—source data 2*). Since the pulsatile nature of Fyn activity was found to be conserved across cell types, we specifically examined the duration/frequency of these pulses in U2OS, C2C12 and HEK-Fyn KD cells. *Figure 7D*, *Figure 7—source data 4* shows the mean of the dominant time-period of *FynSensor* FRET pulses in all three cell-types, determined through power-spectrum density (PSD) analysis performed on the $FRET_T$ time-traces. The PSD analysis reveals the mean of dominant time-period to be ~3.5 min in three very distinct cell types, again highlighting a conserved feature of Fyn activity dynamics. There is increasing evidence to suggest that signaling modules, including growth factor responses in cells may show pulsatile behavior (*Coster et al., 2017*; *Warmflash et al., 2012*; *Weber et al., 2010*). Bursts of activity are expected to bear signatures of complex positive and negative (inhibitory) feedback loops that are integral part of growth factor and other signaling modules (*Avraham and Yarden, 2011*; *Sparta et al., 2015*; *Albeck et al., 2013*). Interestingly, temporal oscillations are highly pronounced in regions of spatially-enhanced Fyn activity as can be seen by plotting *FynSensor* $FRET_T$ against time as well as distance from the celledge. The time-distance 3-D plot of Fyn activity shows bursts of activity in a tight intracellular zone proximal to the edge. (*Figure 7E–I,II*, *Figure 7—source data 5*).

Fyn kinase signals downstream of multiple receptor classes, including integrin and receptor tyrosine kinases (RTKs/Growth factor receptors). Based on prior evidence (*DeRita et al., 2017*;

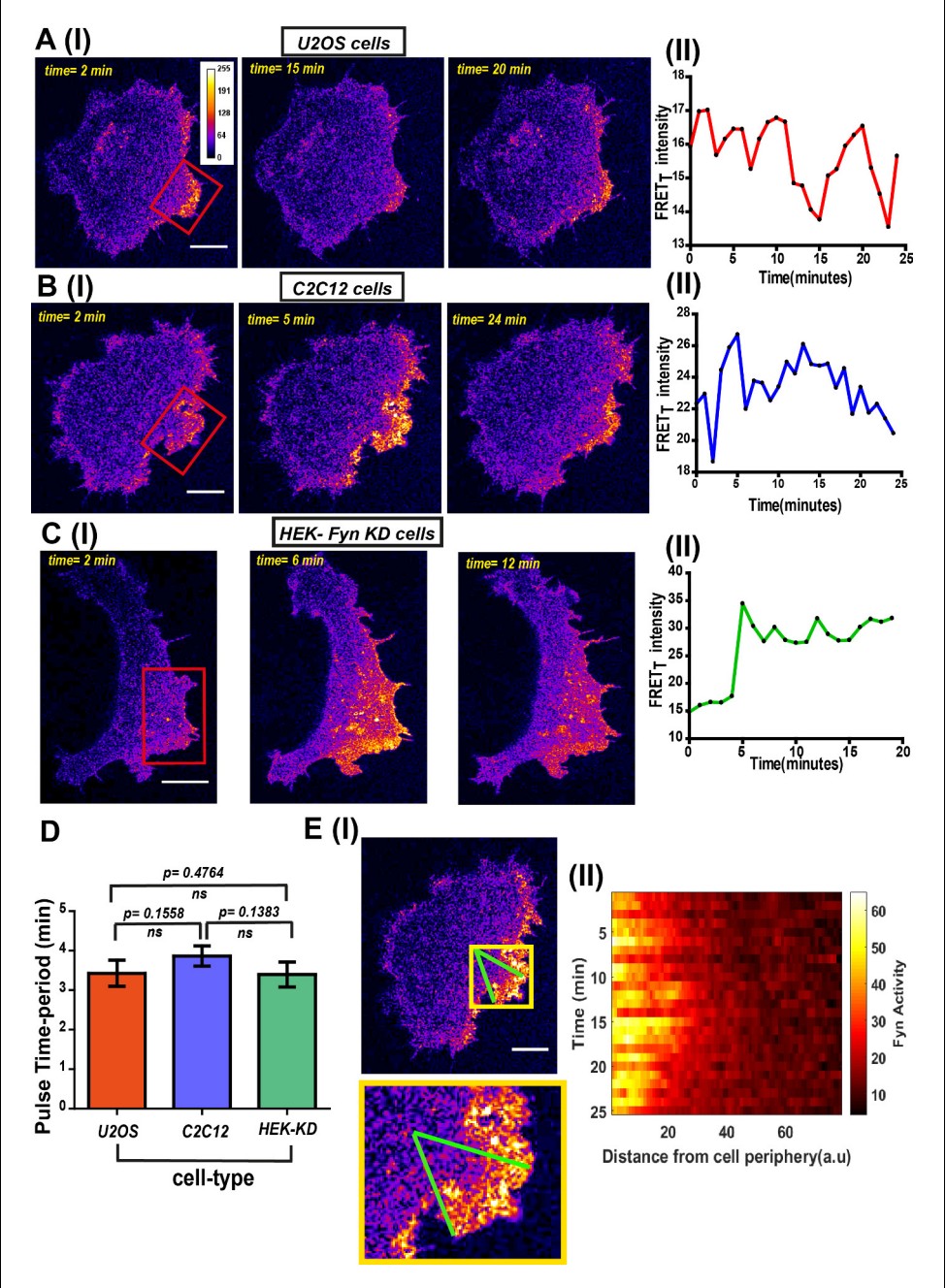

**Figure 7.** *FynSensor* imaging reveals spatially-compartmentalized temporal bursts of kinase activity across distinct cell types. (**A**) A striking feature revealed by *FynSensor* is the pulsatile nature of Fyn activation. (I) Representative confocal micrographs of a U2OS cell at different time points after being plated on FN, as indicated, reveal significant oscillations in the $FRET_T$ intensity (area marked by *red box*) (Scale bar = 10 µm). (II) Time-traces of the $FRET_T$ signal showing temporal pulses of Fyn activity. (**B**) *FynSensor* imaging in C2C12 mouse myoblast cells, a cell-type distinct from the U2OS human osteosarcoma cells, also show an oscillating Fyn activity profile. (I) Confocal micrographs from a C2C12 cell transiently expressing *FynSensor*, at different time-points post FN plating, shows temporal bursts of kinase activity (area marked by *red box*). (II) Time-trace showing the pulsatile temporal pattern in $FRET_T$ signal in the cell shown. (**C**) *FynSensor* imaging reveals pulsatile activity patterns in Fyn knockdown cells that have been 'rescued' with labeled Fyn (mCer-Fyn). Fyn activity pulses are similar to those observed in U2OS and C2C12, showing temporal patterns are robust and conserved. (I) Representative cell confocal micrograph of RNAi-Fyn knockdown HEK-293 (KD) cells expressing *FynSensor* show temporal oscillations in $FRET_T$ levels (area marked by *red box*). (II) Time-trace of $FRET_T$ signal from individual cell revealing pulsatile Fyn

*Figure 7 continued on next page*

*Figure 7 continued*

signal. (**D**) Oscillatory/pulsatile behavior in levels of active Fyn in the different cell-types studied reveal a consistent major time period. Plotted here is the mean of the dominant time-period of *FynSensor* FRET pulses in U2OS C2C12 and HEK-KD cells expressing the sensor, determined through power spectrum density (PSD) analysis on the time-traces of single-cell quadrant averaged total FRET. The PSD analysis reveals the mean of dominant time-period to be ~3.5 min in distinct cell types. N = 37 for U2OS, 32 for C2C12 and 16 for HEK-KD cells. Values are mean ± s.e.m. Student's two-tailed unpaired t-test has been used to determine the p-value. See Materials and methods for details of the analysis. (**E**) Pulsatile Fyn activity is spatially constrained. Intracellular zones of elevated Fyn activity also show clear oscillations over time. (**I**) Confocal fluorescence micrograph of a C2C12 cell shows a zone of high FRET signal (indicated by *yellow* box). For plotting temporal oscillations over distance from edge, an arc (of arbitrary length units) was suspended on the membrane encompassing this zone. A smaller sector was then created on the cell (indicated by *green* lines). (**II**) Plot shows a heat map of $FRET_T$ intensity values in this sector, over time, as a function of distance from the cell-membrane. Data reveals that active kinase to be spatially constrained and oscillating over time. See Materials and methods for details of the analysis.

The online version of this article includes the following video, source data, and figure supplement(s) for figure 7:

**Source data 1.** Analysis of temporal patterns of Fyn activity in U2OS cell.
**Source data 2.** Analysis of temporal patterns of Fyn activity in C2C12 cell.
**Source data 3.** Analysis of temporal patterns of Fyn activity in Fyn-KD HEK 293 T cell.
**Source data 4.** Quantification of Fyn activity pulse time-period across different cell-types.
**Source data 5.** Quantification of Fyn activity relative to distance from cell membrane in C2C12 cell.
**Figure supplement 1.** *FynSensor* reveals spatially localized active Fyn, pulsatile Fyn activity in U2OS osteosarcoma cells.
**Figure supplement 1—source data 1.** Quantification of $FRET_T$ signal over time in multiple U2OS cells.
**Figure supplement 2.** Localized and dynamic kinase activity patterns in C2C12 mouse myoblast cells.
**Figure supplement 2—source data 1.** Quantification of $FRET_T$ signal over time in multiple C2C12 cells.
**Figure supplement 3.** *FynSensor* imaging in HEK-293T cells lacking endogenous Fyn kinase shows activity patterns similar to those seen in U2OS and C2C12 cells.
**Figure supplement 3—source data 1.** Quantification of $FRET_T$ signal over time in multiple Fyn-KD HEK 293 T cells.
**Figure 7—video 1.** Fyn activation in human osteosarcoma cell-line, U2OS cells.
https://elifesciences.org/articles/50571#fig7video1

---

*Edick et al., 2007*; *Samayawardhena et al., 2007*; *Parsons and Parsons, 1997*), we hypothesized that Fyn, functioning as a signaling node may be involved in *dynamically* integrating signals downstream of integrins and RTKs (*Lehembre et al., 2008*; *Kinnunen et al., 1998*; *Arias-Salgado et al., 2005*). We used the *FynSensor* to investigate this putative signal integration. For this, we examined the effect of platelet-derived growth factor (PDGF) on FN-plated, serum-starved U2OS cells and visualized active Fyn using *FynSensor* FRET. As discussed earlier, serum-starved FN-plated cells already showed spatially enhanced and temporally regulated Fyn activity, consistent with constitutive cell polarization and Fyn activation downstream of spatially regulated integrin signaling (*Figure 5A*). However, after stimulation with PDGF, *FynSensor* cells (but not the P41A non-binding control *Figure 8—video 1*) showed a significant increase in Fyn activity as indicated by enhanced FRET signal (*Figure 8A*). Strikingly, despite a global PDGF stimulation, the increase in Fyn activity observed was spatially localized. It appeared that on stimulation, the $FRET_T$ signal, *preferentially* increased in regions that already showed higher FRET in un-stimulated cells (*Figure 8A*; *Figure 8—video 1*, resulting in highly compartmentalized Fyn activity patterns.

To quantitatively examine these activity patterns and signal modulation, we again employed automated image analysis. An automated intracellular quadrant analysis confirms differential *FynSensor* FRET levels across distinct intracellular zones, post-PDGF stimulation (*Figure 8B–I,II*, *Figure 8—source data 1*). These data demonstrate that cells remain 'polarized' with respect to Fyn activity despite a global PDGF stimulation. Notably, quadrant image analysis also confirmed a preferential PDGF-induced enhancement of Fyn activity in pre-activated intracellular zones. *Figure 8B–II* shows that PDGF-induced enhancement in *FynSensor* FRET is greater in intracellular quadrants already showing higher $FRET_T$ pre-stimulation. Overall, this is a powerful demonstration of the ability of *FynSensor* to reveal hitherto unexplored aspects of Fyn signaling dynamics.

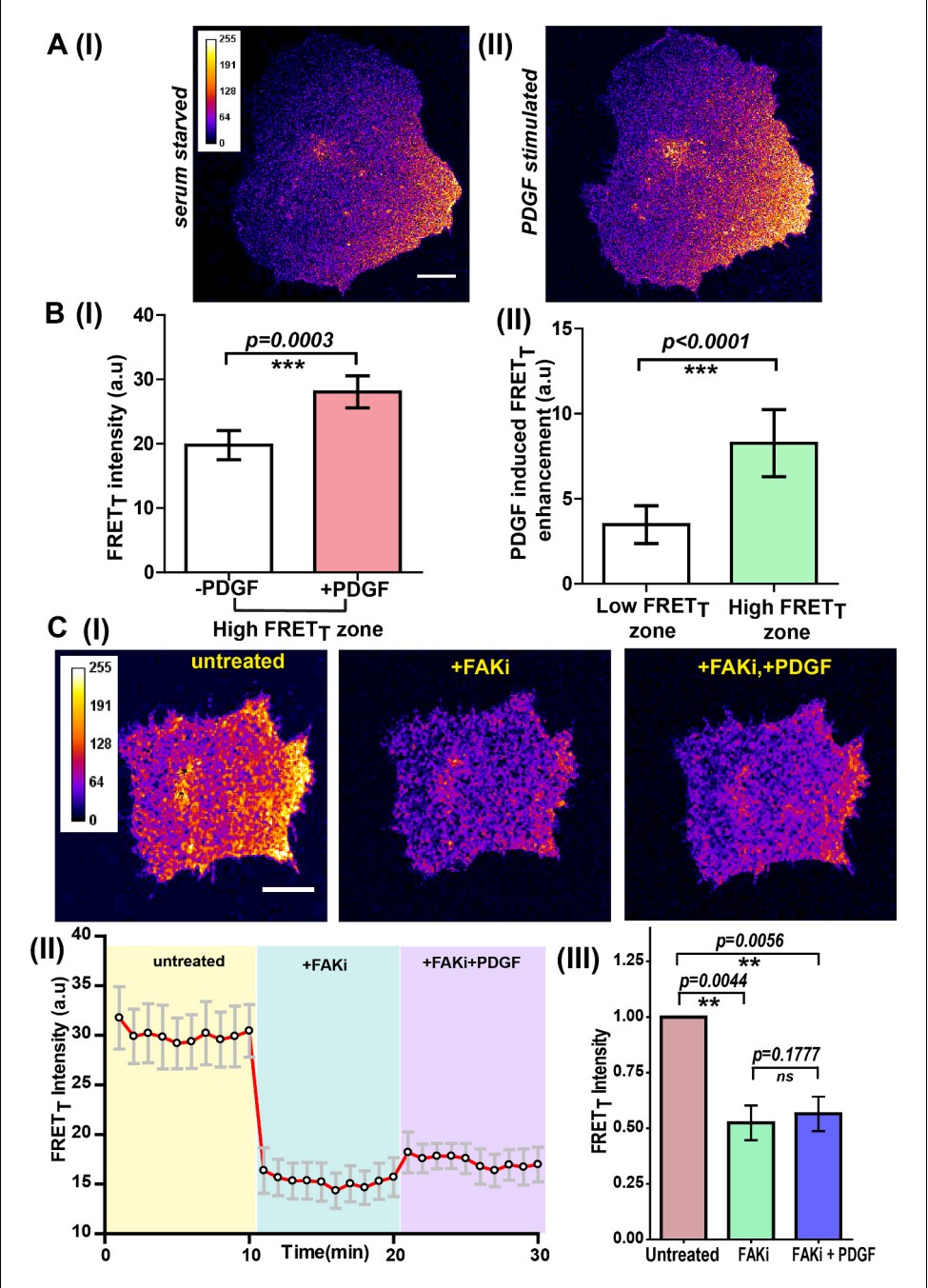

**Figure 8.** Integrin-dependent, spatially compartmentalized, pulsatile Fyn activity can be modulated by growth factor suggesting signaling crosstalk. (**A**) Spontaneously formed zones of active Fyn in serum-starved cells plated on FN can be further modulated by Platelet-derived growth factor (PDGF). Representative confocal fluorescence micrographs showing sensitized emission (FRET$_T$) levels indicative of active Fyn in serum-starved (I) and PDGF-stimulated U2OS cells (II) (Scale bar = 10 μm). (**B**) Quadrant analysis of cells reveal that on a global PDGF stimulation of serum-starved cells, the increase in Fyn activity observed was spatially localized (I) Bar graph comparing the mean of Max-FRET$_T$-HFQ before and after addition of PDGF (10 ng/ml) indicates the FRET$_T$ intensity increase in the high FRET zone on addition of PDGF. Values are mean ± s.e.m. Student's paired one-tailed t-test has been used to determine the p-value (n = 18 cells). (II) PDGF-induced enhancement in Fyn activity is greater in pre-activated intracellular zones. Bar graph show the difference in the mean intensities of the maximum FRET$_T$ (Max-FRET$_T$) obtained before and after PDGF addition for both 'high FRET' as well as 'low FRET' quadrants/zones across single cells. Quadrants were designated as 'HFQ' or 'LFQ' in FN-plated cells, prior to

*Figure 8 continued on next page*

*Figure 8 continued*

PDGF stimulation (n = 18 cells). Values are mean ± s.e.m. Student's paired one-tailed t-test has been used to determine the p-value. (C) Focal adhesion kinase (FAK) activity not only regulates Fyn activity but is required for the modulation/enhancement of Fyn activity by PDGF. (I) Confocal fluorescence micrographs showing sensitized emission (FRET$_T$) levels in untreated/serum starved cells, after treatment with FAK inhibitor (FAKi, PF-562271 and then followed by further treatment with PDGF. Scale bar = 10 µm. FAK inhibition attenuates FRET levels (active Fyn levels) in FN-plated serum-starved cells AND abolishes the enhancement and modulation of Fyn activity by PDGF. FAKi treated cells fail to respond to PDGF as measured through *FynSensor* FRET levels. (II) The mean FRET$_T$ intensity profile for the high FRET$_T$ quadrant in 5 cells plotted over time. Also, marked in color are experimental condition or the treatments; serum starved cells (yellow, t = 0–10 mins), + FAKi (cyan, t = 10–20 mins) AND +FAKi+PDGF (lavender, t = 20–30 mins). Values are mean ± s.e.m. (III) Whole cell FRET$_T$ data from all the cells (n = 5) for untreated cells, +FAKi, and +FAKi+PDGF. The mean intensity value of untreated cells has been used to normalize the data. Student's two-tailed paired t-test has been used to determine the p-value. Values are mean ± s.e.m. Data show no significant enhancement is seen in Fyn activity by PDGF stimulation, if cells are pre-treated with FAKi.

The online version of this article includes the following video, source data, and figure supplement(s) for figure 8:

**Source data 1.** Quantification of PDGF-induced enhancement in Fyn activity across different cellular zones.
**Source data 2.** Quantification of FRET$_T$ levels in untreated cells, followed by FAK inhibitor treatment and further treatment with PDGF.
**Figure supplement 1.** FAK activity is required for modulation of Fyn activity by PDGF.
**Figure supplement 1—source data 1.** Quantification of FRET$_T$ levels in untreated cells, followed by FAK inhibitor treatment and further treatment with PDGF.
**Figure 8—video 1.** Fyn activity in U2OS cells is modulated by PDGF.
https://elifesciences.org/articles/50571#fig8video1
**Figure 8—video 2.** FAK inhibitor treated, U2OS cells fail to respond to PDGF as reported through *FynSensor*.
https://elifesciences.org/articles/50571#fig8video2

---

*FynSensor* shows Fyn activity to be compartmentalized even in serum-starved cells. This is likely due to differential integrin signaling, since Fyn activity is significantly attenuated upon FAK inhibition. This spatially constrained Fyn activity could be further increased through growth factor stimulation, with greater enhancement seen in pre-activated areas despite a global stimulation. These results provide a direct illustration of dynamic and intracellular localized signaling crosstalk between integrins and growth factor receptors; visualized through the activation (conformational change) of a non-receptor tyrosine kinase that get activated by each of these receptor classes. While results so far imply that differential integrin signaling appears to spatially restrict the effect of growth factor, we asked if appropriate integrin signaling is *required* for growth factor mediated modulation of Fyn activity. For this, we inhibited focal adhesion kinase activity and specifically tested if growth factor is still able to modulate Fyn activity levels. Interestingly, when *FynSensor* U2OS expressing cells are treated with PF-562271 (FAK inhibitor), not only are the *FynSensor* FRET levels drastically reduced, the cells become insensitive to growth factor stimulation as measured through Fyn activity. PDGF-stimulation of inhibitor treated cells show little or no change in *FynSensor* FRET levels (*Figure 8C*, *Figure 8—source data 2*, *Figure 8—figure supplement 1*, *Figure 8—figure supplement 1— source data 1*, *Figure 8—video 2*). These data indeed confirm a robust, spatio-temporally modulated and functional crosstalk between integrin and growth factor signaling.

*FynSensor* reveals striking spatio-temporal patterns of Fyn activity. Since Fyn is a key regulator of cell physiology, including cytoskeleton remodeling and adhesion dynamics, we also asked how Fyn activity correlates with membrane motility. For this we acquired images at faster acquisition speeds (~35 s/frame) and performed quantitative image analysis using the Fiji plugin ADAPT (*Barry et al., 2015*) and measured both membrane motility and FRET$_T$ along the periphery of the cell, over the timecourse of the experiment (*Figure 9A* I-II, *Figure 9—source data 1*). Intriguingly, a plot of membrane motility versus FRET$_T$ levels shows that the two parameters tend to be inversely correlated (*Figure 9B*). This is an interesting trend suggesting that intracellular regions showing higher Fyn activity is likely to show reduced overall membrane motility changes. This link between Fyn activity and dampened cell membrane oscillations points to a 'poised' cell membrane when/where Fyn is active. Such a notion is consistent with Fyn's role in regulating cell-matrix adhesions and may need to be probed further (*Figure 9C*). It is remarkable that visualizing the conformational dynamics of a

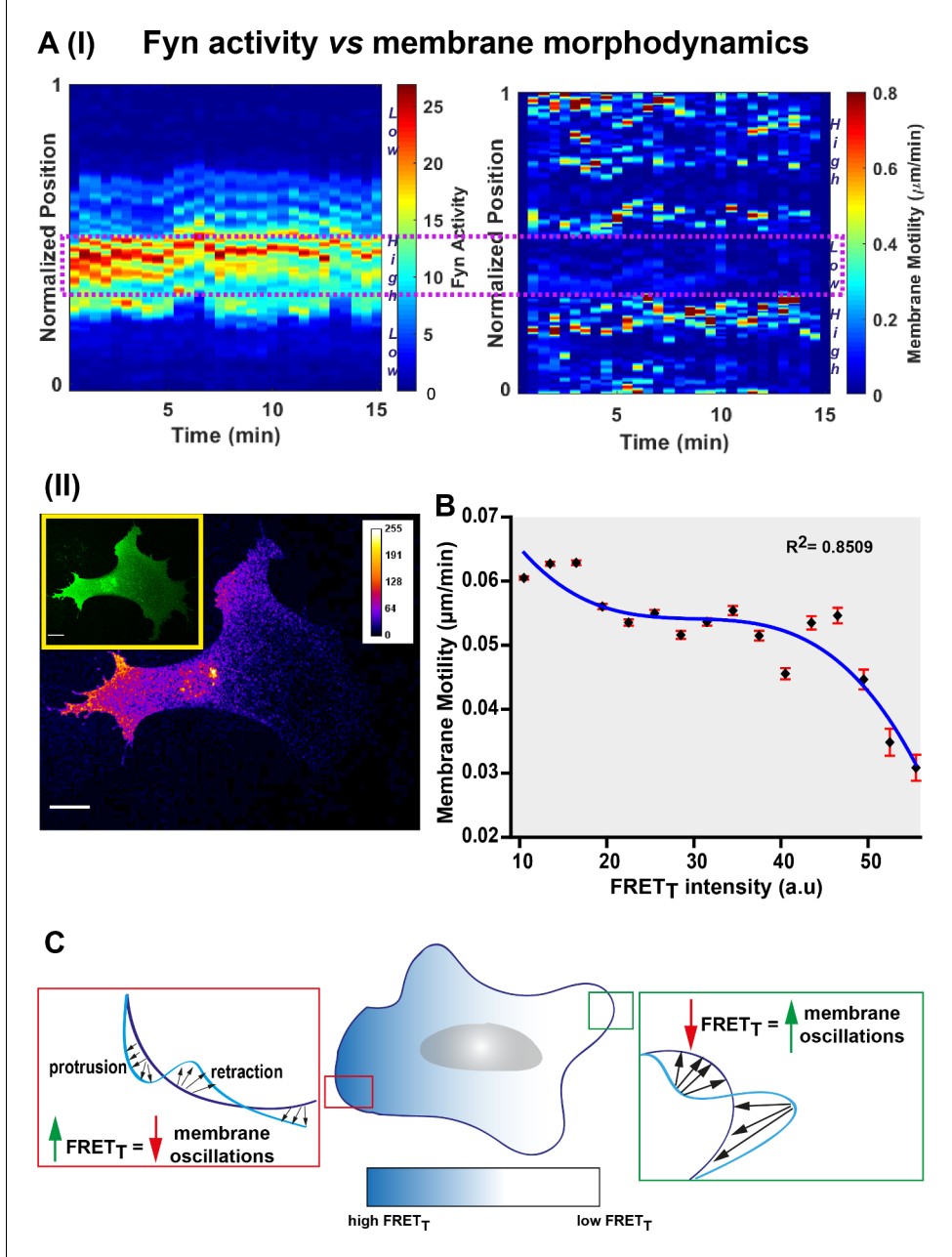

**Figure 9.** Fyn activity couples to cellular morphodynamics. (**A**) Fyn activity correlates with reduced amplitudes of membrane protrusion-retraction as revealed by simultaneous measurement of membrane motility and *FynSensor* FRET$_T$. Quantitative image analysis was performed using the ADAPT plugin [J Cell Biol (2015) 209 (1): 163–180]. (I) Representative 3D plots correlating Fyn activity (left) and membrane motility (right) for a single U2OS cell. Area within the dotted box marks region of high FRET$_T$ and low membrane motility. (II) A representative confocal micrograph (FRET$_T$ image) of the same U2OS cell showing Fyn activity to be spatially constrained (*Inset*: FRET channel image for clearly depicting the whole cell). (**B**) Graph shows the net membrane motility changes occurring over time, plotted against FRET$_T$ intensity for U2OS cells (n = 15 cells). FRET$_T$ levels are inversely correlated to membrane motility (Plot fitted using the equation f = y0+a*x+b*x2+c*x3; R (2) value = 0.8509). Values are mean ± s.e.m. (**C**) An illustration of Fyn activity being negatively correlated to membrane dynamics wherein, cellular zones showing higher Fyn activity (high FRET$_T$) show reduced overall changes in membrane motility (lesser constitutive protrusion/retraction cycles).

The online version of this article includes the following source data for figure 9:

**Source data 1.** Quantification of correlation between membrane motility and Fyn activity across multiple U2OS cells.

membrane-bound, non-receptor kinase reveals pulsatile patterns, which can be modulated through integrin and growth-factor signaling.

## Discussion

Direct visualization of protein activity is essential in order to gain a quantitative understanding of dynamic signaling networks that govern cell behavior. Despite the critical roles played by Src family kinases (SFKs) in regulating physiology (*Chetty et al., 2015*; *Gujral et al., 2014*; *Kim et al., 2009*; *Lewis-Tuffin et al., 2015*; *Nygaard et al., 2014*; *Saad, 2009*; *Timpson et al., 2001*; *Zhang et al., 2013*; *Zhang et al., 2014*), specific tools/sensors to image activity of individual kinases in live cells and tissues are not available. This is particularly important as individual SFKs can perform overlapping but specific, even seemingly opposing roles, to control cellular output (*Zhang et al., 2014*; *Kuo et al., 2005*; *Lowe et al., 1993*; *Lowell and Soriano, 1996*; *Marchetti et al., 1998*; *Molina et al., 1992*; *Palacios-Moreno et al., 2015*). Therefore, understanding the specific activity patterns of individual kinases assumes considerable importance. Our work clearly addresses this issue and establishes a platform for developing new biosensors for visualizing activation of individual Src kinases. Using combinatorial library screening and protein engineering, we develop a biosensor for the critical SFK, Fyn. This specific biosensor reports the activation dynamics of Fyn in live cells, with no interference from other kinases. Fyn kinase is a major regulator of multiple cellular processes and has emerged as a key player in various disease pathologies (*Nygaard et al., 2014*; *Bhaskar et al., 2005*; *Schenone et al., 2011*; *Chin et al., 2005*), but this is the first direct visualization of Fyn activity in cells. Interestingly, currently there are no tools for imaging active Fyn, with even specific antibodies not being available.

Our approach is extremely general and can be used for building sensors for other Src family kinases that are critical players in homeostasis and diseases. In this light, our work on addressing the specificity of biosensor binding using NMR structural analysis is likely to be significant. While our targeted screening and binding analysis yields a new binder (F29) that is highly specific for Fyn (*Figure 1*), the NMR structure of F29 bound to Fyn SH3 provides fascinating new insights into the molecular basis of specificity (*Figure 2*, *Figure 2—figure supplement 1*). We identify key interactions that mediate binding and further use this insight to switch specificity of binding (*Figure 2*). For instance, our Fyn-specific binder F29 does not or weakly bind Src-SH3 (*Figure 1C,E*). However, based on predictions from structural analyses, making just a single residue change in Src SH3 allows it to now bind F29 better (*Figure 2—figure supplement 3C*). These findings offer the possibility of generating specific binders targeting various Src family kinases, leading to new biosensors and tools. More broadly, we showcase how targeted screening of combinatorial protein libraries can be deployed to generate genetically encoded biosensors for visualizing active conformations of signaling proteins. This is a generally applicable strategy for biosensor development. With this method, it is possible to develop sensors even for 'difficult' targets, where structural information is limited and conventional tools like antibodies do not exist.

We demonstrate that *FynSensor* is able to report on the activation status of Fyn in cells. Newly engineered binder F29 binds the active form of Fyn and this binding leads to a FRET response that can be measured in live cells (*Figures 3–9*). Our sensor design and extensive control experiments ensures high sensitivity while minimizing perturbation. We have shown that the fluorescently labeled Fyn retains kinase activity, is fully functional and behaves like the unmodified kinase in terms of its regulation, localization and ability to modulate downstream signaling (*Figure 3—figure supplements 2* and *5–11*). Importantly, the fluorescently labeled Fyn is also able to complement the endogenous Fyn in RNAi-knockdown rescue experiments. For instance, in knockdown-rescue experiments, labeled Fyn is able to rescue downstream ERK phosphorylation similar to untagged Fyn, showing that our tagging of Fyn does not perturb its regulation and signaling function. We also demonstrate that under our imaging conditions (low-to-moderate expression of *FynSensor*) in two very distinct cell types, biosensor does not perturb either downstream signaling or cellular morphodynamics, reiterating the efficacy of our design (*Figure 3—figure supplements 6–11*). When the non-binding control version of the F29 binder (F29P41A) is used in cells, the FRET response is abolished, showing FRET to be dependent on F29 recognizing activated Fyn (*Figure 5B*, *Figure 5—figure supplement 1*). Further, an inhibitor of kinase activity also significantly attenuates FRET

response, showing the biosensor readout to reflect kinase activity (*Figure 5C*). Overall, we show that the *FynSensor* response is specific and reflects cellular activity of Fyn.

A significant advantage of *FynSensor* is that it is highly specific and directly reports on the active conformation of Fyn with high spatial and temporal precision, unlike previously reported kinase sensors. This direct visualization of the active form of a single kinase, within a critically important yet complex family of kinases, has led to new insights into signaling dynamics and regulation. *FynSensor* imaging reveals Fyn activity to be localized and temporally modulated (*Figures 4*, *5*, *6*, *7* and *8*), with greater activity closer to the cell edge. Further, as serum-starved, fibronectin (FN)-plated cells undergo constitutive cycles of protrusion and retraction, we observe intracellular zones of high Fyn activity and even spontaneous cell polarization measured through levels of active Fyn. Critically, we show that this compartmentalized Fyn activity is dependent on integrin signaling. Inhibition of focal adhesion kinase, a key mediator of integrin signaling, abolishes or greatly attenuates Fyn activity. When FN-plated cells with spatially localized Fyn activity are treated with platelet-derived growth factor (PDGF), we observe a *greater* increase in Fyn activity in and around the zones that were already pre-activated. This is indeed remarkable. Despite global PDGF treatment, its effect is highly localized and is dependent on intracellular zones that are established through integrin signaling. Our results suggest a model wherein localized/differential integrin activation in cells undergoing constitutive protrusion-retraction cycles not only helps establish signaling zones/compartments with higher Fyn activity, but also sensitizes these zones to be more responsive to growth factor signaling (*Figure 10*). This could be through either preferential localization or pre-sensitization of PDGF-receptors and/or associated signaling components (*Figure 10*). Thus, compartmentalized integrin signaling is not only maintained but also 'functionally enhanced' through growth factor stimulation. We further show that integrin signaling, not only spatially constraints Fyn activity and subsequent growth factor response, but that appropriate integrin signaling is required for growth factor modulation of Fyn activity. Inhibiting focal adhesion kinase activity not only dramatically attenuates Fyn activity in FN-plated, serum-starved cells, it also renders Fyn insensitive to any further stimulation through PDGF (*Figure 8C*).

Our results offer a striking demonstration of spatially localized crosstalk between integrin and growth factor signaling and shows that integrin can localize and regulate the effect of even globally applied growth factors in activating downstream signaling (*Figure 8*). This illustrates how by visualizing the dynamic activation of a key signaling node (Fyn), it is possible to directly visualize signaling crosstalk between receptors in cells. Prior work has specifically implicated Fyn for its ability to integrate information emanating from different cell-surface receptors (*Colognato et al., 2002*; *Colognato et al., 2004*). For instance, Fyn is plays a crucial role in cell survival and myelination of neurons by oligodendrocytes, specifically by integrating integrin signaling with growth factor signaling. Oligodendrocyte precursors are able to survive and subsequently wrap neurons with a myelin sheath, even in low growth factor conditions through the additive effects of integrin and RTK signaling (*Laursen et al., 2009*; *Laursen et al., 2011*; *Laursen and Ffrench-Constant, 2007*; *Sperber and McMorris, 2001*; *Schäfer et al., 2016*). This signal integration is mediated through Fyn activity (*Laursen et al., 2009*; *Schäfer et al., 2016*). It is striking and highly significant that our biosensor reveals evidence of such signal integration, in single cell imaging experiments in-vitro. This shows that compartmentalized signaling crosstalk between different receptor classes may be a key feature of signaling systems (*Figure 10*).

Biosensor imaging also shows Fyn activity signals to be pulsatile. Recent work has shown that oscillatory signaling patterns are key hallmarks of several signaling modules, including regulation by transcription factors (*Imayoshi et al., 2013*; *Isomura and Kageyama, 2014*) as well as MAP kinases (*Weber et al., 2010*; *Sparta et al., 2015*; *Albeck et al., 2013*; *Antoine-Bertrand et al., 2016*; *Shankaran et al., 2009*). Generally, pulses arise due to the presence of negative feedback loops, which may also be accompanied by additional modulators. Interestingly, there is some evidence that switching temporal activity patterns can alter cellular fate; for instance, pulsatile ERK activity leads to one output while persistent activity leads to another. We observe robust and spontaneous activity pulses of Fyn in very distinct cell types even in serum-starved cells plated on FN (*Figure 7*), and these pulses are further modulated by growth factor stimulation. This behavior strongly suggests the presence of rapidly activated, negative feedback loops that directly modulate Fyn activity.

Fyn activity pulses may be functionally significant in light of its demonstrated role in regulating cell migration, adhesion and actomyosin remodeling. In migration, there is an intrinsic periodicity in

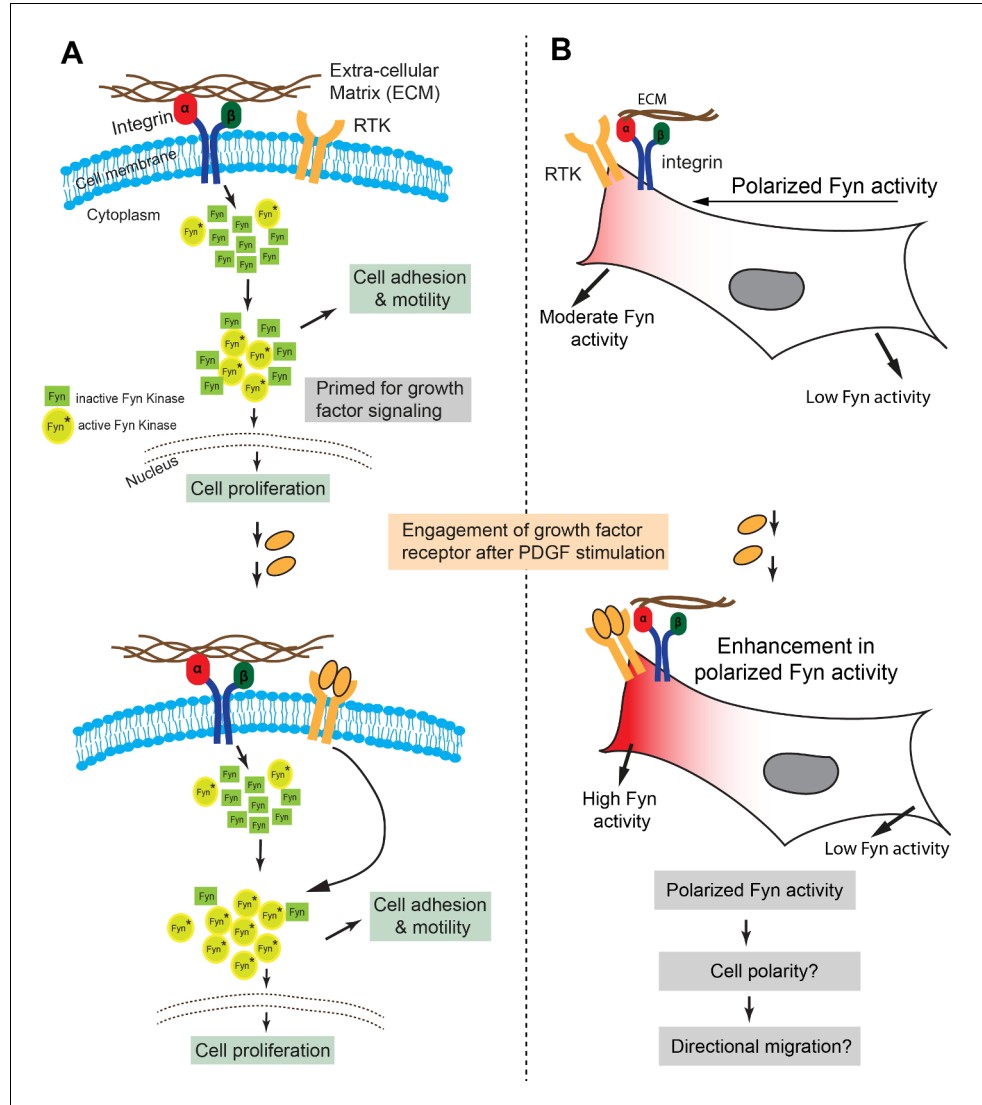

**Figure 10.** Model representing how Fyn activity helps mediate spatio-temporally localized signaling crosstalk. (**A**) Src kinases have been long known to associate with the integrin receptor and growth factor receptor (RTK) signaling, thereby placing these kinases at nodal spots for signal integration. We provide the first visual proof for Fyn's involvement in integrating signals from these two receptor classes and see that the cell is highly polarized with respect to Fyn kinase activity. *FynSensor* shows that integrin receptor engagement results in formation of 'zones' of active Fyn in the cell, causing the cell to become polarized with respect to the activity of the kinase, even in serum-starved, FN-plated cells. PDGF stimulation further increases Fyn activity preferentially in these pre-sensitized regions only, thereby strengthening the spatially localized response. (**B**) Spatio-temporally localized Fyn activity; specifically, set-up through signal integration offers a quantitative picture of how Fyn activity may help regulate cell fate in certain situations. Fyn is known to be crucial for context-dependent cell migration and cell state switching in normal as well malignant cells, therefore, a spatio-temporally polarized response may be important for setting up cell output, especially for processes like that of directional cell migration and adhesion dependent cell switching. Pulsatile Fyn activity further suggests pre-organized signaling clusters with negative and positive feedback loops sensitive to both integrin and growth factor signaling.

cycles of actin polymerization/remodelling/depolymerisation, periods of membrane remodelling as well adhesion assembly and disassembly. A protein that regulates several of these processes is conceptually much more likely to have pulses of activity versus sustained activity. In this light, it is remarkable that we are able to pick up 'activity pulses' at specific locations in the cell. This would need to be explored further in the context of directed migration and control of cell proliferation.

Further, its indeed striking that membrane motility is reduced in the regions/zones where Fyn is more active (*Figure 9*). Spatially-localized pulsatile activation may likely cause transient but limited stabilization of signaling clusters which would be difficult to achieve with sustained activation. These observations taken together suggest that Fyn activity may be correlated with a 'poised' membrane state and formation and stabilization of specific integrin-FAK dependent signaling clusters, which can be further tuned by growth factor signaling. Such specific control of adhesion and actomyosin dynamics may be important in Fyn's ability to mediate directed cell migration, growth control and tissue invasion. Context dependent and spatio-temporally regulated Fyn activity may provide a new perspective on understanding Fyn's roles in regulating diverse and sometimes opposing cellular functions. Fyn activity profiles also provide a direct readout of highly localized signaling crosstalk between distinct receptor classes. Overall, this work sets the stage for detailed investigations of such functional signaling crosstalk in numerous cellular contexts and offers a new paradigm for direct visualization of signaling dynamics mediated by Fyn and other Src kinases.

# Materials and methods

## Chemical and reagents

i.   *Sigma-Aldrich-* IPTG, EDTA, Glutathione, BSA, Manganese(II) chloride, KCl, Bromophenol blue, Potassium acetate, D(+)Galatose, CaCl2, D(+)Glucose, *p*-Coumaric acid, Luminol, DMSO, Imidazole, Biotinamidohexanoic acid N-hydroxysuccinimide-ester, Ampicillin Sodium salt, Penicillin-Streptomycin, Yeast Nitrogen base without amino acids, Fibronectin solution, phenylmethylsulfonyl fluoride, Platelet-derived Growth Factor, Ethidium Bromide, puromycinand Tween-20 and Dialysis tubing.
ii.  *Fisher Scientific-* Disodium Phosphates, Sodium Dihydrogen Phosphate, Tris-buffer.
iii. *HiMedia Biosciences-*NaCl, SDS, McCoy's 5A media, Luria Broth and Agar, FBS.
iv.  *Merck-* Anhydrous Glycerol, Glacial Acetic acid, Hydrochloric acid and Isopropanol
v.   *Thermo-Fisher Scientific-* Dynabeads Biotin Binder (#11047), PVDF-membrane, Lipofectamine LTX and plus reagent, Lipofectamine 3000, RIPA Lysis buffer, Pierce Phosphatase Inhibitor tablets, Pierce BCA assay kit, Page-Ruler Plus prestained protein ladder (10 to 250 kDa), Ni-NTA agarose beads, Agarose, DMEM (high glucose), DMEM (high glucose, HEPES, -phenol red), 1X DPBS, GlutaMax.
vi.  *Bio-RAD-* 30% acrylamide/Bis solution, Stacking and resolving gel buffer, TAE electrophoresis buffer and Econo-chromatography Columns.
vii. *GE healthcare-* Glutathione sepharose 4 Fast Flow and Superdex-75GL Size Exclusion Chromatography Column.
viii. *Zymoresearch-* Zymoprep yeast plasmid miniprep II kit.
ix.  *Promega-* Pure-yield plasmid miniprep and wizard SV gelPCR clean-up system.
x.   *Roche-* Protease inhibitor cocktail tablets, APS.
xi.  *NEB-*Phusion high-fidelity DNA Polymerase, Q5 high fidelity DNA polymerases, T4 DNA ligase, *Dpn*I, *Nco*I-HF, *Bam*HI-HF, *Eco*RI-HF, alkaline phosphatase (CIP), 1 kb and 100 bp DNA Ladder.
xii. *Clontech-*TaKaRa Ex-Taq, LA-Taq r-Taq DNA Polymerase.
xiii. *EMD-Millipore-* SU6656 (572635–1 MG).
xiv. *Selleckchem-*FAK inhibitor PF-562271 (Catalog No. S2890).
xv.  *Santa Cruz Biotechnology-*Polybrene (sc-134220).

## Antibodies

i.   *Invitrogen-c*-Myc chicken IgY antibody; goat anti-chicken IgY (H+L) secondary antibody; Alexa Fluor 633 conjugate; Avidin-Neutravidin FITC conjugate and Streptavidin R-phycoerythrin conjugate (SAPE). Goat anti-mouse IgG secondary antibody-Alexa Fluor 647 conjugate; and Donkey anti-rabbit secondary antibody-Alexa Fluor 546 conjugate.
ii.  *Cell Signaling Technology-* phospho-Src family (Tyr416) antibody; phospho-Src (Tyr527) antibody; total Fyn antibody; β-actin antibody; p44/42 MAPK (Erk1/2) antibody, Phospho-p44/42 MAPK (Erk1/2) (Thr202/Tyr204) antibody, Anti-mouse IgG-HRP-linked antibody and Anti-rabbit IgG- HRP-linked antibody, GFP (D5.1) XP Rabbit mAb (used in *Figure 3—figure supplements 6* and *10*).
iii. *Abcam-* anti-GFP antibody, Anti-Fyn Mouse monoclonal antibody (IF grade)

iv. *Sigma*-monoclonal anti-polyhistidine antibody, monoclonal anti-vinculin antibody, anti-Src polyclonal antibody produced in rabbit, anti-yes polyclonal antibody produced in rabbit.

Note: Please see *Supplementary file 3*.

### Primers procurement and DNA sequencing

Primers used in study were procured from Bioserve Biotechnologies (India). The oligonucleotide synthesis scale was 25 nmol. Primers up to 50 nucleotide lengths were desalted, however, longer primers were PAGE purified. All DNA sequencings were performed on Illumina- MiSeq DNA Sequencer machine at the DNA sequencing facility at NCBS.

### Cell-culture

## Cell line authentication and maintenance

Human Embryonic Kidney Cells (HEK-293T), Mouse Muscle Myoblast (C2C12) and Human Bone Osteosarcoma Epithelial Cells (U2OS) were used. HEK-293T and C2C12 cells were cultured in DMEM media supplemented with 10% FBS with 1% Penicillin Streptomycin (Pen-Strep, $10^3$ U/mL) solution. U2OS cells were grown in McCoy's 5A media supplemented with Sodium Carbonate, 10% FBS and 1% Pen-Strep.

Cell line authentication and analysis was done following recommended standard tests. Cells were periodically tested for the presence of Mycoplasma and found to be negative for mycoplasma. Specifically, we periodically performed mycoplasma testing using the commercially available and standard MycoAlert Mycoplasma Detection Kit (Lonza Pharma and Biotech) as per manufacturer's instructions. Using this high sensitivity luminescence based biochemical test, we confirmed that all cell lines used were free of Mycoplasma. Cell lines were also periodically authenticated using recommended tests. Periodic morphology analysis was performed on all cell lines. We routinely observed and recorded cell morphology features, at different dilutions and performed microscopy at high and low culture densities. We additionally monitored cell growth and analyzed the response of cell cultures to stimulus like serum and platelet derived growth factors. Cells showed expected response times to these stimuli. Population doubling times were periodically recorded. Cell lines showed growth properties that were consistent. Cell spreading on fibronectin and morphology was also systematically examined and were found to be consistent over the period of the study. We also ensured that cell lines were handled separately, with a minimum gap time of 15 min before handling separate cell lines.

## C2C12 mouse muscle myoblasts cell line

C2C12 mouse muscle myoblasts cells were purchased from ATCC (catalog # CRL-1772, lot #70013341). These cells were further authenticated by observing cell and quantifying cell morphology, growth, spreading and response to stimuli. The cell behavior was found to be consistent throughout. These cells were used for biosensor imaging and analyses. For current experiments, all cells were used between passage # 3–10.

## Human bone osteosarcoma epithelial cells line (U2OS)

U2OS cells were a kind gift from Professor Satyajit Mayor (NCBS). This group has recently reported use of this cell line for cellular imaging (*Kalappurakkal et al., 2019*). Cell morphology, growth characteristics, cell spreading and morphology patterns and response to stimuli were carefully observed and found to be consistent throughout the period of the study. U2OS cell morphology was found to be consistent with literature.

## Human embryonic kidney 293 T cells (HEK293T cells)

HEK293T were a kind gift from Professor Jyotsna Dhawan (inStem). This group has recently reported use of this cell line (*Saleh et al., 2019*). Cell morphology, growth characteristics, cell spreading and morphology patterns and response to stimuli were carefully observed and found to be consistent throughout the period of the study. For current experiments, all cells were used between passage # 8–20.

## Protein purification and binder screening

### Sub-cloning of SH3 domains and binder F29

The sequence of thrombin cleavage site 'thr' (*LVPRGSH*) of vector pet14b (kind gift from Dr. Brian Kay Lab, University of Illinois, Chicago) was changed to Prescission protease cleavage site 'ppx' (*LEVLFQGP*) using the quick change mutagenesis (*Figure 3—figure supplement 1*) with SN 1, 2 primes to yield plasmid p-SUMO-ppx-Fyn SH3. Similarly, ppx site was introduced in other SH3 domains. Plasmids p-SUMO-ppx-Src SH3-T99D has been constructed using site directed mutagenesis with primers SN 4, 5. Binder F29 ORF was amplified (primers SN 6, 7) and product was digested with *Nde*I and *BamH*1 and sub-cloned in p-SUMO-ppx-Fyn SH3 to yield plasmid p-SUMO-ppx-F29. Furthermore, F29-ORF was sub-cloned along with linker $(GGGS)_3$ into pGX6P-1 vector under *Bam*H1 and *Xho*I site using overlap extension PCR with primers SN 8,9 yielding plasmid p-GST-ppx-$L_{15}$-F29 (*Supplementary file 2 A, B*).

### Purification of SH3 domains

Domains of individual Src kinase were encoded in a pET-14b plasmid, so that a '6X-His-SUMO-Thr-SH3-domain' fusion protein could be expressed. The plasmid was transformed into competent *E. coli* BL21-(DE3) cells using heat shock method. The late-log phase grown *E. coli* cells (OD ~0.8) were induced with 0.5 mM of IPTG for 3 hr at 37°C, at 200 rpm in a shaker incubator. Cells were harvested and washed once with 1X PBS buffer by centrifugation at 8,000 × g and then re-suspended 1:4 wt./vol in ice-cold Buffer-A (50 mM sodium phosphate buffer, 500 mM NaCl, pH 7.5) having 10 mM imidazole followed by sonication with an ultrasonic processor on ice, with a total of four cycles consisting of 3 min cycle of 4 s ON time and 10 s OFF time at 40% amplitude. The cell-homogenate was centrifuged at 35,000 × g for 35 min. The clear supernatant cell-free extract (CFE) was incubated for 4 hr with Ni-NTA beads (Invitrogen) at 4°C under mild rotation condition. The CFE with beads were packed on to glass econo-columns (Bio-Rad). The unbound protein was washed twice with Buffer-A containing 10 mM imidazole and elutions were carried out using a step-gradient of Buffer-A containing imidazole (50 mM to 250 mM, 2 ml each). Imidazole was removed by dialysis against Buffer-A. Protein fractions were further purified using Superdex 75 GL size exclusion chromatography column (AKTA pure, GE healthcare life science) with Buffer-A. The protein preps were analysed using 12% SDS-PAGE for their purity and homogeneity. The fractions containing the protein of interest were pooled and concentrated using Amicon ultra-centrifugal filter 10 K (Millipore).

### Binder screening steps

Stable Sso7d protein of *Sulfolobus solfataricus* was used as scaffold in binder screening. Earlier, a library of ~$10^8$ Sso7d mutants had been generated by randomizing 10 amino acid residues on the DNA-binding surface of Sso7d, using yeast surface display platform (*Gera et al., 2013*). The Sso7d scaffold library in the yeast surface display platform has been used to isolate the specific binder for SH3 domain of SFK, Fyn. The screening procedures were followed as described earlier (*Gera et al., 2013*) and are explained below and in *Figure 1—figure supplement 1*.

i. *Target biotinylation*- SH3-domains of SFKs were biotinylated using Biotinamidohexanoic acid N-hydroxysuccinimide-ester followed by purification using SephadexG-10 column.

ii. *Target coating on magnetic beads*- Biotinylated SH3 domains (1 μM) were pre-incubated with $2.5 \times 10^6$ dynabeads biotin binder in 1X PBS (100 μl system) at 4°C at 5 rpm on 360° rotor. Target preloaded beads were washed twice with PBS–BSA (0.1% BSA in 1X PBS) and captured using the Dyna-Mag2 magnet.

iii. *Expansion and induction of yeast cell library*-~$2 \times 10^8$ cells were cultured from frozen stock of yeast cell library of Sso7d in 1L of SDCAA media with pen-strep (1:100) and incubated at 30°C, 200 rpm shaker for 72 hr. From the expanded culture,~$10^9$ cells were inoculated into 50 ml of SGCAA media having pen-strep for 20 hr at 30°C in order to induce the expression of mutant proteins on the yeast cell surface.

iv. *Magnetic screening*-~$10^9$ cells from the induced culture (*step iii*) were centrifuged at 2500 x g for 3 min and the pellet dissolved in 1 ml of PBS-BSA. Two rounds of negative selection against beads were carried out by incubating cells with the washed beads for 1 hr at 4°C in a microcentrifuge tube, in a rotator followed by unbound yeast cell collection using magnet. The unbound yeast cells were incubated in a stepwise manner with 1 μM biotinylated target (*SH3 domains other than Fyn*) in microcentrifuge tube for 1 hr at 4°C in a rotator followed by

unbound yeast cell collection using magnet in order to deselect binder. The probable Fyn SH3 binders were selected from previous unbound yeast cells by incubating with 1 μM biotinylated target (Fyn SH3 domain) in a microcentrifuge tube for 1 hr at 4°C in a rotator. The bead-yeast complex was isolated, washed twice with 1X PBS and transferred to a culture tube containing 5 ml SDCAA with pen-strep and incubated for 48 hr in a shaking condition at 30°C, at 200 rpm. The beads were removed using the magnet and the yeast cells were further inoculated in a 50 ml SDCAA culture with pen-strep for 48 hr in a shaking condition at 30°C, at 200 rpm.

v. *Screening using FACS*-The yeast cells obtained after magnetic screening were further screened using flow cytometry to isolate the highest affinity Fyn binders. In this step, in-order to isolate yeast cells bound to the target SH3 domain, two fluorescent signals were used. Target SH3 domain presence was detected by streptavidin, R-Phycoerythrin Conjugate (SAPE, against biotinylated SH3 domain). The yeast cells expressing full-length scaffold were detected by Alexa-Fluor 633 labeled secondary antibody against c-myc. Presence of dual staining in FACS experiments highlights events where yeast cells with scaffold bind the target (SH3 domain) and has been used for screening for binding and validation (*Gera et al., 2013*). The Fyn binder was isolated after six subsequent stringent sorting steps using Fyn SH3 domains as a target. Finally, clean population in FACS were gated and sorted. The specificity of the pool was determined by FACS against SFKs SH3 domain (*Figure 1—figure supplement 1*). Yeast cells from this pool were plated onto SDCAA agar plates and incubated at 30°C for 48 hr. More than 30 randomly picked colonies were inoculated in 5 ml SDCAA media and incubated for 48 hr at 30°C, at 200 rpm in a shaker incubator. Specificity of each clone was tested against other SFK SH3 domains (1 μM). We have selected a yeast clone number 29 for its specificity against Fyn SH3 domain compared to others using FACS (*Figure 1C*, *Figure 1—figure supplement 2*). Plasmid was isolated from the yeast clone no. 29 using the Zymoprep kit (Zymo-Corp USA) and the DNA was sequenced. This clone is hereafter referred to as 'F29' (Fyn binder 29).

## Protein purification for NMR studies

Fyn SH3 domain was encoded in a pET-14b plasmid, so that a 6X-His-SUMO-ppx-SH3 fusion protein can be produced. The plasmid was transformed into *E. coli* BL21-(DE3) cells, and grown in LB to produce unlabeled protein. $^{13}$C, $^{15}$N-labeled protein was produced by growing the *E. coli* in vitamin supplemented M9 minimal media with$^{13}C_6H_{12}O_6$ and (*Lewis-Tuffin et al., 2015*) NH4Cl as the sole source for carbon and nitrogen, respectively. Labeled/Unlabeled Fyn-SH3 domain was purified as follows: Cells were grown up to OD ~0.8, followed by induction with 1 mM IPTG. Four hours post induction, cells were harvested by centrifugation at 4°C, then washed once with Buffer-B (50 mM sodium phosphate buffer, pH 8.0, 300 mM NaCl), supplemented with 0.1% (v/v) Triton X-100, and then stored at –80°C. Frozen cells were re-suspended in Buffer-B supplemented with 2 mM phenylmethylsulfonyl fluoride (1:10, wt./vol.). The suspension was sonicated at 4°C for 30 min for lysis, followed by centrifugation at 4°C. The resultant supernatant was allowed to bind Nickel beads (Thermo Fisher), washed with 20 mM imidazole, and eluted with 100–200 mM imidazole. Fractions containing the fusion protein were pooled, and the imidazole was removed by dialysis against Buffer-B for overnight at 4°C.

The fusion protein (3 mg) was cleaved by Precission Protease (1 mg) in an overnight 1 ml reaction Buffer-B under mild rotating conditions at 4°C. Cleaved Fyn-SH3 was purified by gel-filtration. Fyn-SH3 fractions were pooled and dialyzed against Buffer-C (50 mM sodium phosphate buffer, pH 8.0) overnight at 4°C. The dialyzed protein was applied to a Mono Q column equilibrated in Buffer-C. The protein was eluted with a linear gradient from 0% to 100% of Buffer-D (50 mM sodium phosphate buffer, pH 8.0 with 500 mM NaCl). Fractions containing pure Fyn SH3 were pooled and concentrated after dialysis in Buffer-B (50 mM sodium phosphate buffer, pH 8.0, 300 mM NaCl). p-GST-ppx-L$_{15}$-F29 plasmid was transformed into *E. coli* BL21 (DE3) cells. The proteins were over-expressed in suitable media as given above. Post lysis, the supernatant was passed through a column of glutathione-Sepharose beads (GE-Healthcare) in Buffer-C. After extensive washing, the bound GST-ppx-L$_{15}$-F29 fusion protein was eluted with 15 mM glutathione in Buffer-C. Fractions containing the fusion protein were pooled and allowed dialyze overnight at 4°C in Buffer-C. The fusion protein was cleaved by Precission Protease. Cleaved F29 was purified by size exclusion chromatography. Fractions containing the F29 were pooled and concentrated up to 1mM for NMR studies.

## Binder-Fyn SH3 domain complex structure determination

*NMR Studies-* All NMR spectra were acquired at 298K on an 800 MHz Bruker Avance III spectrometer (*NMR facility-NCBS*). Assignment of backbone resonances of the proteins were carried out using HNCACB and CBCACONH experiment. $^{1}$H and $^{13}$C resonance assignments of side chain atoms in F29 were obtained by collecting H(CC)CONH and (H)CC(CO)NH spectra, respectively. A 3D $^{15}$N-edited NOESY experiment was carried out on a complex of $^{15}$N, $^{2}$H-labeled Fyn-SH3 and unlabeled F29. All the experimental data were processed using NMR pipe (*Delaglio et al., 1995*) and TOP-SPIN3.2 software. Analysis of NMR data was carried out by using Sparky (*Kneller and Kuntz, 1993*). Backbone assignments were obtained by PINE-NMR (*Lee et al., 2009*) and confirmed manually. Structural model of F29 was calculated by CS-ROSETTA (*Lange et al., 2012*). NMR titration was performed by titrating unlabeled F29 (ligand) to a sample of $^{15}$N-labelled Fyn-SH3 domain (protein). At least six different protein:ligand ratios were collected ranging from 1:0.5 to 1:4. $^{1}$H-$^{15}$N HSQC spectra were taken at each titration point. The reverse-titration was done by adding unlabeled Fyn-SH3 to labeled F29.

*Structure calculation-* The structure calculation of the Fyn-SH3/F29 complex was performed using HADDOCK (*van Zundert et al., 2016*). The input structure of Fyn-SH3 is from pdb: 3UA6, and that of F29 is the output lowest energy structure from CS-ROSETTA. The ambiguous restraints were obtained from the CSPs observed in the NMR titration data. Unambiguous restraints obtained from the NOESY spectra were included in the structure calculation. All restraints were used during the docking steps. The interface of F29 and Fyn-SH3 were kept semi-flexible during simulated annealing and the water refinement.

## Pull-down of SH3 domain with binders

Pull-down experiments were performed to validate the interaction of binder F29 and Fyn SH3 domain. In the plasmid p-GST-ppx-L$_{15}$-F29, we have made P41A and R33A mutations in F29 ORF using site directed mutagenesis to yield plasmid p-GST-ppx-L$_{15}$-F29-P41A and p-GST-ppx-L$_{15}$-F29-R331 with primers SN 10,11 and 12,13 respectively.

i. *Bead immobilization with bait protein and pulldown-*The purified GST fused binder (GST-F29) or its point mutant (GST-F29-P41A and GST-F29-R33A) were incubated at room temperature for 2 hr with glutathione sepharose beads (i.e. 10 µl beads with100 µg of protein) under mild 360° rotating conditions (5 rpm). Incubated beads were washed four times with 1X-PBS buffer to remove the unbound, excess protein. From washed pool of GST-F29 saturated beads,~10 µl beads was incubated with varying concentrations of 6X-His-Fyn-SH3 domains (75 nM to 1.25 µM) in a 100 µl of reaction system for 4 hr at 4°C under mild rotating 360° rotations. Each tube was washed four times with 1 ml ice cold 1X-PBS to remove unbound, excess SH3 domain. The beads were then boiled with 30 µl of 1X Laemmli buffer for 10 min at 98° C. Using these samples, we have performed SDS gel (12%) electrophoresis for immunoblot assay. In order to show that GST protein does not interact with 6X-His- Fyn SH3 domain, we have performed a pulldown experiments with 10 µl glutathione sepharose beads saturated with 100 µg of GST protein and incubated with 1000 nM 6X-His-Fyn-SH3 domain in a 100 µl of reaction system using conditions similar to above. Similarly,500, 750 and 1000 nM of 6X-His-Fyn-SH3 was pulled-down using 10 µl beads saturated with GST-F29protein and was run on a 12% SDS-gel. The gel region spanning 29 to 43 kDa region was stained with Coomassie blue dye, and 29 to 16 kDa region was use in immunoblot assay as described below.

ii. *Immunoblot assay-*The proteins from SDS gel were transferred onto PVDF membrane (Invitrogen, 0.2 µm) at 100V for 90 min at 4°C in 1X transfer Buffer (25 mM Tris-HCl; 190 mM glycine and 20% methanol). After complete transfer the PVDF membrane was incubated (facing up) in blocking buffer (5% non-fat dry milk, Blotto) in 1X-TBST (20 mM Tris-Cl pH 7.6, 150 mM NaCl, pH 7.6, and 0.1% Tween 20) for 1 hr at room temperature on a shaker. The blot was then incubated in hybridization bags with a primary antibody (anti-polyhistidine antibody, 1:3000) in antibody dilution buffer (5% BSA in 1X-TBST) at 4°C, overnight under mild shaking conditions. The membrane was washed stringently with 1X-TBST and incubated with anti-mouse IgG, HRP-linked secondary antibody (1:3000) at room temperature for 1 hr. The membrane was washed 4 times with 1X-TBST and imaged using Image-Quant LAS 4000 (GE-Healthcare) in luminol-based enhanced chemiluminescence reagent. The band intensity of

blot was analyzed using Image-J(NIH). Similar experiments were performed with Src SH3 domain and its mutant as well as non-binder mutant etc.

iii. *Fyn SH3 domain and binder F29 dissociation constant ($K_D$) Calculation*-Quantified band intensities of pulled-down His-tagged Fyn SH3 domains (75 nM to 1.25 μM) from immunoblot (given in *Figure 1E*) has been used to calculate the Dissociation constant ($K_D$). The data from three such independent experiments were analysed using GraphPad Prism-5 where a rectangular hyperbola saturation binding curve was plotted (as shown in *Figure 1F*). In the graph, the curve shows saturation at a given concentration of ligand. This has been used to determine the $B_{max}$ and the following standard equation for 'one-site specific' binding has been used to calculate the $K_D$.

$$Y = \frac{B_{max} * X}{K_D + X} \tag{1}$$

Wherein:

X is ligand concentration (Fyn SH3, nM)

Y is amount of bound protein

$B_{max}$ is the maximum binding capacity (same units as the Y-axis)

$K_D$ is the equilibrium dissociation constant (same units as the X-axis, concentration).

## Pull-down of total endo-/exogenously expressed Fyn kinase using F29 binder

Glutathione sepharose beads saturated with GST-F29 and GST protein (bait protein) were used to pull-down the endogenous and exogenously expressed Fyn kinase (prey protein, regulatable; WT as well as open active CA) from the cell lysate of HEK-293T cells. We have followed an established protocol (*Sambrook and Russell, 2006*) with modifications as described below.

a. *Bait protein preparation*-The pGEX-6P-1 vector having ORFs of GST as well as fusion protein GST-F29 were transformed in *E. coli* BL21 competent cells. Single transformed bacterial colony was grown as pre-inoculum for 3 to 4 hr in a 3 ml of Luria-broth having ampicillin (100 μg/ml) in a 50 ml conical screw cap tube at 37°C, 200 rpm shaker. Grown pre-inoculums was transferred to a 100 ml glass flask having 20 ml Luria-broth and ampicillin and further grown for 2 hr. The cells were induced with IPTG (500 μM) at OD = 0.8 for 3 hr for protein expression. Cells were harvested by centrifugation at 7,000 × g and re-suspended in 1.5 ml of ice-cold lysis/wash buffer (1X PBS pH = 7.4, 5% glycerol) followed by sonication with an ultrasonic processor on ice, for a total three cycles constituting of 12 pulses of 3 s ON time and 5 s OFF time at 40% amplitude. The cell-homogenate thus obtained was centrifuged at 20,000 × g for 35 min at 4°C to yield clear cell-free extract (CFE). The total protein was estimated from CFE using BCA Protein assay kit (Pierce). The CFE was used to saturate the Glutathione sepharose beads and further used in pull-down experiments.

b. Endogenous kinase pull-down

i. *Cell culture and whole cell lysate*- HEK-293T cells were cultured in 100 mm plate (~$10^6$ cells) and grown for 48 hr at 37°C in 5% $CO_2$ incubator. Plates were allowed to reach 90% confluency. From the plates, cells were washed with 1X PBS and scrapped gently in 600 μL of RIPA lysis and extraction buffer cocktail (10 ml cocktail includes 1X Pierce RIPA buffer, 1 tablet of Pierce Phosphatase Inhibitor and 1 tablet of cOmplete mini Protease Inhibitor). The cells were centrifuged at 20,000 x g for 20 min at 4°C to remove the cellular debris.

ii. *Bead pre-clearance*-The whole cell lysate (*step i*) was mixed with 50 μl of glutathione sepharose beads and incubated at room temperature under mild 360° rotating conditions (5 rpm) for 2 hr. After centrifugation at 7000 × g for 1 min, the pre-cleared cell lysate was taken and total protein was estimated by BCA assay.

iii. *Bead incubation with bait protein*- Bacterial CFE containing fusion bait protein GST-F29 or GST (control) was incubated with glutathione sepharose beads (50 μl beads with ~7.5 mg of CFE) under mild 360° rotating conditions (5 rpm) for 2 hr at room temperature in a low protein-binding micro-centrifuge tube (Eppendorf). The saturated beads were centrifuged at 7000 × g for 1 min at 4°C and washed with 4 times with 1 ml of lysis/wash buffer.

 iv. *Pull-down*- Protein-saturated glutathione sepharose beads (~50 µl) were incubated with a total of 2.0 mg of pre-cleared cell lysate (from *step ii)* for 2 hr under mild rotation at room temperature. The unbound lysate was removed after centrifugation at 7000 × g for 1 min. The beads were washed thrice with 1X-PBS containing protease and phosphatase inhibitors + 0.1% NP40 under mild rotating conditions for 5 min at 4°C.

 v. *Immunoblot assay* -The remaining beads were incubated with 25 µl of 1X Laemmli buffer and denatured for 10 min at 98°C. Using the extracted sample, we have performed the SDS (8%) gel electrophoresis. The gel region spanning 29 to 43 KDa region (bait protein) was stained with Coomassie blue dye, however 53 to 90 kDa region (prey protein) was used for immunoblot using anti-Fyn antibody (1:500). The expression level of Fyn kinase was also analysed from 50 µg of pre-cleared cell lysate in a separate immunoblot experiment using anti-Fyn antibody. Here, β-actin antibody was used as loading control.

 c. Exogenous Fyn Kinase pull-down

 i. *Transfection, over-expression and whole-cell protein isolation–* HEK-293T cells were cultured in 60 mm plate as described earlier. The cells were transfected with Fyn (WT or CA) DNA construct (1.5 µg) using Lipofectamine LTX-DNA transfection reagents (Invitrogen) and incubated for 24 hr at 37°C in 5% $CO_2$ incubator. The whole cell protein isolation was performed as described above.

 ii. The steps used in pull-down experiment were kept identical as described above, except here, for each case only 25 µl of glutathione sepharose beads were used to saturate 3.75 mg of CFE having bait protein GST-F29 or GST proteins. Here a total of 1.0 mg of bead-pre-cleared lysate of HEK293T expressing WT or CA Fyn has been used. All subsequent immunoblotting steps were identical as described above.

## Biosensor construction

We have constructed intermolecular FRET-donor and acceptor biosensor using h-Fyn gene and binder (F29), respectively. In the FRET-donor, mCerulean ($\lambda_{ex}$=435 nm, $\lambda_{em}$=475 nm) and in FRET-acceptor, mVenus ($\lambda_{ex}$=515 nm, $\lambda_{em}$=525 nm) were used. The source of mCerulean and mVenus were from pTriEx-mCerulean-Rac1 WT and pTriEx-mVenus-CBD constructs, respectively (*gifted from Prof. Klaus Hahn's Lab University of North Carolina Chapel Hill*). The h-Fyn gene source was from pRK5-c-Fyn (*a gift from Dr. Filippo Giancotti, Addgene plasmid # 16032*). Biosensors were constructed using gene fusion, quick change mutagenesis, and overlap extension PCR methods (*Kunkel et al., 1987*; *Zheng et al., 2004*; *Bryksin and Matsumura, 2010*). The key steps are shown in *Figure 3—figure supplement 1* and details of primers and constructs used in the study can be found in *Supplementary file 2*. Biosensors were either made in their original plasmid or in form of PCR fusion-product and were sub-cloned in the pTriEx-4neo vector (Novagen) under *Nco*I and *Bam*H1 site. The strategies used during biosensors construction are summarized here in a step wise manner.

 i. *FRET-donor*- Flexible linker of (GGGGS)$_3$ and its coding nucleotide sequence (5'-GGTGGAGGCGGTTCAGGCGGAGGTGGCTCTGGCGGTGGCGGATCG-3') was incorporated between 243–244 nucleotide of h-Fyn ORF of pRK5c-FynWT plasmid using primer SN 14, 15 with quick change mutagenesis method to yield plasmid pRK-UD-L$_{15}$-FynWT. In the next step mCerulean ORF was amplified from pTriEx-mCerulean-Rac1WT plasmid using primer SN 16, 17. Gel eluted product was used as a mega-primer in the overlap extension PCR using pRK-UD-L$_{15}$-FynWT plasmid as template to yield plasmid pRK-UD-L$_{15}$-mCerulean-L$_{15}$-FynWT. The ORF of this modified kinase was mutated (Y527F, constitutively active open kinase) with primer SN 18, 19 using site directed mutagenesis (SDM) to yield plasmid pRK-UD-L$_{15}$-mCerulean-L$_{15}$-FynCA. Further, these newly modified WT and CA ORFs were amplified using primer SN 20, 21. The product was digested with *Nco*I and *Bam*H1 and sub-cloned into pTriEx-4neo vector yielding plasmid p-UD-L$_{15}$-mCerulean-L$_{15}$-FynWT/CA. Several plasmids were also constructed for control studies. The ORF of unmodified Fyn kinase WT was amplified with primer SN 20, 21 from pRK5c-FynWT plasmid. The amplified product was sub-cloned in pTriEx-4neo vector under *Nco*I and *Bam*H1 site yielding plasmid p-FynWT. Using SDM we have made p-Fyn CA. mCerulean ORF was cloned at the 3' end of Fyn WT ORF flanked by flexible linker yielding plasmid p-UD-FynWT-L$_{15}$-mCerulean.

 ii. *FRET-acceptor*-mVenus and F29 ORF was amplified using primer SN 17, 22 and 24, 25, respectively. These amplified products have an overlap in the coding sequence of linker,

(*incorporated through PCR*). Using these products as a template and primer SN 22, 25, overlap PCR was performed. The fusion product mVenus-L$_{15}$-F29 was further sub-cloned to pTriEx-4neo vector under *Nco*I and *BamH*1 site to yield plasmid p-mVenus-L$_{15}$-F29. In this plasmid the myristoyl group (MGSSKSKPKDPS) was incorporated with primer SN 26, 27 using quick change mutagenesis yielding plasmid p-*myr*-mVenus-L$_{15}$-F29. We have also made the point mutant P41A in the F29 ORF of these acceptor biosensors with primer SN 10, 11 yielding plasmids p-mVenus-L$_{15}$-F29P41A and p-*myr*-mVenus-L$_{15}$-F29P41A respectively. The mCerulean, mVenus ORFs were amplified with primer SN 22, 23 from original source plasmids and further sub-cloned in pTriEx-4neo vector under *Nco*I and *BamH*1 to yield p-mCerulean, p-mVenus, respectively.

## Biosensor characterization

The over-expression, protein integrity and activity of the biosensor was analyzed in live cells. Following methods have been adapted:

i. *Transfection*- HEK-293T cells were cultured in 35-mm plate. Cells were transiently transfected with Lipofectamine LTX and plus reagent according to the manufacturer's instruction (Thermo Fisher Scientific).

ii. *Whole cell fluorescence measurement*- After 18 hr post transfection, cells were washed gently with 1X DPBS (Invitrogen) and re-suspended in 1 ml 1X DBPS. The live cell suspension was used to measure the excitation and emission spectra of fluorescent protein using spectrofluorometer (Fluorolog, Horiba). The slit width and PMT voltage was kept 2 nm and 950 V, respectively. The excitation and emission wavelength was kept identical to respective fluorescent protein as mentioned in methods.

iii. *Whole cell protein isolation*- the cells from suspension (*step ii*) were harvested by gentle centrifugation (1000 x g for 5 min at 4° C) and incubated in presence of ~300 µl of RIPA lysis and extraction buffer cocktail in −80℃ for overnight. The cells were lysed by gentle vortexing and centrifuged at 2000 x g for 20 min at 4° C in order to collect the whole cell lysate.

iv. *Immunoblot assay*- Protein was estimated from cell lysate (*step iii*) using BCA protein assay kit (Pierce). Cell lysate (50 µg) from samples was incubated in 1X laemmli buffer and denatured for 10 min at 95℃. Using these samples, we performed the SDS gel (8%) electrophoresis. The proteins from SDS gel were transferred onto PVDF membrane (Invitrogen, 0.2 µm) at 100V for 90 min at 4℃ in 1X transfer buffer. After complete transfer the PVDF membrane was incubated in blocking buffer (5% non-fat dry milk (Blotto) in 1X-TBST for 1 hr at room temperature on a shaker. The blot regions having protein of interests were incubated in hybridization bags with their specific primary antibody (5% BSA in 1X TBST) buffer at 4℃, overnight under mild shaking conditions. These membranes were washed stringently with 1X-TBST and incubated with anti-mouse or rabbit IgG, HRP-linked secondary antibody (1:3000) at room temperature for 1 hr. The membranes were then washed 4 times with 1X-TBST and imaged using Image-Quant LAS 4000 (GE-Healthcare) in luminol-based enhanced chemiluminescent reagent. The band intensity of blots was analysed using Image-J (NIH).

## Fyn knockdown (KD) using retrovirus-mediated RNA interference

The endogenous Fyn kinase protein of HEK-239T cells was knocked-down (KD) using Retrovirus-mediated RNA interference (sh-RNA) as described earlier (*Zhang et al., 2009*). pRetroSuper-shFyn construct has been used for endogenous Fyn knockdown which was a gift from Dr. Joan Massague (Addgene plasmid # 26985). In this case the Fyn kinase (XM_017010653.1) 3'UTR region: 5'—GAAC TTCCATGGCCCTCAT—3' was used as shRNA target sequence. For scrambled shRNA control, we have used pSUPER retro puro Scr shRNA, which was a gift from John Gurdon (Addgene plasmid # 30520) where the scrambled shRNA sequence: 5'—GCGAAAGATGATAAGCTAA—3'was used. For packaging of the retrovirus we have used AmphoPack-293 Cell Line (Gift from Dr Reety Arora/Prof Jyotsna Dhawan, inStem).~60–70% confluent AmphoPack-293 cells (grown in 10 cm culture plate, with 10 ml DMEM supplemented with 10% FBS) were transfected with 10 µg of shRNA DNA construct using Lipofectamine LTX and Plus reagent according to the manufacturer's instruction (Thermo-Fisher Scientific) in BSL2. Post-transfection the spent media containing packaged retrovirus were collected at 48 and 72 hr. The collected media (~18 ml) was filtered using 0.45 µm filter and used to infect the HEK-293T cells (10 cm plate,~70–80% confluent, grown in DMEM and 10% FBS) in presence of polybrene (10 µg/ml). The plate was further incubated for 24 hr and replaced with fresh

media having 4 µg/ml of puromycin (Sigma). After 48 hr, only ~3 to 4 cells (per 10 cm plate) were found viable in both cases. The media containing puromycin was changed as per requirement until these cells reached ~90% confluency. Finally, the puromycin selected cells were passaged in the media without puromycin and grown. The cells were tested for mycoplasma and found to be negative. The final batches from each case were frozen and stored in liquid N2. The knockdown of Fyn kinase was verified by immunoblot using specific antibodies.

## Probing extracellular signal-regulated kinases (ERK) phosphorylation

Earlier it has been shown that downstream ERK phosphorylation can be regulated through/is sensitive to Src family kinase 'Fyn' signaling (*Wary et al., 1998*). Therefore, ERK phosphorylation offers a sensitive measure of the signaling state. Here we aim to show- (a) cellular Fyn levels regulate downstream ERK signaling, (b) *FynSensor* Fyn is able to rescue downstream ERK signaling in Fyn-knockdown cells efficiently, and (c) *FynSensor,* labeled Fyn and F29 binder do not perturb downstream ERK signaling in multiple cell lines.

*Cellular Fyn levels regulate downstream signaling* - In order to show that ERK phosphorylation is sensitive to Fyn signaling, we used stable cells expressing Fyn sh-RNA (Fyn KD HEK cells) as well as scrambled shRNA (control HEK cells) in this experiment. The cells were seeded at a density of ~3×10$^5$/ml in 35 mm tissue culture dishes. After 24 hr, the cells were incubated in media containing 1% serum. 16 hr post incubation, the cells were washed with 1X DPBS and lysed in 200 µl of RIPA lysis buffer. The cell lysates were resolved on 10% SDS gel followed by transfer on a PVDF membrane. Blot region corresponding to 54 to 90 kDa was probed with anti-Fyn antibody, 29 to 54 kDa with anti-p44/42 MAPK (Erk1/2) antibody and anti-Phospho-p44/42 MAPK (Erk1/2) (Thr202/Tyr204) antibody. Vinculin was used as loading control (116 kDa). The band intensities from at least three such blots were analysed and the levels of phospho-/total ERK were compared using Graph-Pad Prism.

*FynSensor Fyn is able to rescue downstream ERK signaling in Fyn-knockdown cells efficiently* - In the FynKD-HEK cells, the ERK phosphorylation rescue assay was performed by transiently expressing vector (pTriex4 Neo) alone or unlabeled WT Fyn kinase or, *FynSensor* {FRET donor (mCer-Fyn) + FRET acceptor (myr-mVenus-F29)} in reduced serum (1%) media for 18 hr. Using cell lysates from these different conditions, we have performed the immunoblot experiment using specific antibodies as described above. The band intensities from at least three such blots were analysed and the levels of phospho-/total ERK were compared using GraphPad Prism. *FynSensor, labeled Fyn and F29 binder do not perturb downstream ERK signalling in multiple cell lines:*

*U2OS cells* - Cells were seeded at a density of ~3×10$^5$/ml cells/ml in 35 mm culture dishes containing 2 ml media. After 24 hr, cells were transiently transfected with vector, or myr-mVenus or myr-mVenus-F29 (FRET acceptor) or Fyn kinase donor (mCer-Fyn) or *FynSensor.* After 18 hr of transient protein expression, cells were lysed using RIPA buffers. The lysates were resolved on to 10% SDS gel followed by transfer on a PVDF membrane. Blot region corresponding to 54 to 90 kDa was probed with anti-Fyn antibody, 29 to 54 kDa with anti-p44/42 MAPK (Erk1/2) antibody and anti-Phospho-p44/42 MAPK (Erk1/2) (Thr202/Tyr204) antibody. Vinculin antibody was used as loading control (116 kDa). The band intensities from at least three such blots were analysed and the levels of phospho-/total ERK were compared using GraphPad Prism.

*Mouse myoblast C2C12 cells* - Cells were seeded at a density of ~3×10$^5$/ml cells/ml in 35 mm culture dishes containing 2 ml media. After 24 hr, cells were transiently transfected with vector, or myr-mVenus or myr-mVenus-F29 (FRET acceptor) or Fyn kinase donor (mCer-Fyn) or *FynSensor.* After 18 hr of transient protein expression, cells were lysed using RIPA buffers. The lysates were resolved on to 10% SDS gel followed by transfer on a PVDF membrane. Blot region corresponding to 54 to 90 kDa was probed with anti-Fyn antibody, 29 to 54 kDa with anti-p44/42 MAPK (Erk1/2) antibody and anti-Phospho-p44/42 MAPK (Erk1/2) (Thr202/Tyr204) antibody. β-actin was used as loading control (43 kDa).

## Immunofluorescence Assay

Protocol for immunofluorescence assay:

i.   *Coverslip coating* – A 24-well plate containing glass coverslips (10 mm diameter) was used to attach the cells. The coverslips were washed with 1X-DBPS and coated with Fibronectin (FN,

 10 µg/ml, in DPBS, 300 µl for one coverslip in a well). The plate was incubated for 60 min at 37˚C. The extra fibronectin was removed by aspiration and coverslips were washed with DPBS. Plate was either stored at 4˚C or used immediately for cell attachment.

 ii. *Cell attachment*- 70–90% confluent cells (transfected or un-transfected), were trypsinized (0.25% trypsin) and suspension was made in appropriate media. Approximately 50000 cells were seeded in 300 µl culture media onto fibronectin coated coverslips (in the 24-well plate). Cells were allowed to attach for 60 min in an incubator prior to further processing.

 iii. *Fixation*- Attached cells in the wells were washed once with 300 µl of 1X DPBS. 300 µl of paraformaldehyde solution (4% in 1X DPBS having 10% methanol) was added as fixative reagent and incubated under gentle rocking condition for 20 min at room temperature. The cells were then washed thrice with 300 µl with 1X DPBS.

 iv. *Permeabilization*– To each well, 300 µl of permeabilization solution (0.3% Triton-X100 in 1X DPBS) was added and incubated for 20 min under gentle rocking condition. Cells were then washed thrice with 300 µl of 1X DPBS.

 v. *Blocking*- To each well, 300 µl of blocking solution (1% BSA in 1X DBPS and 1% FBS) was added and incubated for 60 min at room temperature under gentle rocking condition.

 vi. *1˚ Antibody staining* – To each well, 200 µl of 1˚ Antibody [Anti-Fyn Mouse monoclonal antibody, Abcam # ab3116 IF Grade diluted in antibody dilution reagent (0.05% tritonX100 in 1X DBPS having 1% BSA, dilution 1:250)] was added and incubated at 4˚C overnight under gentle rocking condition. The following day, the cells were washed thrice with 300 µl of 1X DBPS for 5 min each under gentle rocking condition.

 vii. *2˚ Antibody staining*- To each well, 200 µl of secondary Antibody [Goat anti-mouse IgG secondary antibody, Alexa Fluor 647 conjugate, Invitrogen# 21236 diluted in antibody dilution reagent (0.05% triton-100 in 1X DBPS having 1% BSA, at 1:500 dilutions)] was added and incubated at room temperature under gentle rocking condition for 1 hr in dark. Following that, cells were washed thrice with 300 µl of 1X DBPS.

 viii. *Staining Nucleus*-300 µl of Hoechst 33342 nucleic acid stain (10 mg/ml) was added to a final dilution of (1:1000) in 1XPBS to each well and incubated for 30 min under gentle rocking condition in dark. Following that, cells were washed thrice with 300 µl of 1X DBPS for 2 min.

 ix. *Mounting*- Coverslips with fixed and stained cells were finally mounted onto a clean glass slide using 7 µl of Mowiol mounting medium and stored for 2 days for proper drying in dark. The slides were stored at 4˚C and used for confocal microscopy.

## Fibronectin coating of confocal dishes

Glass-bottom imaging dishes (Nunc Cat. No. 150682) were coated with 10 µg/ml of fibronectin (FN) solution. Briefly, the 1 mg/ml stock was diluted to the desired concentration in DPBS and the solution was added to the plates and incubated at 37˚C for an hour. Post incubation, plates were washed with DPBS and stored for immediate use at 4˚C.

## Fyn localization studies using immunofluorescence assay

Post 16 hr of transfection the HEK 293T Fyn KD (with *FynSensor* WT, or unlabeled WT Fyn) or HEK 293T control cells were plated on glass coverslip coated with FN and allowed to attach for 1 hr. The cells were then fixed and used for immunofluorescence studies (*see* Materials and methods *above*). Alexa Fluor-647 conjugated Fyn kinase protein complex was excited using 640 nm laser. In order to capture the spatial expression of Fyn, several optical slices along the z-axis were collected. The slice with maximum intensity for the emission corresponding to 640 nm excitation channel (specific for Alexa Fluor-647 conjugated Fyn kinase protein) was chosen, its intensity value was measured using Fiji. The data was analyzed using GraphPad Prism and shown in *Figure 3—figure supplement 9*.

## TIRF microscopy for visualizing Fyn localization

U2OS cells transiently expressing the *FynSensor* (WT-mCer-Fyn+mVenus-F29) were plated on FN coated glass bottomed dish and imaged on the Olympus IX83 inverted microscope coupled to the cellTIRF module fitted with an Okolab stage-top live-cell incubator to maintain cells at 37$^O$C at 5% $CO_2$.Cells were excited using the 445 nm laser and imaged using a 100X objective (Olympus UAPON 100X oil TIRF objective, NA = 1.49) at a penetration depth of 100 nm. Images were captured on the Evolve 512 Delta EMCCD camera at an exposure of 400 ms. Images were acquired using the Olympus cellSens software and processed using Fiji.

## Cell treatment and image acquisition for SU6656-APB experiment

The Acceptor photo-bleaching experiment on SU6656 treated cells was carried on the Olympus FV1000 confocal microscope using a 63X oil immersion objective. FRET-donor (mCer-Fyn) was excited using the 405 nm laser and emission from (425-500) nm was collected. Acceptor (myr-mVenus-F29) was excited using the 515 nm laser and emission from (519-619) nm was collected. For bleaching the acceptor, the 515 nm laser was used at a high laser dose (1.38 mW for 27 s). Post bleaching, both the donor and acceptor channel images were acquired. The data analysis was done as described in the Materials and methods section. Cells transfected with *FynSensor* were serum starved and plated as mentioned in Materials and methods. Cells were then treated with 2 µM of SU6656 inhibitor (2 mM stock in 100% DMSO). Post-inhibitor addition, bleaching of the acceptor was carried on. DMSO controls were treated and imaged in the same way.

## Cell treatment and imaging conditions

For single-cell imaging experiments, transient transfections in all the cells (U2OS, C2C12 and Fyn-KD-HEK 293T) were carried out using Lipofectamine 3000 according to the manufacturer's instruction (Life Technologies). Cells were checked for expression of plasmid of interest after 18 hr post transfection by fluorescence microscopy. Cells were then starved for 4–6 hr prior to being plated on glass-bottomed dishes coated with desired density of FN (see Materials and methods) and allowed to adhere for 30–40 min prior to imaging.

Serum-starved cells plated on FN were imaged for 25 min (acquisition settings are mentioned in following section) to record basal Fyn activity (activity in response to integrin activation/engagement) in a serum-starved state. For stimulation experiments, cells were initially imaged for 5–6 min to record basal Fyn activity. PDGF at a final concentration of 10 ng/ml (20 µg/ml stock in 1X DPBS) was then added onto the cells and images for an additional 15–16 min were acquired.

For faster imaging, serum-starved cells plated on FN were imaged for 10–15 min at an inter frame gap of ~35 s (acquisition settings are mentioned in the following section.

For treatment of cells with FAK inhibitor (PF-562271), serum-starved cells plated on FN were imaged for an initial period of 10 min and the inhibitor PF-562271 was added at a final concentration of 10 µM (10 mM stock in 100% DMSO). Post inhibitor addition cells were imaged for an additional 10 min. The cells were then treated with PDGF (10 ng/ml) and imaged for another 10 min.

## Confocal image acquisition

All imaging experiments (unless mentioned) were carried on the Olympus FV3000 microscope equipped with an PLANAPO N 60X oil immersion objective with a NA = 1.42 attached to a stage-top live-cell incubator to maintain cells at 37°C with 5% $CO_2$ (Tokai Hit). The microscope has two High Sensitivity Detectors (HSDs). The microscope also has a Z-Drift compensator and a motorized XY-stage which allows for multi-point-time-lapse imaging. Dex-Dem images were acquired using an excitation wavelength of 405 nm and emission was collected on HSD1 and the bandwidth of collection was from (437-499) nm. Aex-Aem images were acquired using an excitation wavelength of 514 nm and emission was collected on HSD2 and the bandwidth of collection was (528-628) nm. Dex-Aem images were acquired using donor excitation and emission was same as acceptor emission. Voltage and gain settings were kept same while imaging the same cell prior to and after any kind of treatment (Bleaching/addition of PDGF/addition of inhibitor). Two methods were used to study FRET response in cells. The details of the methods used are listed below:

Acceptor Photo-bleaching Method-For cells expressing *FynSensor* or non-binding mutant, pre-bleach and post-bleach Dex-Dem, and Aex-Aem images were acquired as described above. Acceptor bleaching was performed using 514 nm laser at 2X, 5X and 10 X light dosage for 10 s each (X = 0.069 mW) (refer to *Figure 4B & C*) and Dex-Dem, and Aex-Aem images acquired. For determining the amount and pattern of the increase in donor fluorescence post acceptor photo-bleaching, a customized MATLAB program was used. Dex-Dem (donor channel) images were used for this analysis. All the images were subjected to background subtraction and a thresholding operation before analysis. A user-defined global threshold was first used to generate a binary mask using the donor channel pre-bleach image of each cell. This was then used to determine the boundary of the cell using Canny's edge detection algorithm (*Canny, 1986*; *Mathworks, 2019a*). A centroid of each cellular mask was also determined. A 'linescan' algorithm was used to quantify the variation of donor

fluorescence recovery with distance from the cell periphery. For this, lines from every point on the periphery of the cell to the centroid were generated. Donor fluorescent intensities (from raw images) along these lines were calculated, and averages of all intensities were obtained as a function of distance from the periphery. For a given cell, the donor intensity values before and after bleaching (Pre-bleach$_{Donor}$ and Post-bleach$_{Donor}$) were thus determined and used in the following equation to calculate percentage recovery of donor fluorescence as per established methods (*Karpova et al., 2003*).

$$Donor\,recovery(\%) = \left(1 - \frac{\mathrm{IPre-bleachDonor}}{\mathrm{IPost-bleachDonor}}\right) * 100 \qquad (2)$$

where,

I Pre-bleach$_{Donor}$ - donor fluorescence intensity prior to bleaching.

I Post-bleach$_{Donor}$ - donor fluorescence intensity after bleaching.

For generating the representative ΔDonor image**s,** in *Figure 4A* the prebleach donor channel image was subtracted from the postbleach donor channel image of the same cell (ΔDonor = Post bleach$_{Donor}$-Pre-bleach$_{Donor}$) using MATLAB.

Donor Sensitized emission method-The Donor Sensitized Emission (SE) method records the FRET signal by measuring the amount of acceptor emission after excitation of the donor. The method used here has been modified from the original method as described earlier (*Gordon et al., 1998*) and requires the Dex-Dem, Dex-Aem and Aex-Aem images for calculation of FRET. To calculate the levels of both donor and acceptor bleed-through and direct acceptor cross-excitation using donor excitation, 'donor only' and 'acceptor only' samples were imaged in the following way:

i. Donor only sample: Dex-Dem and Dex-Aem
ii. Acceptor only sample: Dex-Aem and Aex-Aem

## Image analysis

All the image analyses were done using MATLAB and plots have been generated using Graph Pad Prism 5, unless otherwise specified. The customized codes written for the purpose of image analysis are available on GitHub.

## Generation of FRET index image

All images acquired were converted to 8-bit format prior to processing. FRET measurements were done on cells expressing both donor and acceptor fluorophores of *FynSensor* using *Equation 3*, followed by a median filtering with a $2 \times 2$ pixel block around each pixel (*Lim, 1990*). The Fiji plugin 'FRET analyzer' (*Hachet-Haas et al., 2006*) was used to determine the bleed through values used in *Equation 3* from donor-only and acceptor-only cells.

FRET index images are shown using ImageJ (NIH) *Fire* LUT and scaled by an arbitrary number only for visual representations.

Briefly, for our sensitized FRET experiments, the following equation was used to generate the FRET index images:

$$\mathrm{FRET\,index\,image} = \mathrm{I_{FRET}} - (\alpha_D \mathrm{I_{Donor}}) - (\alpha_A \mathrm{I_{Acceptor}}) \qquad (3)$$

Wherein:

$\alpha_D$ – is the mean of the slope of the bleed-through signal of the donor alone (here 0.55),

$\alpha_A$ - is the mean of the slope of the bleed-through signal of the acceptor alone (here 0.028),

$I_{FRET}$ - is the FRET channel image,

$I_{Donor}$ – is the Donor channel image, and,

$I_{Acceptor}$ – is the Acceptor channel images.

FRET index images depict total FRET (FRET$_T$)

Calculations used are consistent with other literature reports on intermolecular FRET measurements, like the Fiji plugin 'FRET analyzer' (*Hachet-Haas et al., 2006*) and another very recent report (*Oldach et al., 2018*).

## Quadrant analysis

For cellular area estimation, acceptor channel image time series (Aex-Aem) for each cell was processed, (global thresholding, smoothening by erosion operation and filling holes using **imfill** and removing small area particles by **bwareafilt**). Cellular area processed images (binary images with a value of 1 inside cell and 0 outside) hence generated, were used as masks to further process the FRET index images (or 'YFP channel' or 'CFP channel images as the case may be (*Figure 5—figure supplement 2A* and *Figure 5—figure supplement 3B*). Cell centroids were determined on the binary masks using regionprops function. With respect to the centroid, a vertical and horizontal line was drawn to divide each cell into four quadrants as shown in *Figure 5A–I*. Mean fluorescent intensity (either FRET or 'CFP' or 'YFP') in each quadrant was calculated using *Equation 4*; wherein sum of the fluorescent intensity values from all pixels in a given quadrant is divided by total number of pixels in the quadrant. The mean intensity is followed over time and the variation is plotted in *Figure 7A–II,B–II,C–II*.

$$Mean\,intensity = \frac{\sum_{i=1}^{n} Pixel\,Value_i}{n} \tag{4}$$

n = no. of pixels in the specific quadrant

## Quantifying spatio-temporal Fyn activity patterns

For testing if Fyn activity is spatially enhanced, mean $FRET_T$ values (see above) for every quadrant in each cell was analyzed over time. Max-$FRET_T$-HFQ (HFQ: high-FRET quadrant) refers to the maximum value of mean quadrant FRETT seen in the time series, and was determined for each cell. Max-$FRET_T$-LFQ (LFQ:low FRET quadrant') was the corresponding mean $FRET_T$ value for lowest FRET quadrant at that specific time point, for the given cell. Max-$FRET_T$-HFQ and Max-$FRET_T$-LFQ were averaged over multiple cells for both serum-starved as well as PDGF-stimulated cells.

For calculating PDGF-induced enhancement in Fyn activity, difference in the FRET intensity levels (Max-$FRET_T$-HFQ) before and after PDGF stimulation were calculated. Also calculated were differences in Max-$FRET_T$-LFQ before and after PDGF.

## Characteristic time period of FRET pulses

Power spectral density (PSD) (*Proakis John, 1996*) of the time series was computed using **periodogram** function. Prior to subjecting the $FRET_T$ time traces to frequency decomposition analysis, we removed any quadratic trend using the **detrend** function in line with established procedures (*Mitov, 1998*; *Colak, 2009*). The frequency against which maximum power (*Proakis John, 1996*; *Mathworks, 2019b*) is obtained, known as dominant frequency in the time series, was used to calculate the time-period of $FRET_T$ intensity oscillations in each quadrant. The frequency is converted to time period based on the sampling frequency, and is plotted in the *Figure 7D*.

## Quantifying active Fyn localization at membrane

The maximum FRET intensity region of cell membrane was identified through line scanning.

An arc of arbitrary length unit (40 pixels here) on the membrane encompassing this region of maximum FRET intensity was selected, and a sector was constructed which spanned the membrane at the cell edge and extended towards the centroid of the cell. The intensity in this sector was measured over time and plotted as a function of distance from the cell edge (*Figure 7E*).

## Cell morpho-dynamics analysis

Donor channel images (Dex-Dem) were used to generate a cell mask and calculate the cellular area at each time point (see above). Two consecutive cell mask images were used to determine the temporal change in the cellular area($\Delta A_i = A_{i+1} - A_i$). The absolute value of this temporal change in cell area was then normalized for the initial cell area ($|\Delta A_i|/A_1$) to account for differently sized cells. This was then used to calculate the mean fractional area change to reflect the cumulative area changes over a period of time (*Equation 5a*). Similarly, cumulative cell perimeter changes were determined using the modulus of change in perimeter between consecutive cell frames, normalized for the initial cell boundary length (*Equation 5b*).

$$Mean\,Fractional\,Area\,Change = \frac{\sum_{1}^{n-1}\frac{|\Delta A_i|}{A_1}}{n-1} \qquad (5a)$$

$$Mean\,Fractional\,Perimeter\,Change = \frac{\sum_{1}^{n-1}\frac{|\Delta P_i|}{P_1}}{n-1} \qquad (5b)$$

In the above equation, n is total numbers of frames recorded. Once mean fractional area and perimeter changes were determined for a single cell, averages from multiple such cells were then calculated for each of the parameters in each set has been plotted *Figure 3—figure supplement 7*.

### FRET intensity and membrane motility correlation

We have used the open source plugin ADAPT, developed by *Barry et al. (2015)* quantitatively examine the correlation between membrane motility and FRET intensity. For this purpose, we use the acceptor channel image (Aex-Aem) as 'reference channel' for demarcating the cell boundary. FRET index image (obtained as per *Equation 3*) is used as the 'signal channel' (FRET intensity/Fyn activity). The values used for resolution (micron/pixel) and frames/min are 0.2 and 2, respectively. The 'signal' signifying Fyn activity approximately 1 μm from the membrane and the 'velocity' at the membrane along the cell periphery and over time, are obtained using the plugin. The overall 'membrane motility' map is obtained by performing a modulus operation on the velocity map. 3D plot (*Figure 9A*) shows changes in 'signal' and 'membrane motility' over time and position along cell boundary for a single cell.

To compute a global correlation between FRET intensity (Fyn activity) and 'membrane motility' a one-to-one mapping of these two arrays extracted from ADAPT plugin, was performed. For the purpose of quantification, we have binned the FRET intensity values into bins of 3 intensity units [(9-12 , 12-15 , 15-18)] and so on. The curve-fitting was done using the polynomial cubic equation (f = y0 +a*x+b*x2+c*x3) on SigmaPlot (see *Figure 9B*).

Key resource table: Provided as *Supplementary file 3* of this manuscript.

## Acknowledgements

This research was supported by the Dept. of Biotechnology (DBT), Ministry of Science and Technology, Govt. of India, Grant # BT/PR8071/MED/32/290/2013 to AG. Additional support came from grants #BT/IN/BMBF/03/AG/2015–16 and BT/IN/Denmark/07/RSM/2015–2016 from DBT, India to AG. Financial and institutional support from inStem and DBT, India is gratefully acknowledged. The research was partly funded by financial support from the National Centre for Biological Sciences, Tata Institute of Fundamental Research and the DBT-Ramalingaswamy fellowship (BT/HRD/23/02/2006) to RD. We thank Professor S Ramaswamy and the Technology for the Advancement of Science (TAS) theme for support. AM was supported by The University Grants Commission (UGC), India with JRF-SRF fellowships. Financial support from the DBT-RA Program to RS in Biotechnology and Life Sciences is gratefully acknowledged. We thank Prof. Klaus Hahn (University of North Carolina, Chapel Hill) and Professor Brian Kay (University of Illinois, Chicago) for various DNA constructs. We thank Professor Jyotsna Dhawan (inStem,CCMB), Professor Satyajit Mayor (NCBS) and Professor Apurva Sarin (inStem) for sharing cell lines and reagents. We thank Professor Sudhir Krishna (NCBS) and Dr. Abhijit Majumder (inStem,IIT-Bombay) for sharing reagents.We thank Dr. Sandeep Krishna, Dr. Debshankar Banerjee (NCBS) and Prof. Sugata Munshi (Jadavpur University) for helpful discussion regarding image analysis. We thank Dr. Colin Jamora (inStem), Dr. Sunil Laxman and Dr. Arati Ramesh (NCBS) for critical reading of the manuscript and helpful discussions. We thank Dr. Reety Arora, Nishan Shettigar, other members of the Gulyani lab for discussions and technical help. We thank Central Imaging and Flow Cytometry Facility of NCBS and inStem, especially Dr. H Krishnamurthy, Dr. Manoj Mathew and Dr. Feroz MH Musthafa for support with flow-cytometry and microscopy experiments. We thank the Olympus and DSS Imagetech team for their assistance. The NMR spectra were collected at the NMR Facility at the National Centre for Biological Sciences, which is

partially supported by the B-Life grant from DBT (DBT/PR12422/MED/31/287/2014). We thank the sequencing facility, PC Gautam and the instrumentation team and lab support at NCBS-inStem.

## Additional information

### Funding

| Funder | Grant reference number | Author |
|---|---|---|
| University Grants Commission | JRF-SRF Fellowship | Ananya Mukherjee |
| Department of Biotechnology, Ministry of Science and Technology | DBT-RA Program in Biotechnology and Life Sciences | Randhir Singh |
| Department of Biotechnology, Ministry of Science and Technology | Grant BT/PR8071/MED/32/290/2013 | Akash Gulyani |
| Department of Biotechnology, Ministry of Science and Technology | BT/IN/BMBF/03/AG/2015-16 | Akash Gulyani |
| Department of Biotechnology, Ministry of Science and Technology | BT/IN/Denmark/07/RSM/2015-2016 | Akash Gulyani |

The funders had no role in study design, data collection and interpretation, or the decision to submit the work for publication.

### Author contributions

Ananya Mukherjee, Randhir Singh, Conceptualization, Resources, Data curation, Formal analysis, Funding acquisition, Validation, Investigation, Visualization, Methodology; Sreeram Udayan, Conceptualization, Resources, Formal analysis, Methodology; Sayan Biswas, Conceptualization, Resources, Data curation, Formal analysis, Validation, Methodology; Pothula Purushotham Reddy, Shilpa Kumar, Resources, Data curation, Formal analysis, Validation, Methodology; Saumya Manmadhan, Geen George, Resources, Formal analysis, Methodology; Ranabir Das, Balaji M Rao, Conceptualization, Resources, Data curation, Formal analysis, Supervision, Validation, Investigation, Methodology; Akash Gulyani, Conceptualization, Resources, Data curation, Formal analysis, Supervision, Funding acquisition, Investigation, Methodology, Project administration

### Author ORCIDs

Ananya Mukherjee (ID) https://orcid.org/0000-0002-3112-583X
Randhir Singh (ID) https://orcid.org/0000-0001-7388-225X
Sreeram Udayan (ID) https://orcid.org/0000-0003-3778-9968
Sayan Biswas (ID) https://orcid.org/0000-0003-2515-1940
Ranabir Das (ID) http://orcid.org/0000-0001-5114-6817
Balaji M Rao (ID) https://orcid.org/0000-0001-5695-8953
Akash Gulyani (ID) https://orcid.org/0000-0003-4130-4257

### Decision letter and Author response

Decision letter https://doi.org/10.7554/eLife.50571.sa1
Author response https://doi.org/10.7554/eLife.50571.sa2

## Additional files

### Supplementary files

• Supplementary file 1. Structural statistics and list of interactions observed in the Fyn-SH3:F29 complex.

• Supplementary file 2. List of constructs and primers used in the study.

- Supplementary file 3. Key Resources Table.
- Transparent reporting form

### Data availability

The coordinates of FYN-SH3/ Monobody-binder (F29) complex and associated NMR chemical shift data are deposited in the PDB under accession codes 5ZAU (http://www.rcsb.org/structure/5ZAU) . Source data files provided for Figure 1, Figure 3, Figure 4, Figure 5, Figure 6, Figure 7, Figure 8, Figure 9 and Figure 2—figure supplement 3, Figure 3—figure supplement 2, Figure 3—figure supplement 4, Figure 3—figure supplement 5, Figure 3—figure supplement 6, Figure 3—figure supplement 7, Figure 3—figure supplement 9, Figure 3—figure supplement 10, Figure 3—figure supplement 11, Figure 4—figure supplement 1, Figure 5—figure supplement 1, Figure 5—figure supplement 2, Figure 5—figure supplement 3, Figure 5—figure supplement 4, Figure 6—figure supplement 1, Figure 6—figure supplement 2, Figure 7—figure supplement 1, Figure 7—figure supplement 2, Figure 7—figure supplement 3, Figure 8—figure supplement 1. MATLAB codes developed for image processing and data analysis is available on GitHub (https://github.com/sayan08/FynBS_inStem; copy archived at https://github.com/elifesciences-publications/FynBS_inStem). Plasmid constructs used in the study are available from the corresponding authors upon reasonable request.

The following dataset was generated:

| Author(s) | Year | Dataset title | Dataset URL | Database and Identifier |
|---|---|---|---|---|
| Reddy PP, Gulyani A, Das R | 2019 | Complex of the human FYN SH3 and monobody binder | http://www.rcsb.org/structure/5ZAU | RCSB Protein Data Bank, 5ZAU |

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
