## [Decision Letter]

**Acceptance summary:**

This manuscript describes a novel, two-component FRET sensor of Fyn tyrosine kinase. Screening a library generated by mutagenizing the Sso7d protein from the hyperthermophilic archaeon Sulfolobus solfataricus for binders of the Fyn SH3 domain the authors identified a clone and characterized the binding interface using NMR spectroscopy. They further validated key interfacial interactions that mediate the specificity via site-directed mutagenesis. Subsequently, the authors show that this SH3 binder preferentially binds to open, active Fyn vs. wt Fyn. The authors then develop the FRET sensor as a combination of WT Fyn (donor) and myristoylated sensor (acceptor) and observed polarized activity in multiple cell types, with active Fyn concentrated to local cellular zones and also fluctuating over time (~20 min) in serum-starved cells. They demonstrate that this FRET signal can be globally inhibited using a Fak inhibitor, and increased locally at regions that already have high FRET intensity using PDGF stimulation. Pre-treatment of starved cells with the FAK inhibitor reduces FRET signal and blocks stimulation by subsequent addition of PDGF. Overall, this study integrates an impressive combination of innovative technology development and provocative new insights into cell signaling.

**Decision letter after peer review:**

Thank you for submitting your article "A Fyn-specific biosensor reveals localized, pulsatile kinase activity and spatially regulated signaling crosstalk" for consideration by *eLife*. Your article has been reviewed by three peer reviewers, including Volker Dötsch as the Reviewing Editor and Reviewer #1, and the evaluation has been overseen by Philip Cole as the Senior Editor. The following individual involved in review of your submission has agreed to reveal their identity: Tobias Meyer (Reviewer #3).

The reviewers have discussed the reviews with one another and the Reviewing Editor has drafted this decision to help you prepare a revised submission.

Summary:

In this manuscript Mukherjee et al. describe a novel, two-component FRET sensor of Fyn kinase. The authors screened for binders of the Fyn SH3 domain and validated the F29 clone binding through pulldown experiments, then further characterized the binding interface using NMR and suggest key interfacial interactions that mediate the specificity via site-directed mutagenesis of F29. Subsequently, the authors develop the FRET sensor as a combination of WT Fyn (donor) and myristoylated F29 (acceptor) and observe polarized activity in multiple cell types, with active Fyn concentrated to local cellular zones and also fluctuating over time (~20 min) in serum-starved cells plated on fibronectin. They demonstrate that this FRET signal can be globally inhibited using a Fak inhibitor, and increased locally at regions that already have high FRET intensity using PDGF stimulation. Pre-treatment of starved cells with the FAK inhibitor reduces FRET signal and blocks stimulation by subsequent addition of PDGF. Finally, the authors note a spatial trend that cell quadrants with higher Fyn FRET signal tend to have smaller changes in area.

Essential revisions:

1) The functional characterization of the biosensor is well supported by the provided data. However, in all images showing activity in different cell types, cells always appear to show higher activity in the right-hand side of the image. Such unlikely coincidence may potentially indicate an imaging artifact. The authors should analyze the distribution of activity in different cells and provide images of activation accruing at different sites in the cells.

2) The authors should show a better representative blot in Figure 2—figure supplement 3C (similar to what they show in Figure 1E).

3) At the end of the section entitled "Molecular basis of specific Fyn recognition by F29", the authors discuss the importance of the D99 residue in the SH3 domain of Fyn. However, their analysis shows that SH3 domains of Yes, Lyn and Fgr have D/E at the same position and yet they show no binding to F29. How would the authors explain this discrepancy?

4) Subsection “F29 enables construction of *FynSensor*, a FRET biosensor for active Fyn”. Authors demonstrate that "…increasing the amount of intracellular mVenus-F29 while maintaining constant expression of mCer-Fyn, led to a concomitant increase in FRET signal, followed by saturation of this FRET increase." Then they stated that "Here, the saturation in FRET signal clearly indicates that even increasing concentrations of F29 binder in cells do not simply cause an inadvertent or artefactual activation of the kinase." It is not clear how saturation of FRET with increasing concentration of F29 would suggest that F29 cannot cause activation of the kinase. Saturation could also occur due to full activation of the kinase.

5) The anti-GFP blot in the Figure 3—figure supplement 10Cshows that myr-mVenus-F29 is expressed in all samples. The authors acknowledge that this is due to nonspecific recognition by the antibody. How could the authors be sure that myr-mVenus-F29 is expressed and at what level?

6) Figure 3—figure supplement 9. In the figure legend, the authors stated "For *FynSensor* imaging, we have ectopically expressed labeled Fyn in HEK-293T cells where endogenous Fyn has been knocked down. This ensures that labeled Fyn levels are comparable to endogenous Fyn levels." However, their images and quantification show that labeled Fyn levels are significantly higher than endogenous Fyn. Furthermore, localization of endogenous Fyn appears to be different from ectopically expressed labeled Fyn. This discrepancy should be addressed. Also, images of phospho-Erk show very weak signal that looks like nonspecific background staining. Is this typical for phospho-Erk localization to be in sparse puncta? How does total Erk staining look for these cells? Is this assessment of phospho-Erk localization necessary?

7) It is surprising that the quadrant analysis gives such stark results, despite the fact they are averaging over large portions of the cell. Often it seems that polarized regions are much smaller than a quarter of the cell. Some sort of more granular analysis could strengthen some of the relationships. Especially for Figure 10, the authors must show a cleaner spatio-temporal relationship to make claims about relationships to membrane behavior. Using a FIJI plugin such as ADAPT (Barry et al. (2015) JCB) could answer these questions much more cleanly.

8) It is also important that they make sure to not under-sample temporally when trying to relate the edge dynamics with the biosensor. Imaging faster than every minute might be necessary to properly relate protrusion/retraction behavior with Fyn activity. Currently the relationship seems too noisy at the single trace level to make significant claims.

9) The authors suggest that Fyn activity is regulated by PDGF and integrates signaling from both PDGF and FAK. These claims would be supported by additional work showing the time-traces of FRETT signal of individual cells before and after PDGF addition (e.g., as they did in Figure 8 (ii)).

---

## [Author Response]

Essential revisions:1) The functional characterization of the biosensor is well supported by the provided data. However, in all images showing activity in different cell types, cells always appear to show higher activity in the right-hand side of the image. Such unlikely coincidence may potentially indicate an imaging artifact. The authors should analyze the distribution of activity in different cells and provide images of activation accruing at different sites in the cells.

We do not observe any directionality to the region that gets activated. This issue has now been addressed in the revised manuscript with the new imaging data now showing cells with FRET patterns not necessarily restricted to the right-hand side. This data is provided in new Figure 7—figure supplement 1,2 and 3.

2) The authors should show a better representative blot in Figure 2—figure supplement 3C (similar to what they show in Figure 1e).

To address this concern, we have repeated the immunoblot experiment. As advised, we have now provided an improved representative blot (similar to the one in Figure 1E). The relevant figure has been changed in the revised manuscript (Figure 2—figure supplement 3C).

3) At the end of the section entitled "Molecular basis of specific Fyn recognition by F29", the authors discuss the importance of the D99 residue in the SH3 domain of Fyn. However, their analysis shows that SH3 domains of Yes, Lyn and Fgr have D/E at the same position and yet they show no binding to F29. How would the authors explain this discrepancy?

The reviewer is right in suggesting that our sequence comparisons/analyses as well as our current results suggest that while D99 may contribute to binding specificity of F29 towards Fyn vis-à-vis other SFKs, this residue is unlikely to be the sole determinant of specificity (for the reasons listed above). While we had indicated this in our original manuscript, we have significantly reworded this section to clarify this point. For instance; we clearly state: “Therefore, we hypothesized that D99 in Fyn-SH3 may contribute to the binding specificity of F29 for Fyn-SH3 vis-à-vis to at least some of the SFK SH3 domains. To test this hypothesis, we generated a mutant version of the SH3 domain from Src (Src-SH3) wherein the native threonine was replaced with aspartic acid (T99D). Strikingly, T99D Src-SH3, but not wild-type Src-SH3, showed detectable binding to F29 in pulldown assays (Figure 2—figure supplement 3C). These results suggest that D99 in Fyn-SH3 contributes to the binding specificity of F29 for Fyn over other SFKs, notably Src. Nevertheless, specificity of F29 for Fyn-SH3 over Yes-SH3 (containing E99) (Figure 1C, E) and Lyn/Fgr (containing D99, Figure 1—figure supplement 2), suggests that binding specificity results from a unique combination of multiple interactions at the binding interface that may require further investigation.”

4) Subsection “F29 enables construction of FynSensor, a FRET biosensor for active Fyn”. Authors demonstrate that "…increasing the amount of intracellular mVenus-F29 while maintaining constant expression of mCer-Fyn, led to a concomitant increase in FRET signal, followed by saturation of this FRET increase." Then they stated that "Here, the saturation in FRET signal clearly indicates that even increasing concentrations of F29 binder in cells do not simply cause an inadvertent or artefactual activation of the kinase." It is not clear how saturation of FRET with increasing concentration of F29 would suggest that F29 cannot cause activation of the kinase. Saturation could also occur due to full activation of the kinase.

We agree with the reviewer that a mere saturating FRET signal does not show that F29 binder does not cause artefactual activation of kinase in cells. To clarify this point, we have performed new immunoblot experiments to show that expression of F29 does not cause artefactual activation of kinase as measured through the ratio of active Fyn (pTyr416) to total Fyn. These results are now shown in Figure 3—figure supplement 6 of the revised manuscript.

Additionally, immunoblot experiments show that the expression levels of F29 does not cause any change in the levels of activated ERK (a sensitive downstream signaling readout of Fyn activity) in multiple cell-types (as seen by pERK/total ERK levels) (Figure 3—figure supplement 5,10,11). Further, we also show that F29 expression has no effect on Fyn kinase localization (Figure 3—figure supplement 8,9) and cellular morphodynamics (Figure 3—figure supplement 7).

All these results taken together strongly suggest that binder does not cause activation of kinase.

To ensure clarity on this issue, we have reworded the relevant sentences which now read:

“Here, the saturation in FRET signal suggests that even increasing concentrations of F29 binder in cells does not cause an inadvertent or artefactual activation of Fyn kinase, especially since F29 binder expression in cells does not lead to any measurable changes in autocatalytic activity of Fyn (Y-420 autophosphorylation) (Figure 3—figure supplement 6).”

5) The anti-GFP blot in the Figure 3—figure supplement 10C shows that myr-mVenus-F29 is expressed in all samples. The authors acknowledge that this is due to nonspecific recognition by the antibody. How could the authors be sure that myr-mVenus-F29 is expressed and at what level?

We have addressed this issue in the revised manuscript by providing new data using a different, more specific anti-GFP antibody (see Materials and methods of the revised manuscript). The relevant immunoblot experiments have been repeated with this specific anti-GFP antibody. Figure 3—figure supplement 10C of the revised manuscript confirms that there are no non-specific bands in U2OS cell lysate. Specific bands corresponding to ~33kDa are only observed in lysates of cells expressing either myr-Venus or myr-Venus-F29 or *FynSensor* constructs but not in vector control cells.

Additionally, we have also quantified fluorescence images of cells expressing either myr-Venus or myr-Venus-F29 or *FynSensor* constructs. The quantification of mVenus fluorescence shows that all the constructs are expressed at similar levels in the cells. These new data have been added to the revised Figure 3—figure supplement 10C of the manuscript.

6) Figure 3—figure supplement 9. In the figure legend, the authors stated "For FynSensor imaging, we have ectopically expressed labeled Fyn in HEK-293T cells where endogenous Fyn has been knocked down. This ensures that labeled Fyn levels are comparable to endogenous Fyn levels." However, their images and quantification show that labeled Fyn levels are significantly higher than endogenous Fyn. Furthermore, localization of endogenous Fyn appears to be different from ectopically expressed labeled Fyn. This discrepancy should be addressed. Also, images of phospho-Erk show very weak signal that looks like nonspecific background staining. Is this typical for phospho-Erk localization to be in sparse puncta? How does total Erk staining look for these cells? Is this assessment of phospho-Erk localization necessary?

We thank the reviewers for their valuable suggestions and agree that the assessment of phospho-ERK localization may not be necessary. This portion has therefore been removed from the revised manuscript.

Additionally, we have performed extensive new imaging experiments in Fyn-knockdown HEK-293 cells ectopically expressing either unlabelled-WT Fyn or mCer-Fyn. Data from these cells has then been compared to cells expressing endogenous Fyn for estimating total Fyn protein levels as well as its localization.

Estimation of protein levels from immunoblot analysis as well as fluorescence micrographs shows that ectopic Fyn kinase in Fyn-KD cells is moderately overexpressed as compared to the endogenous kinase. Manuscript has been revised to indicate this.

Importantly though, this moderately overexpressed Fyn localizes similar to the endogenous kinase as seen in the immunofluorescence confocal micrographs. These new data have now been included in the revised manuscript (Figure 3—figure supplement 9).

The reviewers may also note that we have systematically examined the Fyn activity patterns as revealed by *FynSensor* across cells expressing different levels of labelled Fyn. We find that the conserved patterns of Fyn activity are not impacted by changes in *FynSensor* expression levels. In all three cell types examined (including knockdown rescue), the spatial patterns remain conserved independent of the precise levels of labeled Fyn (mCer-Fyn) as well as the binder (mVenus-F29) (Figure 6C and Figure 6—figure supplement 2).

7) It is surprising that the quadrant analysis gives such stark results, despite the fact they are averaging over large portions of the cell. Often it seems that polarized regions are much smaller than a quarter of the cell. Some sort of more granular analysis could strengthen some of the relationships. Especially for Figure 10, the authors must show a cleaner spatio-temporal relationship to make claims about relationships to membrane behavior. Using a FIJI plugin such as ADAPT (Barry et al., 2015) could answer these questions much more cleanly.

We thank the reviewer for their comments and valuable suggestions. As the reviewers note that even a course-grained image analysis is able to pick up localized kinase activity patterns, thus showcasing the efficacy of the sensor as well as the clear compartmentalization of active Fyn. However, as part of this review we have performed extensive new imaging experiments at faster speeds and carried out detailed new image analysis to clarify this issue. Indeed, using the ADAPT plugin for a more granular analysis we are able to better correlate the membrane motility speed to the observed FRET_T_ signal. The new data are included as part of Figure 9 in the revised manuscript. The data generated through faster imaging (35 sec vs 1 min per frame) and the use of ADAPT plugin allows us to see a more robust relationship between membrane motility and FRET signal at the level of both individual cells (e.g Figure 9A) as well trends observed across multiple cells (Figure 9B).

8) It is also important that they make sure to not under-sample temporally when trying to relate the edge dynamics with the biosensor. Imaging faster than every minute might be necessary to properly relate protrusion/retraction behavior with Fyn activity. Currently the relationship seems too noisy at the single trace level to make significant claims.

As suggested by the reviewers we have performed new *FynSensor* imaging experiments at faster acquisition speeds. This faster imaging combined with a *more granular* image analysis regime (using ADAPT) has indeed helped us in establishing a stronger inverse correlation between the edge dynamics and observed FRET_T_ signal. These new data have been incorporated in Figure 9 of the revised manuscript.

9) The authors suggest that Fyn activity is regulated by PDGF and integrates signaling from both PDGF and FAK. These claims would be supported by additional work showing the time-traces of FRETT signal of individual cells before and after PDGF addition (e.g., as they did in Figure 8 (ii)).

We thank the reviewers for their query. In the revised manuscript we provide individual time traces of cells both prior to and after treatment with the inhibitor as well as PDGF. The individual FRET index images and time traces have been included in Figure 8—figure supplement 1 of the revised manuscript.